# A Note on Some Statistical Properties of Signature Transform Under Stochastic Integrals

## Abstract

Signature transforms are iterated path integrals of continuous and discrete-time time series data, and their universal nonlinearity linearizes the problem of feature selection. This paper revisits some statistical properties of signature transform under stochastic integrals with a Lasso regression framework, both theoretically and numerically. Our study shows that, for processes and time series that are closer to Brownian motion or random walk with weaker inter-dimensional correlations, the Lasso regression is more consistent for their signatures defined by Itô integrals; for mean reverting processes and time series, their signatures defined by Stratonovich integrals have more consistency in the Lasso regression. Our findings highlight the importance of choosing appropriate definitions of signatures and stochastic models in statistical inference and machine learning.

## 1 Introduction

**Signature transform.** Originally introduced and studied in algebraic topology (Chen, 1954; 1957), the signature transform, sometimes referred to as the path signature or simply signature, has been adopted and further developed in rough path theory (Lyons et al., 2007; Friz & Victoir, 2010). Given any continuous or discrete time series, the signature transform produces a vector of real-valued features that extract rich and relevant information (Morrill et al., 2020a; Lyons & McLeod, 2022). It has been proven an attractive and powerful tool for feature generation and pattern recognition with state-of-the-art performance in a wide range of domains, including handwriting recognition (Yang et al., 2016b;c; Wilson-Nunn et al., 2018; Kidger et al., 2020; Ibrahim & Lyons, 2022), action recognition (Yang et al., 2016a; Li et al., 2017; Fermanian, 2021; Lee et al., 2022; Yang et al., 2022; Cheng et al., 2023), medical prediction (Kormilitzin et al., 2017; Morrill et al., 2019; Moore et al., 2019; Morrill et al., 2020b; Pan et al., 2023; Bleistein et al., 2023), and finance (Gyurkó et al., 2013; Lyons et al., 2014; Arribas, 2018; Lyons et al., 2019; Kalsi et al., 2020; Salvi et al., 2021; Akyildirim et al., 2022; Cuchiero et al., 2023; Futter et al., 2023; Lemahieu et al., 2023). Comprehensive reviews of successful and potential applications of the signature transform in machine learning can be found in Chevyrev & Kormilitzin (2016) and Lyons & McLeod (2022).

Most of the empirical success and theoretical studies of the signature transform are built upon its most striking *universal nonlinearity* property. It states that every continuous function of the time series may be approximated arbitrarily well by a linear function of its signature (see Section 2.1). Empirical studies (Levin et al., 2016; Lyons & McLeod, 2022; Pan et al., 2023; Bleistein et al., 2023) demonstrate that the nonlinearity property gives the signature several advantages over end-to-end neural-network-based nonlinear methods (Ahmed et al., 2023). First, training linear models of signatures is more computationally efficient than neural networks; second, the linear model allows for interpretability.[1] When learning nonlinear relationships between variables, utilizing linear regression models after applying the signature transform can yield significantly improved out-of-sample prediction performance compared to modeling without the transform.

Signatures are iterated path integrals of time series, and there are multiple definitions of integrals adopted for signatures. Despite the rapidly growing literature on the *probabilistic* characteristics of

---

[1]The signature transform can be understood as a method of feature engineering, and can be interpreted based on the geometric shape of the underlying path; see Appendix A.

signatures and the successful application of the signature transform in machine learning, the *statistical* properties of the signature method are often overlooked. Furthermore, universal nonlinearity can be expressed under different definitions of signatures (see Appendix B), raising the question of which definition has better statistical properties for different processes and time series. To our knowledge, most empirical studies in the literature use a default definition regardless of the specific scenarios being studied. However, using an inappropriate signature definition may lead to suboptimal performance. Therefore, it is time to understand and systematically study the statistical implications of these different forms of signatures on a given time series data.

**Consistency of Lasso.** In practice, feature selection methods like the Lasso (Tibshirani, 1996) are commonly used to identify a sparse set of features from a universe of all signatures (Lyons, 2014; Chevyrev & Kormilitzin, 2016; Levin et al., 2016; Moore et al., 2019; Sugiura & Hosoda, 2020; Sugiura & Kouketsu, 2021; Lemercier et al., 2021; Lyons & McLeod, 2022; Lemahieu et al., 2023; Bleistein et al., 2023; Cuchiero et al., 2023). One of the well-documented and extensively studied issues concerning linear models is the consistency in feature selection by Lasso (Zhao & Yu, 2006; Bickel et al., 2009; Wainwright, 2009). Consistency is an important metric for out-of-sample model performance. Given the different definitions of signatures, the natural starting point is the consistency issue for Lasso regression models under different signature transforms.

**Main results.** This paper studies the consistency issue of Lasso for signature transforms. It focuses on two definitions of signatures: Itô and Stratonovich. It chooses two representative classes of Gaussian processes: multi-dimensional Brownian motion and Ornstein–Uhlenbeck (OU) process, and their respective discrete-time counterparts, i.e., random walk and autoregressive (AR) process. These processes have been widely applied in a number of domains (Uhlenbeck & Ornstein, 1930; Levin et al., 2016; Arribas, 2018; Kidger et al., 2019; Lyons & McLeod, 2022).

To analyze the consistency of Lasso regressions, we first study the correlation structure of signatures for these processes. For Brownian motions, the correlation structure is shown to be block diagonal for Itô signatures (Propositions 1–2), and to have a special odd–even alternating structure for Stratonovich signatures (Propositions 3–4). In contrast, the OU process exhibits this odd–even alternating structure for either definition of the integral (Proposition 5).

Based on the correlation structures of signatures, we investigate the consistency of Lasso regressions for different processes (Propositions 6–8). For time series and processes that are closer to Brownian motion and with weaker inter-dimensional correlations, the Lasso regression is more consistent for their feature selection by Itô signatures; for mean reverting time series and processes, Stratonovich signatures yield more consistency for the Lasso regression.

**Contribution.** Our study takes the first step toward understanding the statistical properties of the signature transform for regression analysis. It fills one of the gaps between the theory and the practice of signature transforms in machine learning. Our work highlights the importance of choosing appropriate signature transforms and stochastic models for feature selection and for general statistical analysis.

## 2 THE FRAMEWORK

In this section, we present the framework for studying the consistency of feature selection in Lasso via signature. All proofs are given in Appendix K.

### 2.1 REVIEW OF SIGNATURES AND THEIR PROPERTIES

Consider a $d$-dimensional continuous-time stochastic process $\mathbf{X}_t = (X_t^1, X_t^2, \ldots, X_t^d)^\top \in \mathbb{R}^d$, $0 \leq t \leq T$, and its signature or signature transform defined as follows:

**Definition 1** (Signature). *For $k \geq 1$ and $i_1, \ldots, i_k \in \{1, 2, \ldots, d\}$, the $k$-th order signature of the process $\mathbf{X}$ with index $(i_1, \ldots, i_k)$ from time 0 to $t$ is defined as*

$$S(\mathbf{X})_t^{i_1, \ldots, i_k} = \int_{0 < t_1 < \cdots < t_k < t} \mathrm{d}X_{t_1}^{i_1} \cdots \mathrm{d}X_{t_k}^{i_k}, \quad 0 \leq t \leq T. \tag{1}$$

*The 0-th order signature of $\mathbf{X}$ from time 0 to $t$ is defined as $S(\mathbf{X})_t^0 = 1$ for any $0 \leq t \leq T$.*

In other words, the $k$-th order signature of $\mathbf{X}$ given by Equation (1) is its $k$-fold iterated path integral along the indices $i_1, \ldots, i_k$. For a given order $k$, there are $d^k$ choices of indices $(i_1, \ldots, i_k)$, therefore the number of all $k$-th order signatures is $d^k$.

The integral in Equation (1) can be specified differently. For example, if $\mathbf{X}$ is a deterministic process, it can be defined via the Riemann/Lebesgue integral. If $\mathbf{X}$ is a multi-dimensional Brownian motion, it is a stochastic integral that can be defined either by the Itô integral or by the Stratonovich integral. For clarity, we write

$$S(\mathbf{X})_t^{i_1,\ldots,i_k,I} = \int_{0<t_1<\cdots<t_k<t} \mathrm{d}X_{t_1}^{i_1} \cdots \mathrm{d}X_{t_k}^{i_k} = \int_{0<s<t} S(\mathbf{X})_s^{i_1,\ldots,i_{k-1},I} \mathrm{d}X_s^{i_k}$$

when considering the Itô integral, and

$$S(\mathbf{X})_t^{i_1,\ldots,i_k,S} = \int_{0<t_1<\cdots<t_k<t} \mathrm{d}X_{t_1}^{i_1} \circ \cdots \circ \mathrm{d}X_{t_k}^{i_k} = \int_{0<s<t} S(\mathbf{X})_s^{i_1,\ldots,i_{k-1},S} \circ \mathrm{d}X_s^{i_k}$$

for the Stratonovich integral.

Throughout the paper, for ease of exposition, we refer to the signature of $\mathbf{X}$ as the Itô (the Stratonovich) signature if the integral is defined in the sense of the Itô (the Stratonovich) integral.

Signatures enjoy several nice probabilistic properties under mild conditions. First, all expected signatures of a stochastic process together characterize the distribution of the process (Chevyrev & Lyons, 2016; Chevyrev & Oberhauser, 2022). Second, the signatures uniquely determine the path of the underlying process (Hambly & Lyons, 2010; Le Jan & Qian, 2013; Boedihardjo et al., 2014).

One of the most striking properties of the signature transform is its universal nonlinearity (Levin et al., 2016; Király & Oberhauser, 2019; Fermanian, 2021; Lemercier et al., 2021; Lyons & McLeod, 2022). It is of particular relevance for feature selection in machine learning or statistical analysis, where one needs to find or learn a (nonlinear) function $f$ that maps time series data $\mathbf{X}$ to a target label $y$. By universal nonlinearity, any such function can be approximately linearized by the signature of $\mathbf{X}$ in the following sense: for any $\varepsilon > 0$, under some technical conditions, there exists $K \geq 1$ and a *linear* function $L$ such that

$$\|f(\mathbf{X}) - L(\mathrm{Sig}_T^K(\mathbf{X}))\| \leq \varepsilon, \tag{2}$$

where $\mathrm{Sig}_T^K(\mathbf{X})$ represents all signatures of $\mathbf{X}$ from time 0 to $T$ truncated to some order $K$. This universal nonlinearity lays the foundation for learning the relationship between the time series $\mathbf{X}$ and a target label $y$ using a linear regression. Appendix B summarizes different statements of the universal nonlinearity of signatures in the literature.[2]

In the next section, we study feature selection via Lasso regression by signature transform.

## 2.2 Feature selection using Lasso with signatures

Suppose that one is given $N$ pairs of samples, $(\mathbf{X}_1, y_1), (\mathbf{X}_2, y_2), \ldots, (\mathbf{X}_N, y_N)$, where $\mathbf{X}_n = \{\mathbf{X}_{n,t}\}_{0 \leq t \leq T}$ is the $n$-th time series, for $n = 1, 2, \ldots, N$. Given a fixed order $K \geq 1$, consider the following regression model:

$$y_n = \beta_0 + \sum_{i_1=1}^{d} \beta_{i_1} S(\mathbf{X}_n)_T^{i_1} + \sum_{i_1,i_2=1}^{d} \beta_{i_1,i_2} S(\mathbf{X}_n)_T^{i_1,i_2} + \cdots + \sum_{i_1,\ldots,i_K=1}^{d} \beta_{i_1,\ldots,i_K} S(\mathbf{X}_n)_T^{i_1,\ldots,i_K} + \varepsilon_n, \tag{3}$$

where $n = 1, 2, \ldots, N$ represents $N$ samples, and $\{\varepsilon_n\}_{n=1}^N$ are independent and identically distributed errors with mean zero and finite variance. Here the number of predictors, i.e., the signature of various orders, is $\frac{d^{K+1}-1}{d-1}$, including the 0-th order signature $S(\mathbf{X})_T^0 = 1$, whose coefficient is $\beta_0$. It has been documented that including signatures up to a small order $K$ as predictors in a linear regression usually suffices to achieve good performances in practice (Morrill et al., 2020a; Lyons & McLeod, 2022).

The goal of Lasso is to identify the true predictors/features among all the predictors included in linear regression (3). A predictor has a zero beta coefficient if it is not in the true model. The

---

[2]The time augmentation is discussed in Section 5 and Appendix H.

selection of predictors with nonzero beta coefficients is a typical feature selection problem. We use $A_k^*$ to represent the set of all signatures of order $k$ with nonzero coefficients in Equation (3). Given any (nonlinear) function $f$ that one needs to learn, let us define the set of true predictors[3] $A^*$ by

$$A^* = \bigcup_{k=0}^{K} A_k^* := \bigcup_{k=0}^{K} \{(i_1, \dots, i_k) : \beta_{i_1, \dots, i_k} \neq 0\}. \tag{4}$$

Here, we begin the union with $k = 0$ to include the 0-th order signature for notational convenience.

Given a tuning parameter $\lambda > 0$ and $N$ samples, we adopt the following Lasso estimator to identify the true predictors:

$$\hat{\boldsymbol{\beta}}^N(\lambda) = \arg\min_{\tilde{\boldsymbol{\beta}}} \left[ \sum_{n=1}^{N} \left( y_n - \tilde{\beta}_0 - \sum_{i_1=1}^{d} \tilde{\beta}_{i_1} \tilde{S}(\mathbf{X}_n)_T^{i_1} - \sum_{i_1, i_2=1}^{d} \tilde{\beta}_{i_1, i_2} \tilde{S}(\mathbf{X}_n)_T^{i_1, i_2} - \cdots \right. \right.$$
$$\left. \left. - \sum_{i_1, \dots, i_K=1}^{d} \tilde{\beta}_{i_1, \dots, i_K} \tilde{S}(\mathbf{X}_n)_T^{i_1, \dots, i_K} \right)^2 + \lambda \left\| \tilde{\boldsymbol{\beta}} \right\|_1 \right], \tag{5}$$

where $\tilde{\boldsymbol{\beta}}$ is the vector containing all coefficients $\tilde{\beta}_{i_1, \dots, i_k}$, and $\|\cdot\|_1$ denotes the $l_1$-norm. Here, $\tilde{S}(\mathbf{X}_n)$ represents the standarized version of $S(\mathbf{X}_n)$ across $N$ samples by the $l_2$-norm, i.e., for any index $(i_1, \dots, i_k)$,

$$\tilde{S}(\mathbf{X}_n)_T^{i_1, \dots, i_k} = \frac{S(\mathbf{X}_n)_T^{i_1, \dots, i_k}}{\sqrt{\sum_{m=1}^{N} \left[ S(\mathbf{X}_m)_T^{i_1, \dots, i_k} \right]^2}}, \quad n = 1, 2, \dots, N.$$

We perform this standardization for two reasons. First, the Lasso estimator is sensitive to the magnitudes of the predictors, and standardization helps prevent the domination of predictors with larger magnitudes in the estimation process (Hastie et al., 2009). Second, the magnitudes of the signatures vary as the order of the signature changes (Lyons et al., 2007), therefore standardization is necessary to ensure that the coefficients of different orders of signatures are on the same scale and can be compared directly. Furthermore, the covariance matrix is now equivalent to the correlation matrix, allowing us to focus on the correlation structure of the signatures in the subsequent analysis.

## 2.3 CONSISTENCY AND THE IRREPRESENTABLE CONDITION OF LASSO

Our goal is to study the consistency of feature selection via signatures using the Lasso estimator in Equation (5). We use the concept of *(strong) sign consistency*, a custom definition of consistency for Lasso proposed in Zhao & Yu (2006).

**Definition 2** (Consistency)**.** *Lasso is (strongly) sign consistent if there exists $\lambda_N$, a function of sample number $N$, such that*

$$\lim_{N \to +\infty} \mathbb{P}\left( \text{sign}\left( \hat{\boldsymbol{\beta}}^N(\lambda_N) \right) = \text{sign}(\boldsymbol{\beta}) \right) = 1,$$

*where $\hat{\boldsymbol{\beta}}^N(\cdot)$ is the Lasso estimator given by Equation (5), $\boldsymbol{\beta}$ is a vector containing all beta coefficients of the true model, Equation (3), and the function $\text{sign}(\cdot)$ maps positive entries to 1, negative entries to $-1$, and 0 to 0.*

In other words, sign consistency requires that a pre-selected $\lambda$ can be used to achieve consistent feature selection via Lasso.

The following irrepresentable condition is almost a necessary and sufficient condition for the Lasso to be sign consistent (Zhao & Yu, 2006).

**Definition 3** (Irrepresentable condition)**.** *The feature selection in Equation (3) satisfies the (strong) irrepresentable condition if there exists a positive constant vector $\boldsymbol{\eta}$ such that*

$$\left| \Sigma_{A^{*c}, A^*} \Sigma_{A^*, A^*}^{-1} \text{sign}(\boldsymbol{\beta}_{A^*}) \right| \leq \mathbf{1} - \boldsymbol{\eta},$$

---

[3]True predictors are predictors with nonzero coefficients in Equation (3).

*where $A^*$ is given by Equation (4) and $A^{*c}$ the complement of $A^*$, $\Sigma_{A^{*c},A^*}$ ($\Sigma_{A^*,A^*}$) represents the covariance matrix[4] between all predictors in $A^{*c}$ and $A^*$ ($A^*$ and $A^*$), $\boldsymbol{\beta}_{A^*}$ represents a vector formed by beta coefficients for all predictors in $A^*$, $\mathbf{1}$ is an all-one vector, $|\cdot|$ calculates the absolute values of all entries, and the inequality "$\leq$" holds element-wise.*

This irrepresentable condition uses the *population* covariance matrix instead of the *sample* covariance matrix in Zhao & Yu (2006). Nevertheless, similar to Zhao & Yu (2006), it means that the irrelevant predictors in $A^{*c}$ cannot be sufficiently represented by the true predictors in $A^*$, implying weak collinearity between the predictors. Appendix C provides more details about the irrepresentable condition and its relationship with the consistency of Lasso.

By the signature transform, predictors in our linear regression (3) are correlated and have special correlation structures that differ from earlier studies (Zhao & Yu, 2006; Bickel et al., 2009; Wainwright, 2009). We will show in the following section that their correlation structures vary with the underlying process $\mathbf{X}$, hence leading to different consistency performances for different processes. Moreover, these correlation structures depend on the choice of integrals used in Equation (1).

## 3 CORRELATION STRUCTURE OF SIGNATURES

To study the consistency of Lasso using signatures, let us investigate the correlation structure of Itô and Stratonovich signatures for two representative Gaussian processes with different characteristics: the Brownian motion and the OU process.

### 3.1 CORRELATION STRUCTURE FOR MULTI-DIMENSIONAL BROWNIAN MOTION

**Definition 4** (Brownian motion). *$\mathbf{X}$ is a $d$-dimensional Brownian motion if it can be expressed as:*

$$\mathbf{X}_t = (X_t^1, X_t^2, \ldots, X_t^d)^\top = \Gamma(W_t^1, W_t^2, \ldots, W_t^d)^\top, \tag{6}$$

*where $W_t^1, W_t^2, \ldots, W_t^d$ are mutually independent standard Brownian motions, and $\Gamma$ is a matrix independent of $t$. In particular, $\mathrm{d}X_t^i \mathrm{d}X_t^j = \rho_{ij}\sigma_i\sigma_j \mathrm{d}t$ with $\rho_{ij}\sigma_i\sigma_j = (\Gamma\Gamma^\top)_{ij}$, where $\sigma_i$ is the volatility of $X_t^i$, and $\rho_{ij} \in [-1,1]$ is the inter-dimensional correlation between $X_t^i$ and $X_t^j$.*

Now we study the correlation structure of Itô and Stratonovich signatures respectively.

#### 3.1.1 ITÔ SIGNATURES FOR BROWNIAN MOTION

The following proposition gives the moments of Itô signatures of a $d$-dimensional Brownian motion.

**Proposition 1.** *Let $\mathbf{X}$ be a $d$-dimensional Brownian motion given by Equation (6). For $m, n = 1, 2, \ldots$ and $m \neq n$, we have:*

$$\mathbb{E}\left[S(\mathbf{X})_t^{i_1,\ldots,i_n,I}\right] = 0, \quad \mathbb{E}\left[S(\mathbf{X})_t^{i_1,\ldots,i_n,I} S(\mathbf{X})_t^{j_1,\ldots,j_m,I}\right] = 0,$$

$$\mathbb{E}\left[S(\mathbf{X})_t^{i_1,\ldots,i_n,I} S(\mathbf{X})_t^{j_1,\ldots,j_n,I}\right] = \frac{t^n}{n!}\prod_{k=1}^n \rho_{i_k j_k}\sigma_{i_k}\sigma_{j_k}.$$

With Proposition 1, the following result explicitly characterizes the correlation structure of Itô signatures for Brownian motions.

**Proposition 2.** *Let $\mathbf{X}$ be a $d$-dimensional Brownian motion given by Equation (6). If we arrange the signatures in recursive order (see Definition A.5 in Appendix D), the correlation matrix for Itô signatures of $\mathbf{X}$ with orders truncated to $K$ is a block diagonal matrix:*

$$\Sigma^I = \mathrm{diag}\{\Omega_0, \Omega_1, \Omega_2, \ldots, \Omega_K\}, \tag{7}$$

---

[4]In this paper, in line with Zhao & Yu (2006), all covariances and correlation coefficients are defined to be uncentered. Specifically, for random variables $X$ and $Y$, we define their covariance as $\mathbb{E}[XY]$, and their correlation coefficient as $\mathbb{E}[XY]/\sqrt{\mathbb{E}[X^2]\mathbb{E}[Y^2]}$. One can easily extend our results to the centered case.

*whose diagonal block $\Omega_k$ represents the correlation matrix for all $k$-th order signatures, which is given by:*

$$\Omega_k = \underbrace{\Omega \otimes \Omega \otimes \cdots \otimes \Omega}_{k}, \quad k = 1, 2, \ldots, K, \tag{8}$$

*and $\Omega_0 = 1$, where $\otimes$ represents Kronecker product, and $\Omega$ is a $d \times d$ matrix whose $(i, j)$-th entry is $\rho_{ij}$.*

Proposition 2 reveals several important facts about Itô signatures for Brownian motions. First, signatures of different orders are mutually independent, leading to a block diagonal correlation structure. Second, the correlation between signatures of the same order has a Kronecker product structure determined by the inter-correlation ($\rho_{ij}$) between different dimensions of $\mathbf{X}$.

### 3.1.2 STRATONOVICH SIGNATURES FOR BROWNIAN MOTION

The moments and correlation structure for Stratonovich signatures of Brownian motions are more complicated. We first provide the moments of Stratonovich signatures.

**Proposition 3.** *Let $\mathbf{X}$ be a $d$-dimensional Brownian motion given by Equation (6). For $m, n = 1, 2, \ldots$, we have*

$$\mathbb{E}\left[S(\mathbf{X})_t^{i_1, \ldots, i_{2n-1}, S}\right] = 0, \quad \mathbb{E}\left[S(\mathbf{X})_t^{i_1, \ldots, i_{2n}, S}\right] = \frac{1}{2^n} \frac{t^n}{n!} \prod_{k=1}^{n} \rho_{i_{2k-1} i_{2k}} \prod_{k=1}^{2n} \sigma_{i_k},$$

$$\mathbb{E}\left[S(\mathbf{X})_t^{i_1, \ldots, i_{2n}, S} S(\mathbf{X})_t^{j_1, \ldots, j_{2m-1}, S}\right] = 0,$$

*and $\mathbb{E}\left[S(\mathbf{X})_t^{i_1, \ldots, i_{2n}, S} S(\mathbf{X})_t^{j_1, \ldots, j_{2m}, S}\right]$ and $\mathbb{E}\left[S(\mathbf{X})_t^{i_1, \ldots, i_{2n-1}, S} S(\mathbf{X})_t^{j_1, \ldots, j_{2m-1}, S}\right]$ can be calculated using formulas provided in Proposition A.1 in Appendix D.*

The following result explicitly characterizes the correlation structure of Stratonovich signatures for Brownian motions.

**Proposition 4.** *Let $\mathbf{X}$ be a $d$-dimensional Brownian motion given by Equation (6). The correlation matrix for all Stratonovich signatures of $\mathbf{X}$ with orders truncated to $2K$ has the following odd–even alternating structure:*

$$\Sigma^S = \begin{pmatrix} \Psi_{0,0} & 0 & \Psi_{0,2} & 0 & \cdots & 0 & \Psi_{0,2K} \\ 0 & \Psi_{1,1} & 0 & \Psi_{1,3} & \cdots & \Psi_{1,2K-1} & 0 \\ \Psi_{2,0} & 0 & \Psi_{2,2} & 0 & \cdots & 0 & \Psi_{2,2K} \\ 0 & \Psi_{3,1} & 0 & \Psi_{3,3} & \cdots & \Psi_{3,2K-1} & 0 \\ \vdots & \vdots & \vdots & \vdots & \ddots & \vdots & \vdots \\ 0 & \Psi_{2K-1,1} & 0 & \Psi_{2K-1,3} & \cdots & \Psi_{2K-1,2K-1} & 0 \\ \Psi_{2K,0} & 0 & \Psi_{2K,2} & 0 & \cdots & 0 & \Psi_{2K,2K} \end{pmatrix}, \tag{9}$$

*where $\Psi_{m,n}$ is the correlation matrix between all $m$-th and $n$-th order signatures, which can be calculated using Proposition 3. In particular, if we re-arrange the indices of the signatures by putting all odd-order signatures and all even-order signatures together respectively, the correlation matrix has the following block diagonal form:*

$$\tilde{\Sigma}^S = \mathrm{diag}\{\Psi_{\mathrm{odd}}, \Psi_{\mathrm{even}}\},$$

*where $\Psi_{\mathrm{odd}}$ and $\Psi_{\mathrm{even}}$ are given respectively by*

$$\begin{pmatrix} \Psi_{1,1} & \Psi_{1,3} & \cdots & \Psi_{1,2K-1} \\ \Psi_{3,1} & \Psi_{3,3} & \cdots & \Psi_{3,2K-1} \\ \vdots & \vdots & \cdots & \vdots \\ \Psi_{2K-1,1} & \Psi_{2K-1,3} & \cdots & \Psi_{2K-1,2K-1} \end{pmatrix} \quad and \quad \begin{pmatrix} \Psi_{0,0} & \Psi_{0,2} & \cdots & \Psi_{0,2K} \\ \Psi_{2,0} & \Psi_{2,2} & \cdots & \Psi_{2,2K} \\ \vdots & \vdots & \cdots & \vdots \\ \Psi_{2K,0} & \Psi_{2K,2} & \cdots & \Psi_{2K,2K} \end{pmatrix}. \tag{10}$$

Propositions 2 and 4 reveal a striking difference between Itô and Stratonovich signatures for Brownian motions. Specifically, Itô signatures of different orders are uncorrelated, leading to a block diagonal correlation structure; Stratonovich signatures, in contrast, are uncorrelated only if they have different parity, leading to an odd–even alternating structure. This difference has significant implications for the consistency of the two types of signatures, which will be discussed in Section 4.

## 3.2 CORRELATION STRUCTURE FOR MULTI-DIMENSIONAL OU PROCESS

**Definition 5** (OU process). $\mathbf{X}$ *is a $d$-dimensional Ornstein–Uhlenbeck (OU) process if it can be expressed as:*

$$\mathbf{X}_t = (X_t^1, X_t^2, \ldots, X_t^d)^\top = \Gamma(Y_t^1, Y_t^2, \ldots, Y_t^d)^\top, \tag{11}$$

*where $\Gamma$ is a matrix independent of $t$, and $Y_t^1, Y_t^2, \ldots, Y_t^d$ are mutually independent OU processes driven by the following stochastic differential equations:*

$$\mathrm{d}Y_t^i = -\kappa_i Y_t^i \mathrm{d}t + \mathrm{d}W_t^i, \quad Y_0^i = 0,$$

*for $i = 1, 2, \ldots, d$. Here $\kappa_i > 0$, and $W_t^i$ are independent standard Brownian motions.*

The parameter $\kappa_i$ of the OU process controls the speed of mean reversion. A higher $\kappa_i$ implies a stronger mean reversion. When $\kappa_i = 0$, $Y_t^i$ reduces to a standard Brownian motion.

The following proposition shows that the odd–even alternating structure we observe in Proposition 4 holds for both Itô and Stratonovich signatures of the OU process.

**Proposition 5.** *Let $\mathbf{X}$ be a $d$-dimensional OU process given by Equation* (11). *The correlation matrix for all Itô signatures and the correlation matrix for all Stratonovich signatures of $\mathbf{X}$, with orders truncated to $2K$, both have the odd–even alternating structure given by Equation* (9).

Proposition 5 can be regarded as a generalization of the correlation structures for Itô and Stratonovich signatures of the Brownian motion in Propositions 2 and 4. In particular, for Itô signatures of the Brownian motion, all off-diagonal blocks in the odd–even alternating structure reduce to zero, as we observe in Proposition 2. However, the calculation of moments for the OU process is much more complicated than that for the Brownian motion, which we discuss in Appendix D.

## 4  CONSISTENCY OF SIGNATURES USING LASSO

This section investigates the consistency of feature selection in Lasso using signatures for both classes of Gaussian processes: the Brownian motion and the OU process. We also provide results for their discrete-time counterparts: the random walk and the AR process, respectively.

### 4.1  CONSISTENCY OF SIGNATURES FOR BROWNIAN MOTION AND RANDOM WALK

The following propositions characterize when the irrepresentable condition holds for signatures of Brownian motion.

**Proposition 6.** *For a multi-dimensional Brownian motion given by Equation* (6)*, the irrepresentable condition holds if and only if it holds for each block in the block-diagonal correlation matrix. In particular, for Itô signatures this is true when the irrepresentable condition holds for each $\Omega_k$ in Equation* (8)*; for Stratonovich signatures this is true when the irrepresentable condition holds for both $\Psi_{\mathrm{odd}}$ and $\Psi_{\mathrm{even}}$ in Equation* (10).

**Proposition 7.** *For a multi-dimensional Brownian motion given by Equation* (6)*, the irrepresentable condition holds for the correlation matrix of Itô signatures given by Equation* (7) *if*

$$|\rho_{ij}| < \frac{1}{2 \max_{0 \le k \le K}\{\#A_k^*\} - 1}, \tag{12}$$

*where $A_k^*$ is defined in Equation* (4).

Proposition 6 demonstrates both the similarity and difference between Itô signatures and Stratonovich signatures for Brownian motions. In particular, the difference in the block structure of their correlation matrices leads to the difference in the consistency of their feature selection.

Proposition 7 provides a sufficient condition for Itô signatures that can be easily used in practice: the Lasso is consistent when different dimensions of the multi-dimensional Brownian motion are not strongly correlated, with a sufficient bound by Equation (12). Appendix E discusses the tightness of this bound.

Empirically, it has been documented that a small $K$ suffices to provide a reasonable approximation in applications (Morrill et al., 2020a; Lyons & McLeod, 2022). Therefore, $\max_{0 \leq k \leq K}\{\#A_k^*\}$ is typically small, which implies that the bound given by Equation (12) is fairly easy to satisfy.

The consistency study for Stratonovich signatures reveals a different picture: the irrepresentable condition may fail even when all dimensions of $\mathbf{X}$ are mutually independent. This is shown in Example A.4 in Appendix D. It suggests that the statistical properties of Lasso may be worse for Stratonovich signatures.

Simulations further confirm this implication.[5] Consider a two-dimensional ($d = 2$) Brownian motion with inter-dimensional correlation $\rho$; assume that there are $q = \#A^*$ true predictors in the true model (3), and all of these predictors are signatures of orders no greater than $K = 4$. Now, first randomly choose $q$ true predictors from all $\frac{d^{K+1}-1}{d-1} = 31$ signatures; next randomly set each beta coefficient of these true predictors from the standard normal distribution; next generate 100 samples from this true model with error term $\varepsilon_n$ drawn from a normal distribution with mean zero and standard error 0.01; then run a Lasso regression given by Equation (5) to select predictors based on these 100 samples; and finally check whether the Lasso is sign consistent according to Definition 2. Repeat the above procedure by 1,000 times and calculate the *consistency rate*, which is defined as the proportion of consistent results among these 1,000 experiments.[6]

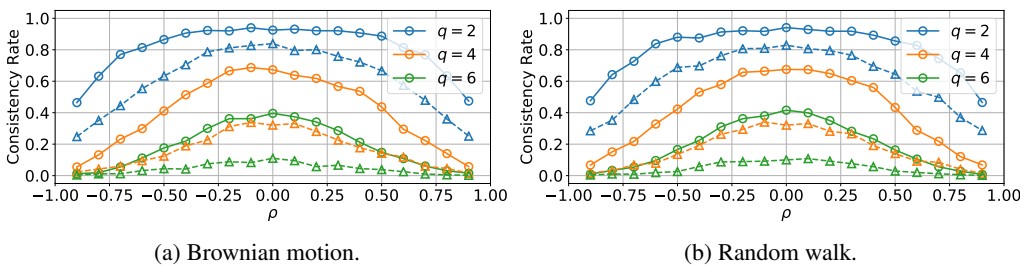

(a) Brownian motion.  (b) Random walk.

Figure 1: Consistency rates for the Brownian motion and the random walk with different values of inter-dimensional correlation, $\rho$, and different numbers of true predictors, $q$. Solid lines correspond to Itô signatures and dashed lines correspond to Stratonovich signatures.

Figure 1 shows the consistency rates for different values of inter-dimensional correlation, $\rho$, and different numbers of true predictors, $q$. Figure 1a shows the results for the Brownian motion, and Figure 1b for its discrete version—the random walk. First, signatures for both Brownian motion and random walk are similar: they both exhibit higher consistency rates when the absolute value of $\rho$ is small, i.e., when the inter-dimensional correlations of the Brownian motion (random walk) are weak; second, as the number of true predictors $q$ increases, both consistency rates decrease. These findings are consistent with our theoretical results.

Finally, consistency rates for Itô signatures are consistently higher than those for Stratonovich signatures, holding other variables constant ($\rho$ and $q$). This can be attributed to the difference between the definitions of Itô and Stratonovich integrals. Recall that, given a partition of $[0, T]$, Itô integrals use only the value of the integrand at the left endpoint of each subinterval, whereas Stratonovich integrals use the values at both the left and the right endpoints. The interaction between the two endpoints for Stratonovich integrals introduces more collinearity between Stratonovich signatures. This is also observed in Propositions 2 and 4—signatures of different orders are uncorrelated when using Itô signatures but become correlated when using Stratonovich signatures. The collinearity between Stratonovich signatures contributes to their lower consistency for Lasso.

## 4.2 Consistency of signatures for OU processes and AR processes

For both the Itô and the Stratonovich signatures of the OU process, we have the following necessary and sufficient condition for the irrepresentable condition. However, it appears difficult to derive the analogue of Proposition 7 for OU processes.

---

[5]Appendix F reports more details including its computational cost and robustness checks.

[6]Appendices I examines the impact of the number of dimension $d$ and the number of samples.

**Proposition 8.** *For a multi-dimensional OU process given by Equation* (11)*, the irrepresentable condition holds for the correlation matrix of signatures if and only if it holds for both* $\Psi_{\mathrm{odd}}$ *and* $\Psi_{\mathrm{even}}$ *given by Equation* (10)*. This result holds for both Itô and Stratonovich signatures.*

Now we study the impact of mean reversion on the consistency of Lasso, for both the OU process and its discrete version—the autoregressive AR(1) model with parameter $\phi$. Recall that higher values of $\kappa$ for the OU process and lower values of $\phi$ for the AR(1) model imply stronger levels of mean reversion. We consider two-dimensional OU processes and AR(1) processes, with both dimensions sharing the same parameters ($\kappa$ and $\phi$). The inter-dimensional correlation matrix $\Gamma\Gamma^{\top}$ is randomly drawn from the Wishart$(2, 2)$ distribution. Other simulation setups are the same as in Section 4.1.

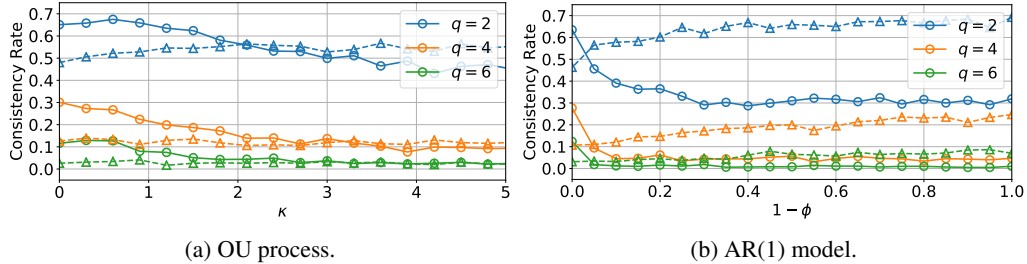

(a) OU process.

(b) AR(1) model.

Figure 2: Consistency rates for the OU process and the AR(1) model with different parameters ($\kappa$ and $1 - \phi$) and different numbers of true predictors, $q$. Solid lines correspond to Itô signatures and dashed lines correspond to Stratonovich signatures.

Figure 2 shows the simulation results for the consistency rates of both processes. First, the Itô signature reaches the highest consistency rate when $\kappa$ and $1 - \phi$ approach 0, which correspond respectively to a Brownian motion and a random walk. Second, when the process is sufficiently mean reverting, Stratonovich signatures have higher consistency rates than Itô signatures. Finally, as observed in Section 4.1, Lasso gets less consistent when the number of true predictors $q$ increases. These results suggest that, in practice, for processes that are sufficiently rough or mean reverting (Gatheral et al., 2018), using Lasso with Stratonovich signatures will likely lead to higher statistical consistency compared to Itô signatures. More theoretical explanations are provided in Appendix D. Appendix J examines the more complex ARIMA processes.

## 5  DISCUSSION

**Other performance metrics.**  We have adopted the sign consistency of Lasso (Zhao & Yu, 2006), defined as whether the Lasso can select *all* true predictors with correct signs. This restrictive notion of consistency may be relaxed in the context of signatures. Extensions of the sign consistency for signatures are given in Appendix G. Overall, a lower sign consistency implies poorer performances when using other metrics to measure the performance of feature selection using Lasso, such as the out-of-sample mean squared error.

**Time augmentation.**  Time augmentation is a widely used technique in signature-based analysis, which involves adding a time dimension $t$ to the original time series, $\mathbf{X}_t$ (Chevyrev & Kormilitzin, 2016). Time augmentation lowers the consistency rate of Lasso, as presented in Appendix H.

**Other feature selection techniques.**  While Lasso is a popular feature selection technique, there are also other commonly used techniques, such as the bridge regression (Frank & Friedman, 1993). The research on the consistency of signatures using other techniques is left for further investigation.

## 6  CONCLUSION

This paper studies the statistical consistency of Lasso regression for signatures. It finds that consistency is highly dependent on the definition of the signatures and the characteristics of the underlying processes. These findings call for further statistical studies for signature transform before its potential for machine learning can be fully realized.

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

## A    GEOMETRIC INTERPRETATION OF SIGNATURES

Signatures can be interpreted from a geometric perspective. Consider a $d$-dimensional piecewise linear process $\mathbf{X}_t$, Figure A.1 illustrates the geometric interpretation of its first two orders of signatures. (The Itô and Stratonovich signatures are the same for a piecewise linear process.) The green line represents the path of the $i$-th and the $j$-th dimensions of $\mathbf{X}_t$, $(X_t^i, X_t^j)$, from time 0 to $T$. By definition, the first order signatures are $S(\mathbf{X})_T^i = X_T^i - X_0^i$ and $S(\mathbf{X})_T^j = X_T^j - X_0^j$, which correspond to the increments of the path along the $i$-th and the $j$-th dimensions, respectively. Both increments can be visualized as the length and height of the rectangle in Figure A.1.

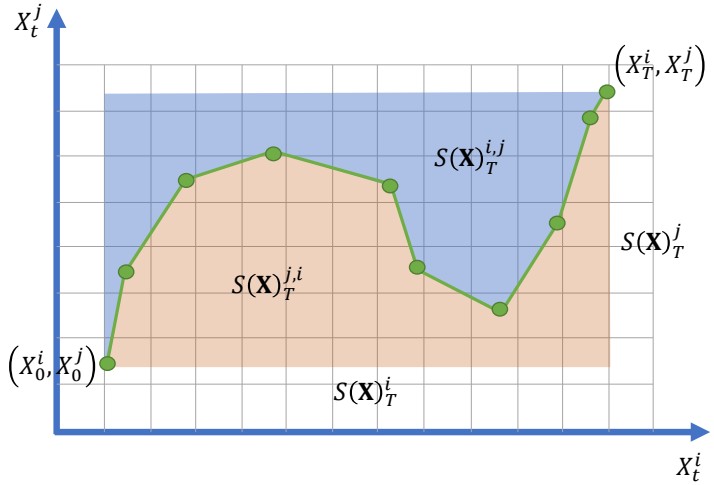

Figure A.1: The geometric interpretation of signatures.

Now we consider the second order signatures. By definition, we have $S(\mathbf{X})_T^{i,i} = (X_T^i - X_0^i)^2/2$ and $S(\mathbf{X})_T^{j,j} = (X_T^j - X_0^j)^2/2$, which are functions of the first order signatures, $S(\mathbf{X})_T^i$ and $S(\mathbf{X})_T^j$. In addition, path integral theory implies that $S(\mathbf{X})_T^{i,j}$ and $S(\mathbf{X})_T^{j,i}$ are the areas of the blue and orange regions, respectively (Chevyrev & Kormilitzin, 2016). Therefore, all the first two orders of signatures can be interpreted using Figure A.1.

When $\mathbf{X}_t$ is a continuous martingale, such as a Brownian motion or an OU process, the geometric interpretation mentioned above still applies to the Stratonovich signatures of $\mathbf{X}_t$. The Itô signatures can be interpreted as Stratonovich signatures adjusted using quadratic variations. This arises from the following relationship between Itô and Stratonovich integrals:

$$\int_0^t A_s \mathrm{d}B_s = \int_0^t A_s \circ \mathrm{d}B_s - \frac{1}{2}[A,B]_t,$$

where $[A,B]_t$ is the quadratic covariation between processes $A$ and $B$. For example, if $\mathbf{X}_t = (W_1, W_2, \ldots, W_d)$ is a $d$ dimensional standard Brownian motion with correlations $\rho_{ij}$ between dimensions $i$ and $j$, we have

$$S(\mathbf{X})_t^{i,I} = S(\mathbf{X})_t^{i,S},$$
$$S(\mathbf{X})_t^{i,j,I} = S(\mathbf{X})_t^{i,j,S} - \frac{1}{2}\rho_{ij}t,$$

for any $i,j = 1,2,\ldots,d$. Therefore, Itô signatures can be regarded as Stratonovich signatures adjusted by the quadratic variation of the underlying process.

Higher order signatures may capture additional geometric features of the underlying path (Chevyrev & Kormilitzin, 2016; Levin et al., 2016; Morrill et al., 2020a). This geometric interpretation of signatures has also led to successful applications in character recognition and quantitative finance (Levin et al., 2016).

## B    TECHNICAL DETAILS FOR UNIVERSAL NONLINEARITY OF SIGNATURES

There are different statements of universal nonlinearity of signatures in the literature, which we summarize in Table A.1.

Table A.1: Universal nonlinearity in the literature.

| Path | With time augmentation | Integral |
|---|---|---|
| Cuchiero et al. (2022) cadlag rough path | Yes | Rough |
| Cuchiero et al. (2023) Continuous semimartingale | Yes | Stratonovich |
| Arribas (2018); Lyons et al. (2020) Continuous rough path | Yes | Stratonovich |
| Levin et al. (2016) Continuous rough path | No | Stratonovich or Itô |
| Levin et al. (2016); Király & Oberhauser (2019); Fermanian (2021) Bounded variation path | No | Riemann/Lebesgue |

The following theorem gives the precise statement of universal nonlinearity proposed in Levin et al. (2016).

**Theorem 1** (Universal nonlinearity, Theorem 3.1 of Levin et al. (2016)). *Let $\mathbf{X}_t$ be a $\mathbb{R}^d$-valued continuous path with finite $p$-variation, and let $\mathcal{S}$ be a compact subset of signature paths of $\mathbf{X}_t$ from time 0 to T. Assume that $f : \mathcal{S} \to \mathbb{R}$ is a continuous function. Then, for any $\varepsilon > 0$, there exists a linear functional $L : \mathbb{R}^\infty \to \mathbb{R}$ such that for every $s \in \mathcal{S}$:*

$$|f(s) - L(s)| \leq \varepsilon.$$

The proof of this theorem can be found in Levin et al. (2016). Other versions of universal nonlinearity can be found in, for example, Arribas (2018); Király & Oberhauser (2019); Lyons et al. (2020); Fermanian (2021); Cuchiero et al. (2022; 2023).

## C    TECHNICAL DETAILS FOR THE IRREPRESENTABLE CONDITION AND SIGN CONSISTENCY

In our main article, we briefly introduce the definitions of the sign consistency and the irrepresentable condition for Lasso as proposed by Zhao & Yu (2006). This appendix provides more technical details on these definitions and their relationships.

We have introduced the *strong sign consistency* in our main paper; see Definition 2. To enhance readability, we present the definition again below.

**Definition A.1** (Strong Sign Consistency). *Lasso is strongly sign consistent if there exists $\lambda_N$, a function of sample number $N$, such that*

$$\lim_{N \to +\infty} \mathbb{P}\left(\text{sign}\left(\hat{\boldsymbol{\beta}}^N(\lambda_N)\right) = \text{sign}(\boldsymbol{\beta})\right) = 1,$$

*where $\hat{\boldsymbol{\beta}}^N(\cdot)$ is the Lasso estimator given by Equation (5), $\boldsymbol{\beta}$ is a vector containing all beta coefficients of the true model, Equation (3), and the function $\text{sign}(\cdot)$ maps positive entries to 1, negative entries to $-1$, and 0 to 0.*

There is another version of sign consistency of Lasso, *general sign consistency*, which is defined as follows.

**Definition A.2** (General Sign Consistency). *Lasso is general sign consistent if*

$$\lim_{N \to +\infty} \mathbb{P}\left(\exists \lambda \geq 0, \text{sign}\left(\hat{\boldsymbol{\beta}}^N(\lambda)\right) = \text{sign}(\boldsymbol{\beta})\right) = 1,$$

*where the notations are defined as Definition A.1.*

Strong sign consistency implies that using a preselected $\lambda_N$ can achieve consistent predictor selection via Lasso. General sign consistency means that there is an appropriate value of $\lambda$ that selects the true predictors. Strong sign consistency implies general sign consistency.

We also have introduced the strong irrepresentable condition in our main paper; see Definition 3. To enhance readability, we present the definition again below.

**Definition A.3** (Strong Irrepresentable Condition). *The feature selection in Equation* (3) *satisfies the strong irrepresentable condition if there exists a positive constant vector $\boldsymbol{\eta}$ such that*

$$\left| \hat{\Sigma}_{A^{*c},A^*}^N (\hat{\Sigma}_{A^*,A^*}^N)^{-1} \mathrm{sign}(\boldsymbol{\beta}_{A^*}) \right| \leq \mathbf{1} - \boldsymbol{\eta},$$

*where $A^*$ is given by Equation* (4) *and $A^{*c}$ the complement of $A^*$, $\hat{\Sigma}_{A^{*c},A^*}^N$ ($\hat{\Sigma}_{A^*,A^*}^N$) represents the sample covariance matrix between all predictors in $A^{*c}$ and $A^*$ ($A^*$ and $A^*$), $\boldsymbol{\beta}_{A^*}$ represents a vector formed by beta coefficients for all predictors in $A^*$, $\mathbf{1}$ is an all-one vector, $|\cdot|$ calculates the absolute values of all entries, and the inequality "$\leq$" holds element-wise.*

The weak version is defined as follows.

**Definition A.4** (Weak Irrepresentable Condition). *The feature selection in Equation* (3) *satisfies the weak irrepresentable condition if*

$$\left| \hat{\Sigma}_{A^{*c},A^*}^N (\hat{\Sigma}_{A^*,A^*}^N)^{-1} \mathrm{sign}(\boldsymbol{\beta}_{A^*}) \right| < \mathbf{1},$$

*where the inequality "$<$" holds element-wise, and other notations are defined as Definition A.3.*

As explained by Zhao & Yu (2006), the irrepresentable conditions resemble a regularization constraint on the regression coefficients of the false predictors on true predictors. In particular, when signs of the true beta coefficients are unknown, for the irrepresentable conditions to hold for all possible signs, we need the $L_1$ norms of the regression coefficients to be smaller than 1, i.e.,

$$\left| (\hat{\Sigma}_{A^*,A^*}^N)^{-1} \hat{\Sigma}_{A^*,A^{*c}}^N \right| < \mathbf{1}.$$

That is, the total amount of a false predictor represented by the true predictors is not to reach 1, which explains the name of "irrepresentable condition."

Zhao & Yu (2006) demonstrate that the irrepresentable condition is *almost* a necessary and sufficient condition for the consistency of Lasso. This "almost" equivalence is established by the following two theorems, which are Theorems 1 and 2 of Zhao & Yu (2006):

**Theorem 2** (Theorem 1 of Zhao & Yu (2006)). *The feature selection in Equation* (3) *is strongly sign consistent if the strong irrepresentable condition holds.*

**Theorem 3** (Theorem 2 of Zhao & Yu (2006)). *The feature selection in Equation* (3) *is general sign consistent only if there exists $n$ such that weak irrepresentable condition holds for $N > n$.*

Theorems 2 and 3 can be summarized as:

$$\text{strong irrepresentable condition} \Rightarrow \text{strong sign consistency}$$
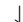
$$\text{weak irrepresentable condition} \Leftarrow \text{general sign consistency}$$

Therefore, Zhao & Yu (2006) documented that the irrepresentable condition is "almost necessary and sufficient" for sign consistency.

## D    TECHNICAL DETAILS AND EXAMPLES FOR THE CALCULATION OF CORRELATION MATRICES

This appendix provides details and examples for calculating the correlation structures of signatures. Appendices D.1 and D.2 discuss the Brownian motion and the OU process, respectively.

**Itô signature.** Propositions 1–2 in the main paper give explicit formulas for calculating the correlation structure of Itô signatures for Brownian motions. The "recursive order" mentioned in Proposition 2 is defined as follows.

**Definition A.5** (Recursive Order). *Consider a $d$-dimensional process* **X**. *We order the indices of all of its 1st order signatures as:*

$$1 \quad 2 \quad \cdots \quad d.$$

*Then, if all $k$-th order signatures are ordered as:*

$$r_1 \quad r_2 \quad \cdots \quad r_{d^k},$$

*we define the orders of all $(k+1)$-th order signatures as:*

$$r_1,1 \quad r_2,1 \quad \cdots \quad r_{d^k},1 \quad r_1,2 \quad r_2,2 \quad \cdots \quad r_{d^k},2 \quad \cdots\cdots\cdots \quad r_1,d \quad r_2,d \quad \cdots \quad r_{d^k},d.$$

*For example, for a $d = 3$ dimensional process, the recursive order of its signatures is:*

- 1st order: 1  2  3

- 2nd order: 1,1  2,1  3,1  1,2  2,2  3,2  1,3  2,3  3,3

- 3rd order: 1,1,1  2,1,1  3,1,1  1,2,1  2,2,1  3,2,1  1,3,1  2,3,1  3,3,1
  1,1,2  2,1,2  3,1,2  1,2,2  2,2,2  3,2,2  1,3,2  2,3,2  3,3,2
  1,1,3  2,1,3  3,1,3  1,2,3  2,2,3  3,2,3  1,3,3  2,3,3  3,3,3

- ...

To provide intuition for Propositions 1–2 in the main paper, the following two examples show the correlation structures of Itô signatures for 2-dimensional Brownian motions with inter-dimensional correlations $\rho = 0.6$ and $\rho = 0$, respectively.

**Example A.1.** *Consider a 2-dimensional Brownian motion given by Equation (6) with inter-dimensional correlation $\rho = 0.6$. Figure A.2 shows the correlation matrix of its Itô signatures with orders truncated to 4 calculated using Proposition 1. The figure illustrates Proposition 2—the correlation matrix has a block diagonal structure, and each block of the matrix is the Kronecker product of the inter-dimensional correlation matrix $\begin{pmatrix} 1 & 0.6 \\ 0.6 & 1 \end{pmatrix}$.*

| | 0th | 1st | | 2nd order | | | | 3rd order | | | | | | | | 4th order | | | | | | | | | | | | | | | |
|---|---|---|---|---|---|---|---|---|---|---|---|---|---|---|---|---|---|---|---|---|---|---|---|---|---|---|---|---|---|---|---|
| **0th** | 1 | 0 | 0 | 0 | 0 | 0 | 0 | 0 | 0 | 0 | 0 | 0 | 0 | 0 | 0 | 0 | 0 | 0 | 0 | 0 | 0 | 0 | 0 | 0 | 0 | 0 | 0 | 0 | 0 | 0 | 0 |
| **1st** | 0 | 1 | 0.6 | 0 | 0 | 0 | 0 | 0 | 0 | 0 | 0 | 0 | 0 | 0 | 0 | 0 | 0 | 0 | 0 | 0 | 0 | 0 | 0 | 0 | 0 | 0 | 0 | 0 | 0 | 0 | 0 |
| | 0 | 0.6 | 1 | 0 | 0 | 0 | 0 | 0 | 0 | 0 | 0 | 0 | 0 | 0 | 0 | 0 | 0 | 0 | 0 | 0 | 0 | 0 | 0 | 0 | 0 | 0 | 0 | 0 | 0 | 0 | 0 |
| **2nd** | 0 | 0 | 0 | 1 | 0.6 | 0.6 | 0.36 | 0 | 0 | 0 | 0 | 0 | 0 | 0 | 0 | 0 | 0 | 0 | 0 | 0 | 0 | 0 | 0 | 0 | 0 | 0 | 0 | 0 | 0 | 0 | 0 |
| | 0 | 0 | 0 | 0.6 | 1 | 0.36 | 0.6 | 0 | 0 | 0 | 0 | 0 | 0 | 0 | 0 | 0 | 0 | 0 | 0 | 0 | 0 | 0 | 0 | 0 | 0 | 0 | 0 | 0 | 0 | 0 | 0 |
| | 0 | 0 | 0 | 0.6 | 0.36 | 1 | 0.6 | 0 | 0 | 0 | 0 | 0 | 0 | 0 | 0 | 0 | 0 | 0 | 0 | 0 | 0 | 0 | 0 | 0 | 0 | 0 | 0 | 0 | 0 | 0 | 0 |
| | 0 | 0 | 0 | 0.36 | 0.6 | 0.6 | 1 | 0 | 0 | 0 | 0 | 0 | 0 | 0 | 0 | 0 | 0 | 0 | 0 | 0 | 0 | 0 | 0 | 0 | 0 | 0 | 0 | 0 | 0 | 0 | 0 |
| **3rd** | 0 | 0 | 0 | 0 | 0 | 0 | 0 | 1 | 0.6 | 0.6 | 0.36 | 0.6 | 0.36 | 0.36 | 0.22 | 0 | 0 | 0 | 0 | 0 | 0 | 0 | 0 | 0 | 0 | 0 | 0 | 0 | 0 | 0 | 0 |
| | 0 | 0 | 0 | 0 | 0 | 0 | 0 | 0.6 | 1 | 0.36 | 0.6 | 0.36 | 0.6 | 0.22 | 0.36 | 0 | 0 | 0 | 0 | 0 | 0 | 0 | 0 | 0 | 0 | 0 | 0 | 0 | 0 | 0 | 0 |
| | 0 | 0 | 0 | 0 | 0 | 0 | 0 | 0.6 | 0.36 | 1 | 0.6 | 0.36 | 0.22 | 0.6 | 0.36 | 0 | 0 | 0 | 0 | 0 | 0 | 0 | 0 | 0 | 0 | 0 | 0 | 0 | 0 | 0 | 0 |
| | 0 | 0 | 0 | 0 | 0 | 0 | 0 | 0.36 | 0.6 | 0.6 | 1 | 0.22 | 0.36 | 0.36 | 0.6 | 0 | 0 | 0 | 0 | 0 | 0 | 0 | 0 | 0 | 0 | 0 | 0 | 0 | 0 | 0 | 0 |
| | 0 | 0 | 0 | 0 | 0 | 0 | 0 | 0.6 | 0.36 | 0.36 | 0.22 | 1 | 0.6 | 0.6 | 0.36 | 0 | 0 | 0 | 0 | 0 | 0 | 0 | 0 | 0 | 0 | 0 | 0 | 0 | 0 | 0 | 0 |
| | 0 | 0 | 0 | 0 | 0 | 0 | 0 | 0.36 | 0.6 | 0.22 | 0.36 | 0.6 | 1 | 0.36 | 0.6 | 0 | 0 | 0 | 0 | 0 | 0 | 0 | 0 | 0 | 0 | 0 | 0 | 0 | 0 | 0 | 0 |
| | 0 | 0 | 0 | 0 | 0 | 0 | 0 | 0.36 | 0.22 | 0.6 | 0.36 | 0.6 | 0.36 | 1 | 0.6 | 0 | 0 | 0 | 0 | 0 | 0 | 0 | 0 | 0 | 0 | 0 | 0 | 0 | 0 | 0 | 0 |
| | 0 | 0 | 0 | 0 | 0 | 0 | 0 | 0.22 | 0.36 | 0.36 | 0.6 | 0.36 | 0.6 | 0.6 | 1 | 0 | 0 | 0 | 0 | 0 | 0 | 0 | 0 | 0 | 0 | 0 | 0 | 0 | 0 | 0 | 0 |
| **4th** | 0 | 0 | 0 | 0 | 0 | 0 | 0 | 0 | 0 | 0 | 0 | 0 | 0 | 0 | 0 | 1 | 0.6 | 0.6 | 0.36 | 0.6 | 0.36 | 0.36 | 0.22 | 0.6 | 0.36 | 0.36 | 0.22 | 0.36 | 0.22 | 0.22 | 0.13 |
| | 0 | 0 | 0 | 0 | 0 | 0 | 0 | 0 | 0 | 0 | 0 | 0 | 0 | 0 | 0 | 0.6 | 1 | 0.36 | 0.6 | 0.36 | 0.6 | 0.22 | 0.36 | 0.36 | 0.6 | 0.22 | 0.36 | 0.22 | 0.36 | 0.13 | 0.22 |
| | 0 | 0 | 0 | 0 | 0 | 0 | 0 | 0 | 0 | 0 | 0 | 0 | 0 | 0 | 0 | 0.6 | 0.36 | 1 | 0.6 | 0.36 | 0.22 | 0.6 | 0.36 | 0.36 | 0.22 | 0.6 | 0.36 | 0.22 | 0.13 | 0.36 | 0.22 |
| | 0 | 0 | 0 | 0 | 0 | 0 | 0 | 0 | 0 | 0 | 0 | 0 | 0 | 0 | 0 | 0.36 | 0.6 | 0.6 | 1 | 0.22 | 0.36 | 0.36 | 0.6 | 0.22 | 0.36 | 0.36 | 0.6 | 0.13 | 0.22 | 0.22 | 0.36 |
| | 0 | 0 | 0 | 0 | 0 | 0 | 0 | 0 | 0 | 0 | 0 | 0 | 0 | 0 | 0 | 0.6 | 0.36 | 0.36 | 0.22 | 1 | 0.6 | 0.6 | 0.36 | 0.36 | 0.22 | 0.22 | 0.13 | 0.6 | 0.36 | 0.36 | 0.22 |
| | 0 | 0 | 0 | 0 | 0 | 0 | 0 | 0 | 0 | 0 | 0 | 0 | 0 | 0 | 0 | 0.36 | 0.6 | 0.22 | 0.36 | 0.6 | 1 | 0.36 | 0.6 | 0.22 | 0.36 | 0.13 | 0.22 | 0.36 | 0.6 | 0.22 | 0.36 |
| | 0 | 0 | 0 | 0 | 0 | 0 | 0 | 0 | 0 | 0 | 0 | 0 | 0 | 0 | 0 | 0.36 | 0.22 | 0.6 | 0.36 | 0.6 | 0.36 | 1 | 0.6 | 0.22 | 0.13 | 0.36 | 0.22 | 0.36 | 0.22 | 0.6 | 0.36 |
| | 0 | 0 | 0 | 0 | 0 | 0 | 0 | 0 | 0 | 0 | 0 | 0 | 0 | 0 | 0 | 0.22 | 0.36 | 0.36 | 0.6 | 0.36 | 0.6 | 0.6 | 1 | 0.13 | 0.22 | 0.22 | 0.36 | 0.22 | 0.36 | 0.36 | 0.6 |
| | 0 | 0 | 0 | 0 | 0 | 0 | 0 | 0 | 0 | 0 | 0 | 0 | 0 | 0 | 0 | 0.6 | 0.36 | 0.36 | 0.22 | 0.36 | 0.22 | 0.22 | 0.13 | 1 | 0.6 | 0.6 | 0.36 | 0.6 | 0.36 | 0.36 | 0.22 |
| | 0 | 0 | 0 | 0 | 0 | 0 | 0 | 0 | 0 | 0 | 0 | 0 | 0 | 0 | 0 | 0.36 | 0.6 | 0.22 | 0.36 | 0.22 | 0.36 | 0.13 | 0.22 | 0.6 | 1 | 0.36 | 0.6 | 0.36 | 0.6 | 0.22 | 0.36 |
| | 0 | 0 | 0 | 0 | 0 | 0 | 0 | 0 | 0 | 0 | 0 | 0 | 0 | 0 | 0 | 0.36 | 0.22 | 0.6 | 0.36 | 0.22 | 0.13 | 0.36 | 0.22 | 0.6 | 0.36 | 1 | 0.6 | 0.36 | 0.22 | 0.6 | 0.36 |
| | 0 | 0 | 0 | 0 | 0 | 0 | 0 | 0 | 0 | 0 | 0 | 0 | 0 | 0 | 0 | 0.22 | 0.36 | 0.36 | 0.6 | 0.13 | 0.22 | 0.22 | 0.36 | 0.36 | 0.6 | 0.6 | 1 | 0.22 | 0.36 | 0.36 | 0.6 |
| | 0 | 0 | 0 | 0 | 0 | 0 | 0 | 0 | 0 | 0 | 0 | 0 | 0 | 0 | 0 | 0.36 | 0.22 | 0.22 | 0.13 | 0.6 | 0.36 | 0.36 | 0.22 | 0.6 | 0.36 | 0.36 | 0.22 | 1 | 0.6 | 0.6 | 0.36 |
| | 0 | 0 | 0 | 0 | 0 | 0 | 0 | 0 | 0 | 0 | 0 | 0 | 0 | 0 | 0 | 0.22 | 0.36 | 0.13 | 0.22 | 0.36 | 0.6 | 0.22 | 0.36 | 0.36 | 0.6 | 0.22 | 0.36 | 0.6 | 1 | 0.36 | 0.6 |
| | 0 | 0 | 0 | 0 | 0 | 0 | 0 | 0 | 0 | 0 | 0 | 0 | 0 | 0 | 0 | 0.22 | 0.13 | 0.36 | 0.22 | 0.36 | 0.22 | 0.6 | 0.36 | 0.36 | 0.22 | 0.6 | 0.36 | 0.6 | 0.36 | 1 | 0.6 |
| | 0 | 0 | 0 | 0 | 0 | 0 | 0 | 0 | 0 | 0 | 0 | 0 | 0 | 0 | 0 | 0.13 | 0.22 | 0.22 | 0.36 | 0.22 | 0.36 | 0.36 | 0.6 | 0.22 | 0.36 | 0.36 | 0.6 | 0.36 | 0.6 | 0.6 | 1 |

Figure A.2: Correlation matrix of Itô signatures with orders truncated to 4 for a 2-dimensional Brownian motion with inter-dimensional correlation $\rho = 0.6$.

**Example A.2.** *Consider a 2-dimensional Brownian motion given by Equation* (6) *with inter-dimensional correlation* $\rho = 0$. *Figure A.3 shows the correlation matrix of its Itô signatures with orders truncated to 4 calculated using Proposition 1. When* $\rho = 0$, *the block diagonal correlation matrix reduces to an identity matrix, indicating that all of its Itô signatures are mutually uncorrelated.*

| | 0th | 1st | | 2nd order | | | | 3rd order | | | | | | | | 4th order | | | | | | | | | | | | | | | |
|---|---|---|---|---|---|---|---|---|---|---|---|---|---|---|---|---|---|---|---|---|---|---|---|---|---|---|---|---|---|---|---|
| **0th** | 1 | 0 | 0 | 0 | 0 | 0 | 0 | 0 | 0 | 0 | 0 | 0 | 0 | 0 | 0 | 0 | 0 | 0 | 0 | 0 | 0 | 0 | 0 | 0 | 0 | 0 | 0 | 0 | 0 | 0 | 0 |
| **1st** | 0 | 1 | 0 | 0 | 0 | 0 | 0 | 0 | 0 | 0 | 0 | 0 | 0 | 0 | 0 | 0 | 0 | 0 | 0 | 0 | 0 | 0 | 0 | 0 | 0 | 0 | 0 | 0 | 0 | 0 | 0 |
| | 0 | 0 | 1 | 0 | 0 | 0 | 0 | 0 | 0 | 0 | 0 | 0 | 0 | 0 | 0 | 0 | 0 | 0 | 0 | 0 | 0 | 0 | 0 | 0 | 0 | 0 | 0 | 0 | 0 | 0 | 0 |
| **2nd** | 0 | 0 | 0 | 1 | 0 | 0 | 0 | 0 | 0 | 0 | 0 | 0 | 0 | 0 | 0 | 0 | 0 | 0 | 0 | 0 | 0 | 0 | 0 | 0 | 0 | 0 | 0 | 0 | 0 | 0 | 0 |
| | 0 | 0 | 0 | 0 | 1 | 0 | 0 | 0 | 0 | 0 | 0 | 0 | 0 | 0 | 0 | 0 | 0 | 0 | 0 | 0 | 0 | 0 | 0 | 0 | 0 | 0 | 0 | 0 | 0 | 0 | 0 |
| | 0 | 0 | 0 | 0 | 0 | 1 | 0 | 0 | 0 | 0 | 0 | 0 | 0 | 0 | 0 | 0 | 0 | 0 | 0 | 0 | 0 | 0 | 0 | 0 | 0 | 0 | 0 | 0 | 0 | 0 | 0 |
| | 0 | 0 | 0 | 0 | 0 | 0 | 1 | 0 | 0 | 0 | 0 | 0 | 0 | 0 | 0 | 0 | 0 | 0 | 0 | 0 | 0 | 0 | 0 | 0 | 0 | 0 | 0 | 0 | 0 | 0 | 0 |
| **3rd** | 0 | 0 | 0 | 0 | 0 | 0 | 0 | 1 | 0 | 0 | 0 | 0 | 0 | 0 | 0 | 0 | 0 | 0 | 0 | 0 | 0 | 0 | 0 | 0 | 0 | 0 | 0 | 0 | 0 | 0 | 0 |
| | 0 | 0 | 0 | 0 | 0 | 0 | 0 | 0 | 1 | 0 | 0 | 0 | 0 | 0 | 0 | 0 | 0 | 0 | 0 | 0 | 0 | 0 | 0 | 0 | 0 | 0 | 0 | 0 | 0 | 0 | 0 |
| | 0 | 0 | 0 | 0 | 0 | 0 | 0 | 0 | 0 | 1 | 0 | 0 | 0 | 0 | 0 | 0 | 0 | 0 | 0 | 0 | 0 | 0 | 0 | 0 | 0 | 0 | 0 | 0 | 0 | 0 | 0 |
| | 0 | 0 | 0 | 0 | 0 | 0 | 0 | 0 | 0 | 0 | 1 | 0 | 0 | 0 | 0 | 0 | 0 | 0 | 0 | 0 | 0 | 0 | 0 | 0 | 0 | 0 | 0 | 0 | 0 | 0 | 0 |
| | 0 | 0 | 0 | 0 | 0 | 0 | 0 | 0 | 0 | 0 | 0 | 1 | 0 | 0 | 0 | 0 | 0 | 0 | 0 | 0 | 0 | 0 | 0 | 0 | 0 | 0 | 0 | 0 | 0 | 0 | 0 |
| | 0 | 0 | 0 | 0 | 0 | 0 | 0 | 0 | 0 | 0 | 0 | 0 | 1 | 0 | 0 | 0 | 0 | 0 | 0 | 0 | 0 | 0 | 0 | 0 | 0 | 0 | 0 | 0 | 0 | 0 | 0 |
| | 0 | 0 | 0 | 0 | 0 | 0 | 0 | 0 | 0 | 0 | 0 | 0 | 0 | 1 | 0 | 0 | 0 | 0 | 0 | 0 | 0 | 0 | 0 | 0 | 0 | 0 | 0 | 0 | 0 | 0 | 0 |
| | 0 | 0 | 0 | 0 | 0 | 0 | 0 | 0 | 0 | 0 | 0 | 0 | 0 | 0 | 1 | 0 | 0 | 0 | 0 | 0 | 0 | 0 | 0 | 0 | 0 | 0 | 0 | 0 | 0 | 0 | 0 |
| **4th** | 0 | 0 | 0 | 0 | 0 | 0 | 0 | 0 | 0 | 0 | 0 | 0 | 0 | 0 | 0 | 1 | 0 | 0 | 0 | 0 | 0 | 0 | 0 | 0 | 0 | 0 | 0 | 0 | 0 | 0 | 0 |
| | 0 | 0 | 0 | 0 | 0 | 0 | 0 | 0 | 0 | 0 | 0 | 0 | 0 | 0 | 0 | 0 | 1 | 0 | 0 | 0 | 0 | 0 | 0 | 0 | 0 | 0 | 0 | 0 | 0 | 0 | 0 |
| | 0 | 0 | 0 | 0 | 0 | 0 | 0 | 0 | 0 | 0 | 0 | 0 | 0 | 0 | 0 | 0 | 0 | 1 | 0 | 0 | 0 | 0 | 0 | 0 | 0 | 0 | 0 | 0 | 0 | 0 | 0 |
| | 0 | 0 | 0 | 0 | 0 | 0 | 0 | 0 | 0 | 0 | 0 | 0 | 0 | 0 | 0 | 0 | 0 | 0 | 1 | 0 | 0 | 0 | 0 | 0 | 0 | 0 | 0 | 0 | 0 | 0 | 0 |
| | 0 | 0 | 0 | 0 | 0 | 0 | 0 | 0 | 0 | 0 | 0 | 0 | 0 | 0 | 0 | 0 | 0 | 0 | 0 | 1 | 0 | 0 | 0 | 0 | 0 | 0 | 0 | 0 | 0 | 0 | 0 |
| | 0 | 0 | 0 | 0 | 0 | 0 | 0 | 0 | 0 | 0 | 0 | 0 | 0 | 0 | 0 | 0 | 0 | 0 | 0 | 0 | 1 | 0 | 0 | 0 | 0 | 0 | 0 | 0 | 0 | 0 | 0 |
| | 0 | 0 | 0 | 0 | 0 | 0 | 0 | 0 | 0 | 0 | 0 | 0 | 0 | 0 | 0 | 0 | 0 | 0 | 0 | 0 | 0 | 1 | 0 | 0 | 0 | 0 | 0 | 0 | 0 | 0 | 0 |
| | 0 | 0 | 0 | 0 | 0 | 0 | 0 | 0 | 0 | 0 | 0 | 0 | 0 | 0 | 0 | 0 | 0 | 0 | 0 | 0 | 0 | 0 | 1 | 0 | 0 | 0 | 0 | 0 | 0 | 0 | 0 |
| | 0 | 0 | 0 | 0 | 0 | 0 | 0 | 0 | 0 | 0 | 0 | 0 | 0 | 0 | 0 | 0 | 0 | 0 | 0 | 0 | 0 | 0 | 0 | 1 | 0 | 0 | 0 | 0 | 0 | 0 | 0 |
| | 0 | 0 | 0 | 0 | 0 | 0 | 0 | 0 | 0 | 0 | 0 | 0 | 0 | 0 | 0 | 0 | 0 | 0 | 0 | 0 | 0 | 0 | 0 | 0 | 1 | 0 | 0 | 0 | 0 | 0 | 0 |
| | 0 | 0 | 0 | 0 | 0 | 0 | 0 | 0 | 0 | 0 | 0 | 0 | 0 | 0 | 0 | 0 | 0 | 0 | 0 | 0 | 0 | 0 | 0 | 0 | 0 | 1 | 0 | 0 | 0 | 0 | 0 |
| | 0 | 0 | 0 | 0 | 0 | 0 | 0 | 0 | 0 | 0 | 0 | 0 | 0 | 0 | 0 | 0 | 0 | 0 | 0 | 0 | 0 | 0 | 0 | 0 | 0 | 0 | 1 | 0 | 0 | 0 | 0 |
| | 0 | 0 | 0 | 0 | 0 | 0 | 0 | 0 | 0 | 0 | 0 | 0 | 0 | 0 | 0 | 0 | 0 | 0 | 0 | 0 | 0 | 0 | 0 | 0 | 0 | 0 | 0 | 1 | 0 | 0 | 0 |
| | 0 | 0 | 0 | 0 | 0 | 0 | 0 | 0 | 0 | 0 | 0 | 0 | 0 | 0 | 0 | 0 | 0 | 0 | 0 | 0 | 0 | 0 | 0 | 0 | 0 | 0 | 0 | 0 | 1 | 0 | 0 |
| | 0 | 0 | 0 | 0 | 0 | 0 | 0 | 0 | 0 | 0 | 0 | 0 | 0 | 0 | 0 | 0 | 0 | 0 | 0 | 0 | 0 | 0 | 0 | 0 | 0 | 0 | 0 | 0 | 0 | 1 | 0 |
| | 0 | 0 | 0 | 0 | 0 | 0 | 0 | 0 | 0 | 0 | 0 | 0 | 0 | 0 | 0 | 0 | 0 | 0 | 0 | 0 | 0 | 0 | 0 | 0 | 0 | 0 | 0 | 0 | 0 | 0 | 1 |

Figure A.3: Correlation matrix of Itô signatures with orders truncated to 4 for a 2-dimensional Brownian motion with inter-dimensional correlation $\rho = 0$.

**Stratonovich signature.** Propositions 3–4 in the main paper provide formulas for calculating the correlation structure of Stratonovich signatures for Brownian motions. The following proposition gives recursive formulas for calculating $\mathbb{E}\left[S(\mathbf{X})_t^{i_1,\ldots,i_{2n},S} S(\mathbf{X})_t^{j_1,\ldots,j_{2m},S}\right]$ and $\mathbb{E}\left[S(\mathbf{X})_t^{i_1,\ldots,i_{2n-1},S} S(\mathbf{X})_t^{j_1,\ldots,j_{2m-1},S}\right]$, which extends Proposition 3 in the main paper.

**Proposition A.1.** *Let* $\mathbf{X}$ *be a* $d$-*dimensional Brownian motion given by Equation* (6). *For any* $l, t \geq 0$ *and* $m, n = 1, 2, \ldots$, *define* $f_{2n,2m}(l,t) := \mathbb{E}\left[S(\mathbf{X})_l^{i_1,\ldots,i_{2n},S} S(\mathbf{X})_t^{j_1,\ldots,j_{2m},S}\right]$, *we have:*

$$f_{2n,2m}(l,t) = g_{2n,2m}(l,t) + \frac{1}{2}\rho_{j_{2m-1}j_{2m}}\sigma_{j_{2m-1}}\sigma_{j_{2m}}\int_0^t f_{2n,2m-2}(l,s)\mathrm{d}s, \tag{A.1}$$

$$g_{2n,2m}(l,t) = \rho_{i_{2n}j_{2m}}\sigma_{i_{2n}}\sigma_{j_{2m}}\int_0^{l\wedge t} f_{2n-1,2m-1}(s,s)\mathrm{d}s$$

$$+ \frac{1}{2}\rho_{i_{2n-1}i_{2n}}\sigma_{i_{2n-1}}\sigma_{i_{2n}}\int_0^l g_{2n-2,2m}(s,t)\mathrm{d}s, \tag{A.2}$$

*with initial conditions*

$$f_{0,0}(l,t) = 1, \tag{A.3}$$

$$g_{0,2m}(l,t) = 0. \tag{A.4}$$

*In addition, define* $f_{2n-1,2m-1}(l,t) := \mathbb{E}\left[S(\mathbf{X})_l^{i_1,\ldots,i_{2n-1},S} S(\mathbf{X})_t^{j_1,\ldots,j_{2m-1},S}\right]$, *we have:*

$$f_{2n-1,2m-1}(l,t) = g_{2n-1,2m-1}(l,t) + \frac{1}{2}\rho_{j_{2m-2}j_{2m-1}}\sigma_{j_{2m-2}}\sigma_{j_{2m-1}}\int_0^t f_{2n-1,2m-3}(l,s)\mathrm{d}s, \tag{A.5}$$

$$g_{2n-1,2m-1}(l,t) = \rho_{i_{2n-1}j_{2m-1}}\sigma_{i_{2n-1}}\sigma_{j_{2m-1}}\int_0^{l\wedge t} f_{2n-2,2m-2}(s,s)\mathrm{d}s$$

$$+ \frac{1}{2}\rho_{i_{2n-2}i_{2n-1}}\sigma_{i_{2n-2}}\sigma_{i_{2n-1}}\int_0^l g_{2n-3,2m-1}(s,t)\mathrm{d}s, \tag{A.6}$$

*with initial conditions*

$$f_{1,1}(l,t) = \rho_{i_1 j_1}\sigma_{i_1}\sigma_{j_1}(l \wedge t), \tag{A.7}$$

$$g_{1,2m-1}(l,t) = \rho_{i_1 j_{2m-1}}\frac{1}{2^{m-1}}\frac{(l\wedge t)^{m-1}}{(m-1)!}\sigma_{i_1}\prod_{k=1}^{2m-1}\sigma_{j_k}\prod_{k=1}^{m-1}\rho_{j_{2k-1}j_{2k}}. \tag{A.8}$$

*Here, $x \wedge y$ represents the smaller value between $x$ and $y$.*

The following two examples show the correlation structures of Stratonovich signatures for 2-dimensional Brownian motions with inter-dimensional correlations $\rho = 0.6$ and $\rho = 0$, respectively, calculated using Propositions 3–4 in the main paper and Proposition A.1.

**Example A.3.** *Consider a 2-dimensional Brownian motion given by Equation (6) with inter-dimensional correlation $\rho = 0.6$. Figure A.4 shows the correlation matrix of its Stratonovich signatures with orders truncated to 4 calculated using Propositions 3 and A.1. The figure illustrates that the correlation matrix has an odd–even alternating structure.*

|  | 0th | 1st | | 2nd order | | | | 3rd order | | | | | | | | 4th order | | | | | | | | | | | | | | | |
|---|---|---|---|---|---|---|---|---|---|---|---|---|---|---|---|---|---|---|---|---|---|---|---|---|---|---|---|---|---|---|---|
| 0th | 1 | 0 | 0 | 0.58 | 0.39 | 0.39 | 0.58 | 0 | 0 | 0 | 0 | 0 | 0 | 0 | 0 | 0.29 | 0.21 | 0.23 | 0.35 | 0.23 | 0.15 | 0.14 | 0.21 | 0.21 | 0.14 | 0.15 | 0.23 | 0.35 | 0.23 | 0.21 | 0.29 |
| 1st | 0 | 1 | 0.6 | 0 | 0 | 0 | 0 | 0.77 | 0.54 | 0.59 | 0.61 | 0.54 | 0.36 | 0.61 | 0.46 | 0 | 0 | 0 | 0 | 0 | 0 | 0 | 0 | 0 | 0 | 0 | 0 | 0 | 0 | 0 | 0 |
|  | 0 | 0.6 | 1 | 0 | 0 | 0 | 0 | 0.46 | 0.61 | 0.36 | 0.54 | 0.61 | 0.59 | 0.54 | 0.77 | 0 | 0 | 0 | 0 | 0 | 0 | 0 | 0 | 0 | 0 | 0 | 0 | 0 | 0 | 0 | 0 |
| 2nd | 0.58 | 0 | 0 | 1 | 0.68 | 0.68 | 0.57 | 0 | 0 | 0 | 0 | 0 | 0 | 0 | 0 | 0.85 | 0.6 | 0.67 | 0.66 | 0.67 | 0.44 | 0.61 | 0.49 | 0.6 | 0.42 | 0.44 | 0.44 | 0.66 | 0.44 | 0.49 | 0.41 |
|  | 0.39 | 0 | 0 | 0.68 | 1 | 0.46 | 0.68 | 0 | 0 | 0 | 0 | 0 | 0 | 0 | 0 | 0.57 | 0.78 | 0.45 | 0.68 | 0.66 | 0.71 | 0.55 | 0.78 | 0.4 | 0.55 | 0.3 | 0.45 | 0.56 | 0.66 | 0.4 | 0.57 |
|  | 0.39 | 0 | 0 | 0.68 | 0.46 | 1 | 0.68 | 0 | 0 | 0 | 0 | 0 | 0 | 0 | 0 | 0.57 | 0.4 | 0.66 | 0.56 | 0.45 | 0.3 | 0.55 | 0.4 | 0.78 | 0.55 | 0.71 | 0.66 | 0.68 | 0.45 | 0.78 | 0.57 |
|  | 0.58 | 0 | 0 | 0.57 | 0.68 | 0.68 | 1 | 0 | 0 | 0 | 0 | 0 | 0 | 0 | 0 | 0.41 | 0.49 | 0.44 | 0.66 | 0.44 | 0.44 | 0.42 | 0.6 | 0.49 | 0.61 | 0.44 | 0.67 | 0.66 | 0.67 | 0.6 | 0.85 |
| 3rd | 0 | 0.77 | 0.46 | 0 | 0 | 0 | 0 | 1 | 0.7 | 0.76 | 0.64 | 0.7 | 0.46 | 0.64 | 0.45 | 0 | 0 | 0 | 0 | 0 | 0 | 0 | 0 | 0 | 0 | 0 | 0 | 0 | 0 | 0 | 0 |
|  | 0 | 0.54 | 0.61 | 0 | 0 | 0 | 0 | 0.7 | 1 | 0.53 | 0.75 | 0.57 | 0.72 | 0.5 | 0.64 | 0 | 0 | 0 | 0 | 0 | 0 | 0 | 0 | 0 | 0 | 0 | 0 | 0 | 0 | 0 | 0 |
|  | 0 | 0.59 | 0.36 | 0 | 0 | 0 | 0 | 0.76 | 0.53 | 1 | 0.72 | 0.53 | 0.35 | 0.72 | 0.46 | 0 | 0 | 0 | 0 | 0 | 0 | 0 | 0 | 0 | 0 | 0 | 0 | 0 | 0 | 0 | 0 |
|  | 0 | 0.61 | 0.54 | 0 | 0 | 0 | 0 | 0.64 | 0.75 | 0.72 | 1 | 0.5 | 0.53 | 0.57 | 0.7 | 0 | 0 | 0 | 0 | 0 | 0 | 0 | 0 | 0 | 0 | 0 | 0 | 0 | 0 | 0 | 0 |
|  | 0 | 0.54 | 0.61 | 0 | 0 | 0 | 0 | 0.7 | 0.57 | 0.53 | 0.5 | 1 | 0.72 | 0.75 | 0.64 | 0 | 0 | 0 | 0 | 0 | 0 | 0 | 0 | 0 | 0 | 0 | 0 | 0 | 0 | 0 | 0 |
|  | 0 | 0.36 | 0.59 | 0 | 0 | 0 | 0 | 0.46 | 0.72 | 0.35 | 0.53 | 0.72 | 1 | 0.53 | 0.76 | 0 | 0 | 0 | 0 | 0 | 0 | 0 | 0 | 0 | 0 | 0 | 0 | 0 | 0 | 0 | 0 |
|  | 0 | 0.61 | 0.54 | 0 | 0 | 0 | 0 | 0.64 | 0.5 | 0.72 | 0.57 | 0.75 | 0.53 | 1 | 0.7 | 0 | 0 | 0 | 0 | 0 | 0 | 0 | 0 | 0 | 0 | 0 | 0 | 0 | 0 | 0 | 0 |
|  | 0 | 0.46 | 0.77 | 0 | 0 | 0 | 0 | 0.45 | 0.64 | 0.46 | 0.7 | 0.64 | 0.76 | 0.7 | 1 | 0 | 0 | 0 | 0 | 0 | 0 | 0 | 0 | 0 | 0 | 0 | 0 | 0 | 0 | 0 | 0 |
| 4th | 0.29 | 0 | 0 | 0.85 | 0.57 | 0.57 | 0.41 | 0 | 0 | 0 | 0 | 0 | 0 | 0 | 0 | 1 | 0.7 | 0.79 | 0.67 | 0.79 | 0.52 | 0.69 | 0.5 | 0.7 | 0.49 | 0.52 | 0.44 | 0.67 | 0.44 | 0.5 | 0.36 |
|  | 0.21 | 0 | 0 | 0.6 | 0.78 | 0.4 | 0.49 | 0 | 0 | 0 | 0 | 0 | 0 | 0 | 0 | 0.7 | 1 | 0.56 | 0.78 | 0.67 | 0.81 | 0.56 | 0.73 | 0.5 | 0.7 | 0.36 | 0.51 | 0.53 | 0.66 | 0.39 | 0.5 |
|  | 0.23 | 0 | 0 | 0.67 | 0.45 | 0.66 | 0.44 | 0 | 0 | 0 | 0 | 0 | 0 | 0 | 0 | 0.79 | 0.56 | 1 | 0.73 | 0.62 | 0.41 | 0.78 | 0.51 | 0.67 | 0.47 | 0.74 | 0.57 | 0.59 | 0.39 | 0.66 | 0.44 |
|  | 0.35 | 0 | 0 | 0.66 | 0.68 | 0.56 | 0.66 | 0 | 0 | 0 | 0 | 0 | 0 | 0 | 0 | 0.67 | 0.78 | 0.73 | 1 | 0.59 | 0.61 | 0.62 | 0.78 | 0.53 | 0.62 | 0.52 | 0.73 | 0.58 | 0.59 | 0.53 | 0.67 |
|  | 0.23 | 0 | 0 | 0.67 | 0.66 | 0.45 | 0.44 | 0 | 0 | 0 | 0 | 0 | 0 | 0 | 0 | 0.79 | 0.67 | 0.62 | 0.59 | 1 | 0.74 | 0.78 | 0.66 | 0.56 | 0.47 | 0.41 | 0.39 | 0.73 | 0.57 | 0.51 | 0.44 |
|  | 0.15 | 0 | 0 | 0.44 | 0.71 | 0.3 | 0.44 | 0 | 0 | 0 | 0 | 0 | 0 | 0 | 0 | 0.52 | 0.81 | 0.41 | 0.61 | 0.74 | 1 | 0.56 | 0.81 | 0.36 | 0.56 | 0.27 | 0.41 | 0.52 | 0.74 | 0.36 | 0.52 |
|  | 0.14 | 0 | 0 | 0.61 | 0.55 | 0.55 | 0.42 | 0 | 0 | 0 | 0 | 0 | 0 | 0 | 0 | 0.69 | 0.56 | 0.78 | 0.62 | 0.78 | 0.56 | 1 | 0.7 | 0.56 | 0.44 | 0.56 | 0.47 | 0.62 | 0.47 | 0.7 | 0.49 |
|  | 0.21 | 0 | 0 | 0.49 | 0.78 | 0.4 | 0.6 | 0 | 0 | 0 | 0 | 0 | 0 | 0 | 0 | 0.5 | 0.73 | 0.51 | 0.78 | 0.66 | 0.81 | 0.7 | 1 | 0.39 | 0.56 | 0.36 | 0.56 | 0.53 | 0.67 | 0.5 | 0.7 |
|  | 0.21 | 0 | 0 | 0.6 | 0.4 | 0.78 | 0.49 | 0 | 0 | 0 | 0 | 0 | 0 | 0 | 0 | 0.7 | 0.5 | 0.67 | 0.53 | 0.56 | 0.36 | 0.56 | 0.39 | 1 | 0.7 | 0.81 | 0.66 | 0.78 | 0.51 | 0.73 | 0.5 |
|  | 0.14 | 0 | 0 | 0.42 | 0.55 | 0.55 | 0.61 | 0 | 0 | 0 | 0 | 0 | 0 | 0 | 0 | 0.49 | 0.7 | 0.47 | 0.62 | 0.47 | 0.56 | 0.44 | 0.56 | 0.7 | 1 | 0.56 | 0.78 | 0.62 | 0.78 | 0.56 | 0.69 |
|  | 0.15 | 0 | 0 | 0.44 | 0.3 | 0.71 | 0.44 | 0 | 0 | 0 | 0 | 0 | 0 | 0 | 0 | 0.52 | 0.36 | 0.74 | 0.52 | 0.41 | 0.27 | 0.56 | 0.36 | 0.81 | 0.56 | 1 | 0.74 | 0.61 | 0.41 | 0.81 | 0.52 |
|  | 0.23 | 0 | 0 | 0.44 | 0.45 | 0.66 | 0.67 | 0 | 0 | 0 | 0 | 0 | 0 | 0 | 0 | 0.44 | 0.51 | 0.57 | 0.73 | 0.39 | 0.41 | 0.47 | 0.56 | 0.66 | 0.78 | 0.74 | 1 | 0.59 | 0.62 | 0.67 | 0.79 |
|  | 0.35 | 0 | 0 | 0.66 | 0.56 | 0.68 | 0.66 | 0 | 0 | 0 | 0 | 0 | 0 | 0 | 0 | 0.67 | 0.53 | 0.59 | 0.58 | 0.73 | 0.52 | 0.62 | 0.53 | 0.78 | 0.62 | 0.61 | 0.59 | 1 | 0.73 | 0.78 | 0.67 |
|  | 0.23 | 0 | 0 | 0.44 | 0.66 | 0.45 | 0.67 | 0 | 0 | 0 | 0 | 0 | 0 | 0 | 0 | 0.44 | 0.66 | 0.39 | 0.59 | 0.57 | 0.74 | 0.47 | 0.67 | 0.51 | 0.78 | 0.41 | 0.62 | 0.73 | 1 | 0.56 | 0.79 |
|  | 0.21 | 0 | 0 | 0.49 | 0.4 | 0.78 | 0.6 | 0 | 0 | 0 | 0 | 0 | 0 | 0 | 0 | 0.5 | 0.39 | 0.66 | 0.53 | 0.51 | 0.36 | 0.7 | 0.5 | 0.73 | 0.56 | 0.81 | 0.67 | 0.78 | 0.56 | 1 | 0.7 |
|  | 0.29 | 0 | 0 | 0.41 | 0.57 | 0.57 | 0.85 | 0 | 0 | 0 | 0 | 0 | 0 | 0 | 0 | 0.36 | 0.5 | 0.44 | 0.67 | 0.44 | 0.52 | 0.49 | 0.7 | 0.5 | 0.69 | 0.52 | 0.79 | 0.67 | 0.79 | 0.7 | 1 |

Figure A.4: Correlation matrix of Stratonovich signatures with orders truncated to 4 for a 2-dimensional Brownian motion with inter-dimensional correlation $\rho = 0.6$.

**Example A.4.** *Consider a 2-dimensional Brownian motion given by Equation (6) with inter-dimensional correlation $\rho = 0$. Figure A.5 shows the correlation matrix of its Stratonovich signatures with orders truncated to 4 calculated using Propositions 3 and A.1. The figure demonstrates that the correlation matrix has an odd–even alternating structure, even though different dimensions of the Brownian motion are mutually independent ($\rho = 0$). This is different from the result for Itô signatures shown in Example A.2, where all Itô signatures are mutually uncorrelated.*

*In this case, suppose that one includes all Stratonovich signatures of orders up to $K = 4$ in the Lasso regression given by Equation (5), and the true model given by Equation (3) has beta coefficients*

Figure A.5: Correlation matrix of Stratonovich signatures with orders truncated to 4 for a 2-dimensional Brownian motion with inter-dimensional correlation $\rho = 0$.

$\beta_0 = 0$, $\beta_1 > 0$, $\beta_2 > 0$, $\beta_{1,1} > 0$, $\beta_{1,2} > 0$, $\beta_{2,1} > 0$, $\beta_{2,2} < 0$, and $\beta_{i_1,i_2,i_3} = \beta_{i_1,i_2,i_3,i_4} = 0$. Then, by Proposition 3,

$$\Sigma^S_{A^{*c},A^*}(\Sigma^S_{A^*,A^*})^{-1}\mathrm{sign}(\boldsymbol{\beta}_{A^*}) = (0, 0.77, 0.5, 0, 0.5, 0.5, 0, 0.5, 0.77, 1.01, 0.73, 0.47, 0,$$
$$0.47, 0, 0.58, 0.73, 0.73, -0.58, 0, 0.47, 0, 0.47, 0.73, -1.01),$$

which does not satisfy the irrepresentable condition defined in Definition 3 because $1.01 > 1$.

## D.2 OU PROCESS

Deriving explicit formulas for calculating the exact correlation between signatures of OU processes (both Itô and Stratonovich) is complicated. Here we provide an example to show the general approach for calculating the correlation. The proof of this example is given in Appendix K, and one can use a similar routine to compute the correlation for other setups of OU processes.

**Example A.5.** *Consider a 1-dimensional OU process* $\mathbf{X}_t = Y_t$ *with a mean reversion speed* $\kappa > 0$:

$$\mathrm{d}Y_t = -\kappa Y_t \mathrm{d}t + \mathrm{d}W_t, \quad Y_0 = 0. \tag{A.9}$$

*The correlation coefficients between its 0-th order and 2nd order signatures are*

$$\frac{\mathbb{E}\left[S(\mathbf{X})^{0,I}_T S(\mathbf{X})^{1,1,I}_T\right]}{\sqrt{\mathbb{E}\left[S(\mathbf{X})^{0,I}_T\right]^2 \mathbb{E}\left[S(\mathbf{X})^{1,1,I}_T\right]^2}} = \frac{-2\kappa T - e^{-2\kappa T} + 1}{\sqrt{4\kappa T e^{-2\kappa T} + 3e^{-4\kappa T} - 6e^{-2\kappa T} - 4\kappa T + 3 + 4\kappa^2 T^2}},$$

$$\frac{\mathbb{E}\left[S(\mathbf{X})^{0,S}_T S(\mathbf{X})^{1,1,S}_T\right]}{\sqrt{\mathbb{E}\left[S(\mathbf{X})^{0,S}_T\right]^2 \mathbb{E}\left[S(\mathbf{X})^{1,1,S}_T\right]^2}} = \frac{\sqrt{3}}{3},$$

*for Itô and Stratonovich signatures, respectively. The proof is provided in Appendix K.*

*Figure A.6a shows the absolute values of correlation coefficients between the 0-th order and 2nd order signatures calculated using the formulas above under different values of* $\kappa$. *Notably, the correlation for Itô signatures rises with* $\kappa$, *while the correlation for Stratonovich signatures remains fixed at* $\frac{\sqrt{3}}{3}$.

*We further perform simulations to estimate the correlation coefficients for higher-order signatures of the OU process. We generate 10,000 sample paths of the OU process using the methods discussed*

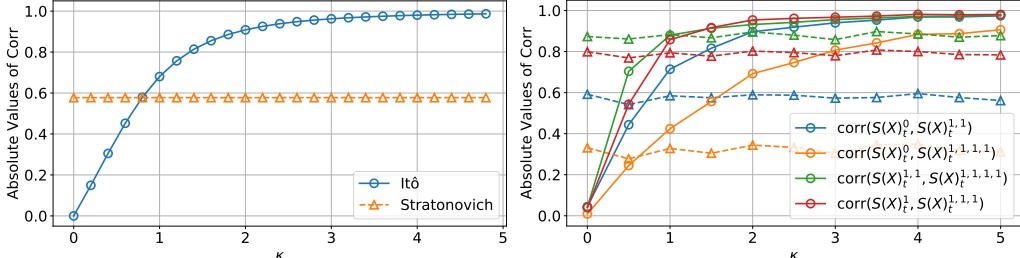

(a) Correlation between the 0-th and the 2nd order signatures.

(b) Correlation between the first four order signatures.

Figure A.6: Absolute values of correlation coefficients between signatures of the 1-dimensional OU process. Solid lines correspond to Itô signatures and dashed lines correspond to Stratonovich signatures.

*in Appendix F. For each path, we calculate the corresponding signatures and then estimate the sample correlation matrix based on the 10,000 simulated samples. Figure A.6b shows the simulation results for the absolute values of correlation coefficients between the first four order signatures under different values of $\kappa$. Consistent with the observation in Figure A.6a, the correlation for Itô signatures rises with $\kappa$, while the correlation for Stratonovich signatures remains relatively stable. Notably, the correlations for Itô signatures are zero when $\kappa = 0$, which reduces to a Brownian motion. In addition, when $\kappa$ is sufficiently large, the absolute values of correlation coefficients for Itô signatures exceed those for Stratonovich signatures.*

Recall that the irrepresentable condition, as defined in Definition 3, illustrates that a higher correlation generally leads to poorer consistency. Therefore, based on Example A.5, we can expect that the Lasso is more consistent when using Itô signatures for small values of $\kappa$ (weaker mean reversion), and more consistent when using Stratonovich signatures for large values of $\kappa$ (stronger mean reversion). This provides a theoretical explanation for our observations in Section 4.2 of the main paper: When processes are sufficiently rough or mean reverting (El Euch et al., 2018; Gatheral et al., 2018), using Lasso with Stratonovich signatures will likely lead to higher statistical consistency compared to Itô signatures.

## E  IRREPRESENTABLE CONDITION FOR ITÔ SIGNATURES OF BROWNIAN MOTION WITH EQUAL INTER-DIMENSIONAL CORRELATION

In this appendix, we investigate the irrepresentable condition for Itô signatures of a multi-dimensional Brownian motion with equal inter-dimensional correlation. This analysis not only provides further insights into the irrepresentable condition but also demonstrates the tightness of the sufficient condition presented in Proposition 7 in our main paper.

The following proposition characterizes whether the irrepresentable condition holds under different values of inter-dimensional correlation for the Brownian motion when using Itô signatures. For mathematical simplicity, we assume that only the first order signatures are included in the regression model.

**Proposition A.2.** *For a multi-dimensional Brownian motion given by Equation* (6) *with equal inter-dimensional correlation $\rho = \rho_{ij}$, assume that only its first order signatures are included in* (3), *and all true beta coefficients are positive. Then, the irrepresentable condition for the correlation matrix of Itô signatures holds if $\rho \in (-\frac{1}{2\#A_1^*}, 1)$, and does not hold if $\rho \in (-\frac{1}{\#A_1^*}, -\frac{1}{2\#A_1^*}]$.*

**Remark A.1.** *Proposition A.2 only discusses the results for $\rho \in (-\frac{1}{\#A_1^*}, 1)$ because, if $\rho < -\frac{1}{\#A_1^*}$, the inter-dimensional correlation matrix for the Brownian motion is not positive definite.*

Proposition A.2 provides insights into the sufficient condition given by Proposition 7. In particular, Proposition A.2 demonstrates that the sufficient condition (12) is tight under the equal inter-dimensional correlation setup for $\rho < 0$. Meanwhile, it also reveals that, for $\rho > 0$, the irrep-

resentable condition always holds but may not satisfy (12). In other words, in this equal inter-dimensional correlation setup, (12) is tight when $\rho < 0$, and is loose when $\rho > 0$.

## F  DETAILS FOR SIMULATIONS

This appendix provides additional technical details, computational cost, and robustness checks for the simulations conducted in this paper.

### F.1  MORE TECHNICAL DETAILS

**Simulation of processes.**  We simulate the $i$-th dimension of the Brownian motion, $W_t^i$, and OU process, $Y_t^i$, by discretizing the stochastic differential equations of the processes using the Euler–Maruyama method:

- Brownian motion: $W_{t_{k+1}}^i = W_{t_k}^i + \sqrt{\Delta t}\varepsilon_k^i$, $W_0^i = 0$;
- OU process: $Y_{t_{k+1}}^i = Y_{t_k}^i - \kappa_i Y_{t_k}^i \Delta t + \sqrt{\Delta t}\varepsilon_k^i$, $Y_0^i = 0$.

Here, $0 = t_0 < t_1 < \cdots < t_N = T$, $t_{k+1} - t_k = \Delta t = T/N$ for any $k$, and $\varepsilon_k^i$ are randomly drawn from the standard normal distribution. The number of steps is set to $N = 100$.

The $i$-th dimension of the random walk and AR(1) model, both denoted by $Z_t^i$, are simulated using the following formulas:

- Random walk: $Z_{t_{k+1}}^i = Z_{t_k}^i + e_k^i$, $Z_0^i = 0$;
- AR(1) model: $Z_{t_{k+1}}^i = \phi_i Z_{t_k}^i + \varepsilon_k^i$, $Z_0^i = 0$.

Here, $0 = t_0 < t_1 < \cdots < t_N = T$, $t_{k+1} - t_k = \Delta t = T/N$ for any $k$, $e_k^i$ are randomly drawn from the following distribution:

$$\mathbb{P}(e_k^i = +1) = \mathbb{P}(e_k^i = -1) = 0.5,$$

and $\varepsilon_k^i$ are randomly drawn from the standard normal distribution. The number of steps is set to $N = 100$.

After simulating each dimension of the processes, we simulate the inter-dimensional correlation between different dimensions of the processes using the Cholesky decomposition. Specifically, we set the inter-dimensional correlation matrix $\Gamma\Gamma^\top$ based on the setups described in the main paper and calculate $\Gamma$ using the Cholesky decomposition. Finally, we generate $\mathbf{X}$ using Equations (6) or (11).

In all of our simulations, we set the length of the processes to $T = 1$, and the initial values of the processes to zero. These choices have no impact on the results because the signatures of a path $\mathbf{X}$ are invariant under a time reparametrization and a shift of the starting point of $\mathbf{X}$, see, for example, Chevyrev & Kormilitzin (2016).

**Calculation of integrals.**  The calculation of Itô and Stratonovich signatures requires the calculation of Itô and Stratonovich integrals. By definition, these integrals are computed using the following schemes:

- Itô integral: $\int_0^T A_t \mathrm{d}B_t \approx \sum_{k=0}^{N-1} A_{t_k}(B_{t_{k+1}} - B_{t_k})$;
- Stratonovich integral: $\int_0^T A_t \circ \mathrm{d}B_t \approx \sum_{k=0}^{N-1} \frac{1}{2}(A_{t_k} + A_{t_{k+1}})(B_{t_{k+1}} - B_{t_k})$.

Here, we set $0 = t_0 < t_1 < \cdots < t_N = T$ and $t_{k+1} - t_k = \Delta t = T/N$ for any $k$.

### F.2  COMPUTATIONAL DETAILS

- The simulations are implemented using Python 3.7.
- The simulations are run on a laptop with an Intel(R) Core(TM) i7-9750H CPU @ 2.60GHz.

- The random seed is set to 0 for reproducibility.

- The Lasso regressions are performed using the `sklearn.linear_model.lars_path` package.

- Each individual experiment, including generating 100 paths, calculating their signatures, and performing the Lasso regression, can be completed within one second.

## F.3 ROBUSTNESS CHECKS

To show the robustness of our simulations shown in Figures 1 and 2 in Section 4 of the main paper, we present Figures A.7 and A.8, which include confidence intervals (shaded regions) for the estimated consistency rates of the Brownian motion/random walk and OU process/AR(1) model, respectively.

In Figures A.7 and A.8, we estimate the consistency rate by repeating the procedure described in Section 4 100 times, and this process is repeated 30 times to obtain the confidence interval for the estimation. Thus, these confidence intervals are based on 30 estimations of the consistency rate, with each estimation calculated using 100 experiments.

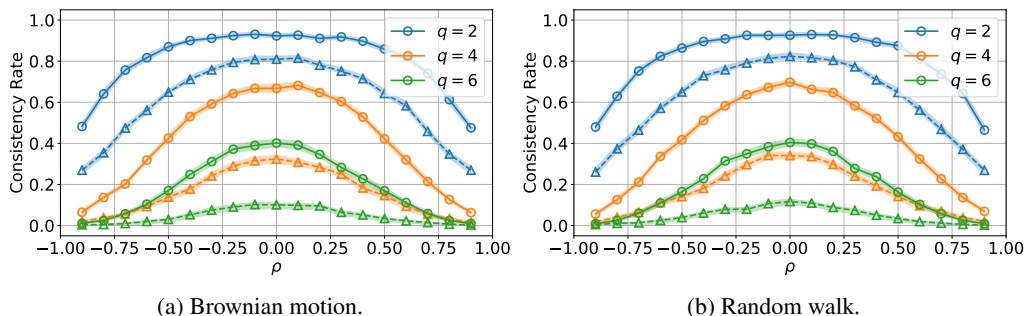

(a) Brownian motion.  (b) Random walk.

Figure A.7: Consistency rates for the Brownian motion and the random walk with different values of inter-dimensional correlation, $\rho$, and different numbers of true predictors, $q$. Solid lines correspond to Itô signatures and dashed lines correspond to Stratonovich signatures. Shaded regions are confidence intervals of the experiments.

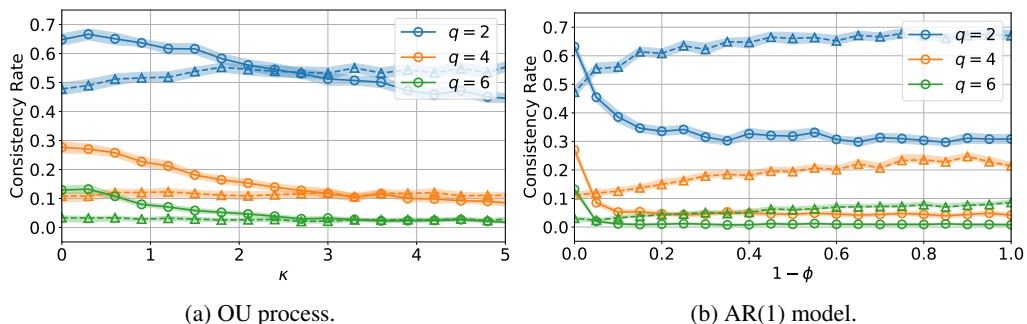

(a) OU process.  (b) AR(1) model.

Figure A.8: Consistency rates for the OU process and the AR(1) model with different parameters ($\kappa$ and $1 - \phi$) and different numbers of true predictors, $q$. Solid lines correspond to Itô signatures and dashed lines correspond to Stratonovich signatures. Shaded regions are confidence intervals of the experiments.

We observe that the confidence intervals of the consistency rates shown in Figures A.7 and A.8 are narrow. Moreover, the observations made in Figures 1 and 2 are consistent with the results presented here, further confirming the robustness of our findings.

# G  EXTENSIONS OF THE DEFINITION OF CONSISTENCY

As remarked in Section 5 of the main paper, the sign consistency may be too restrictive and can be relaxed in the context of signatures. This appendix provides numerical experiments that explore several extensions of consistency measures for signatures.

## G.1  PRECISION, RECALL, AND F1-SCORE

One possible approach for extending the definition of consistency is to use precision, recall, and the F1-score to evaluate the performance of the Lasso regression in selecting true predictors.

In particular, for a given tuning parameter $\lambda$ in the Lasso regression, Equation (5), let $A(\lambda)$ denote the set of selected predictors based on the Lasso:

$$A(\lambda) = \bigcup_{k=0}^{K} \{(i_1, \ldots, i_k) : \tilde{\beta}_{i_1, \ldots, i_k}(\lambda) \neq 0\},$$

where $\tilde{\beta}_{i_1, \ldots, i_k}(\lambda)$ are beta coefficients estimated using Equation (5). The true predictor set, $A^*$, is defined in Equation (4). We can calculate the true positive (TP), false positive (FP), true negative (TN), and false negative (TN) counts as follows:

$$\text{TP}(\lambda) = \#A(\lambda) \cap A^*, \ \text{FP}(\lambda) = \#A(\lambda) \cap A^{*c}, \ \text{TN}(\lambda) = \#A(\lambda)^c \cap A^{*c}, \ \text{FN}(\lambda) = \#A(\lambda)^c \cap A^*,$$

where $A^{*c}$ and $A(\lambda)^c$ represent the complements of $A^*$ and $A(\lambda)$, respectively. The precision, recall, and F1-score can then be defined as:

$$\text{precision}(\lambda) = \frac{\text{TP}(\lambda)}{\text{TP}(\lambda) + \text{FP}(\lambda)},$$

$$\text{recall}(\lambda) = \frac{\text{TP}(\lambda)}{\text{TP}(\lambda) + \text{FN}(\lambda)},$$

$$\text{F1-score}(\lambda) = \frac{2}{1/\text{recall}(\lambda) + 1/\text{precision}(\lambda)}.$$

We can then examine the maximum values of precision, recall, and F1-score as we vary the tuning parameter $\lambda$ in the Lasso. These maximum values reflect the best performance in terms of feature selection by the Lasso.

To assess these measures of consistency, we conducted simulations similar to those in Section 4.1 for the Brownian motion. Figure A.9 shows the maximum values of precision, recall, and F1-score for the Brownian motion with different inter-dimensional correlation values ($\rho$) and numbers of true predictors ($q$), averaged over 1,000 experiments. The results demonstrate that, similar to the findings in Figure 1, the maximum precision (Figure A.9a) and F1-score (Figure A.9c) reach their highest values when $\rho = 0$, and decrease as the dimensions of the Brownian motion become more correlated. It is also observed that the results for Itô signatures consistently outperform those for Stratonovich signatures. In addition, compared to Figure 1, these alternative measures of consistency exhibit a less pronounced decrease in performance as the absolute value of $\rho$ increases.

It is worth noting that the maximum recall rate (Figure A.9b) remains close to 1 under different parameters. This occurs because none of the predictors are selected by the Lasso when the tuning parameter $\lambda$ is extremely large. As a result, $\text{FN}(\lambda)$ is zero, and $\text{recall}(\lambda)$ is equal to 1. Consequently, the maximum recall rate is not an appropriate measure of consistency in the context of signatures when using the Lasso.

## G.2  OUT-OF-SAMPLE $R^2$

The out-of-sample $R^2$ is a commonly used measure of model performance in machine learning. It generalizes the notion of consistency as it considers the performance of the model on out-of-sample data, rather than solely focusing on the selection of true predictors by the Lasso. This aligns well with practical applications. Furthermore, it is consistent with the concept of universal nonlinearity in signatures, as described by Equation (2). Universal nonlinearity suggests that the true model can

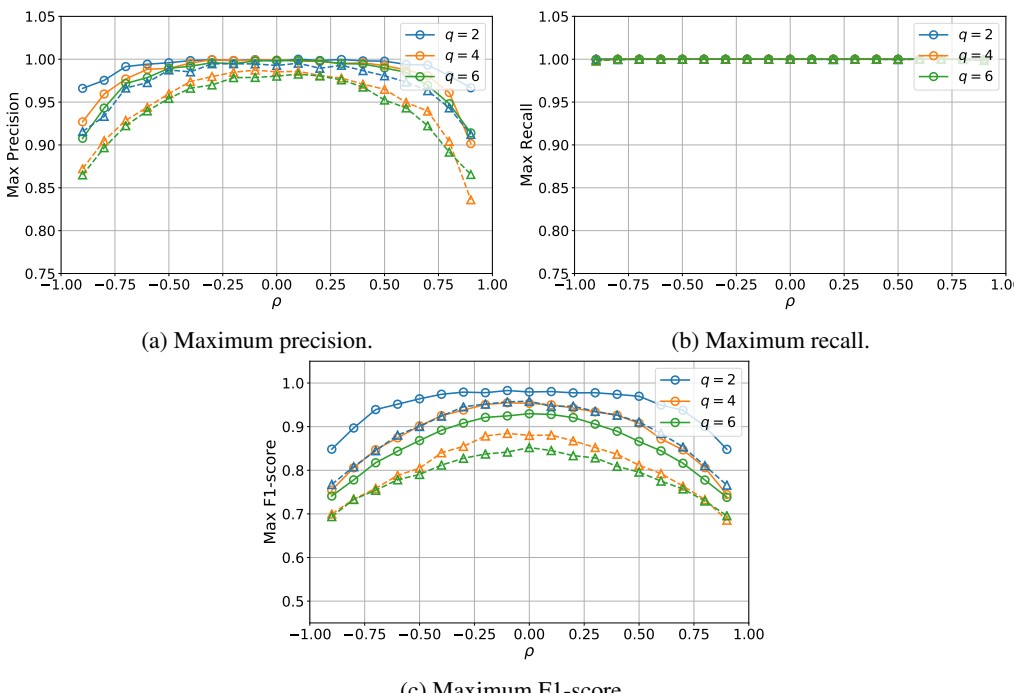

(a) Maximum precision.

(b) Maximum recall.

(c) Maximum F1-score.

Figure A.9: Maximum values of precision, recall, and F1-score for the Brownian motion with different values of inter-dimensional correlation, $\rho$, and different numbers of true predictors, $q$. Solid lines correspond to Itô signatures and dashed lines correspond to Stratonovich signatures.

be approximated by a linear combination of signatures, without requiring the exact selection of true predictors by the Lasso regression.

We conduct simulations to study the out-of-sample $R^2$ in the context of signatures using Lasso. Consider a two-dimensional ($d = 2$) Brownian motion with inter-dimensional correlation $\rho$; assume that there are $q = \#A^*$ true predictors in the true model (3), and all of these predictors are signatures of orders no greater than $K = 4$. Now, first randomly choose $q$ true predictors from all $\frac{d^{K+1}-1}{d-1} = 31$ signatures; next randomly set each beta coefficient of these true predictors from the standard normal distribution; next generate 200 samples from this true model with error term $\varepsilon_n$ drawn from a normal distribution with mean zero and standard error 0.0001. We divide the 200 samples into a training set and a test set, with 100 samples assigned to each. Then, we run a Lasso regression given by Equation (5) to select predictors based on the training set. The tuning parameter $\lambda$ is chosen using 5-fold cross-validation. Finally, we calculate the out-of-sample $R^2$ using the chosen $\lambda$ on the test set. We repeat the above procedure by 1,000 times and calculate the average out-of-sample $R^2$.

Figure A.10 shows the out-of-sample $R^2$ for different values of inter-dimensional correlation, $\rho$, and different numbers of true predictors, $q$. We can find that, first, Lasso exhibits lower out-of-sample $R^2$ when the absolute value of $\rho$ is small, i.e., when the inter-dimensional correlations of the Brownian motion are weak. Second, as the number of true predictors $q$ increases, the out-of-sample $R^2$ increases. Finally, Itô signatures have lower out-of-sample $R^2$ than those for Stratonovich signatures, holding other variables constant ($\rho$ and $q$).

All these findings are consistent with our analysis of sign consistency in the main paper. A higher consistency rate corresponds to a higher precision, a higher F1-score, and a lower out-of-sample $R^2$. This consistency across different metrics reinforces the applicability of our theoretical results when using alternative measures to evaluate the performance of Lasso in the context of signatures. It confirms that the theoretical insights derived from sign consistency extend to other evaluation metrics, demonstrating the robustness and broad applicability of our findings.

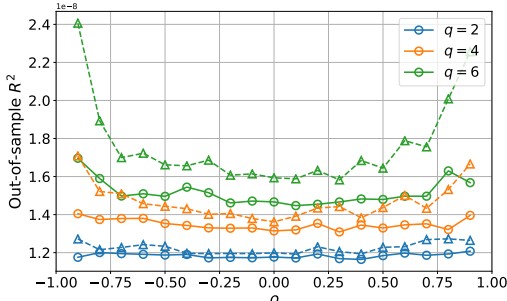

Figure A.10: Average out-of-sample $R^2$ for the Brownian motion with different values of inter-dimensional correlation, $\rho$, and different numbers of true predictors, $q$. Solid lines correspond to Itô signatures and dashed lines correspond to Stratonovich signatures.

## H  TIME AUGMENTATION

Time augmentation is a widely used technique in signature-based analysis, which involves adding a time dimension $t$ to the original time series, $\mathbf{X}_t$ (Chevyrev & Kormilitzin, 2016; Lyons & McLeod, 2022). In this section, we consider the time-augmented Brownian motion defined as follows:

**Definition A.6** (Time-augmented Brownian motion). $\hat{\mathbf{X}}$ *is a* $(d+1)$*-dimensional time-augmented Brownian motion if it can be expressed as:*

$$\hat{\mathbf{X}}_t = \left(t, \mathbf{X}_t^\top\right)^\top = \left(t, X_t^1, X_t^2, \ldots, X_t^d\right)^\top, \tag{A.10}$$

*where* $\mathbf{X}_t = (X_t^1, X_t^2, \ldots, X_t^d)^\top$ *is a* $d$*-dimensional Brownian motion given by Equation* (6) *in the main paper. For notational simplicity, let* $X_t^0 = t$*, then*

$$\hat{\mathbf{X}}_t = \left(X_t^0, X_t^1, X_t^2, \ldots, X_t^d\right)^\top.$$

Now we discuss the correlation structure of signatures and the consistency of signatures using Lasso for the time-augmented Brownian motion.

### H.1  CORRELATION STRUCTURE OF SIGNATURES FOR TIME-AUGMENTED BROWNIAN MOTION

The following proposition shows the moments of Itô signatures for the time-augmented Brownian motion. Note that an index of 0 corresponds to the time dimension, $X_t^0 = t$.

**Proposition A.3.** *Let* $\hat{\mathbf{X}}$ *be a* $(d+1)$*-dimensional time-augmented Brownian motion given by Equation* (A.10)*. For any* $l, t \geq 0$ *and* $m, n = 1, 2, \ldots,$ $f_{n,m}(l,t) := \mathbb{E}\left[S(\hat{\mathbf{X}})_l^{i_1,\ldots,i_n,I} S(\hat{\mathbf{X}})_t^{j_1,\ldots,j_m,I}\right]$ *can be calculated recursively by:*

$$f_{n,m}(l,t) = \begin{cases} \int_0^l \int_0^t f_{n-1,m-1}(s,\tau)\mathrm{d}\tau\mathrm{d}s, & \text{if } i_n = 0, j_m = 0, \\ \int_0^l f_{n-1,m}(s,t)\mathrm{d}s, & \text{if } i_n = 0, j_m \neq 0, \\ \int_0^t f_{n,m-1}(l,s)\mathrm{d}s, & \text{if } i_n \neq 0, j_m = 0, \\ \rho_{i_n j_m}\sigma_{i_n}\sigma_{j_m}\int_0^{l\wedge t} f_{n-1,m-1}(s,s)\mathrm{d}s, & \text{if } i_n \neq 0, j_m \neq 0, \end{cases}$$

*with initial conditions*

$$f_{0,m}(l,t) = \begin{cases} \frac{t^m}{m!} & \text{if } j_1 = j_2 = \cdots = j_m = 0, \\ 0, & \text{otherwise}, \end{cases}$$

$$f_{n,0}(l,t) = \begin{cases} \frac{t^n}{n!} & \text{if } i_1 = i_2 = \cdots = i_n = 0, \\ 0, & \text{otherwise}. \end{cases}$$

The following proposition shows the moments of Stratonovich signatures for the time-augmented Brownian motion.

**Proposition A.4.** *Let* $\hat{\mathbf{X}}$ *be a* $(d+1)$*-dimensional time-augmented Brownian motion given by Equation* (A.10). *For any* $l, t \geq 0$ *and* $m, n = 1, 2, \ldots$, $f_{n,m}(l,t) := \mathbb{E}\left[S(\hat{\mathbf{X}})_l^{i_1,\ldots,i_n,S} S(\hat{\mathbf{X}})_t^{j_1,\ldots,j_m,S}\right]$ *can be calculated recursively by:*

$$f_{n,m}(l,t) =$$
$$\begin{cases}
\int_0^l \int_0^t f_{n-1,m-1}(s,\tau)\mathrm{d}\tau\mathrm{d}s, & \text{if } i_n = 0, j_m = 0, \\
\int_0^l f_{n-1,m}(s,t)\mathrm{d}s, & \text{if } i_n = 0, j_m \neq 0, \\
\rho_{i_n j_m}\sigma_{i_n}\sigma_{j_m}\int_0^{l\wedge t} f_{n-1,m-1}(s,s)\mathrm{d}s, & \text{if } i_n \neq 0, j_m \neq 0, i_{n-1} = 0, j_{m-1} = 0, \\
\rho_{i_n j_m}\sigma_{i_n}\sigma_{j_m}\int_0^{l\wedge t} f_{n-1,m-1}(s,s)\mathrm{d}s \\
\quad + \frac{1}{2}\rho_{i_{n-1}i_n}\sigma_{i_{n-1}}\sigma_{i_n}\int_0^l g_{n-2,m}(s,t)\mathrm{d}s, & \text{if } i_n \neq 0, j_m \neq 0, i_{n-1} \neq 0, j_{m-1} = 0, \\
\rho_{i_n j_m}\sigma_{i_n}\sigma_{j_m}\int_0^{l\wedge t} f_{n-1,m-1}(s,s)\mathrm{d}s \\
\quad + \frac{1}{2}\rho_{i_{n-1}i_n}\sigma_{i_{n-1}}\sigma_{i_n}\int_0^l g_{n-2,m}(s,t)\mathrm{d}s \\
\quad + \frac{1}{2}\rho_{j_{m-1}j_m}\sigma_{j_{m-1}}\sigma_{j_m}\int_0^t \tilde{g}_{m-2,n}(s,l)\mathrm{d}s \\
\quad + \frac{1}{4}\rho_{i_{n-1}i_n}\sigma_{i_{n-1}}\sigma_{i_n}\rho_{j_{m-1}j_m}\sigma_{j_{m-1}}\sigma_{j_m} \\
\qquad\qquad \cdot \int_0^l \int_0^t f_{n-2,m-2}(s,\tau)\mathrm{d}\tau\mathrm{d}s, & \text{if } i_n \neq 0, j_m \neq 0, i_{n-1} \neq 0, j_{m-1} \neq 0,
\end{cases}$$

$$g_{n,m}(l,t) =$$
$$\begin{cases}
\int_0^l g_{n-1,m}(s,t)\mathrm{d}s, & \text{if } i_n = 0, \\
\rho_{i_n j_m}\sigma_{i_n}\sigma_{j_m}\int_0^{l\wedge t} f_{n-1,m-1}(s,s)\mathrm{d}s + \frac{1}{2}\rho_{i_{n-1}i_n}\sigma_{i_{n-1}}\sigma_{i_n}\int_0^l g_{n-2,m}(s,t)\mathrm{d}s, & \text{if } i_n \neq 0,
\end{cases}$$

$$\tilde{g}_{m,n}(t,l) =$$
$$\begin{cases}
\int_0^t \tilde{g}_{m-1,n}(s,l)\mathrm{d}s, & \text{if } j_m = 0, \\
\rho_{i_n j_m}\sigma_{i_n}\sigma_{j_m}\int_0^{l\wedge t} f_{n-1,m-1}(s,s)\mathrm{d}s + \frac{1}{2}\rho_{j_{m-1}j_m}\sigma_{j_{m-1}}\sigma_{j_m}\int_0^t \tilde{g}_{m-2,n}(s,l)\mathrm{d}s, & \text{if } j_m \neq 0,
\end{cases}$$

*with initial conditions*

$$f_{0,0}(l,t) = 1,$$
$$g_{0,m}(l,t) = \begin{cases} \int_0^t f_{0,m-1}(l,s)\mathrm{d}s, & \text{if } j_m = 0, \\ 0, & \text{if } j_m \neq 0, \end{cases}$$
$$\tilde{g}_{0,n}(l,t) = \begin{cases} \int_0^l f_{n-1,0}(s,t)\mathrm{d}s, & \text{if } i_n = 0, \\ 0, & \text{if } i_n \neq 0. \end{cases}$$

The following example shows the correlation structures of Itô and Stratonovich signatures for a (1+1)-dimensional time-augmented Brownian motion $\hat{\mathbf{X}}$.

**Example A.6.** *Consider a (1+1)-dimensional time-augmented Brownian motion $\hat{X}$. Figures A.11 and A.12 show the correlation structures of Itô and Stratonovich signatures of $\hat{X}$, respectively, calculated through simulations. In particular, we first simulate 10,000 paths of $\hat{X}$, then calculate the signatures of each path, and finally calculate the sample correlation matrix of the signatures.*

*From the figures, we can observe that the correlation matrix of the time-augmented Brownian motion does not exhibit the same special structures (block diagonal or odd–even alternating) we observe in the main paper. In addition, the correlation between Stratonovich signatures is generally stronger than that between Itô signatures.*

*By comparing Figures A.11 and A.12 with Figures A.3 and A.5 in the main paper, we can find that the time augmentation generally increases the correlation between signatures. Therefore, we expect that the time augmentation will lead to a lower consistency rate for Lasso.*

### H.2 CONSISTENCY OF SIGNATURES FOR TIME-AUGMENTED BROWNIAN MOTION

We conduct simulations to study the consistency of signatures using Lasso for the time-augmented Brownian motion.

Figure A.11: Correlation matrix of Itô signatures with orders truncated to 4 for a (1+1)-dimensional time-augmented Brownian motion $\hat{\mathbf{X}}$.

Figure A.12: Correlation matrix of Stratonovich signatures with orders truncated to 4 for a (1+1)-dimensional time-augmented Brownian motion $\hat{\mathbf{X}}$.

Consider a (2+1)-dimensional time-augmented Brownian motion $\hat{\mathbf{X}}$, which is the time-augmented version of a 2-dimensional Brownian motion $\mathbf{X}$. The inter-dimensional correlation of $\mathbf{X}$ is $\rho$. We perform the same experiment as conducted in Section 4.1 for $\hat{\mathbf{X}}$.

Figure A.13 shows the consistency rates for different values of inter-dimensional correlation, $\rho$, and different numbers of true predictors, $q$. The observations are similar to the results for Brownian motion without time augmentation (Figure 1a). In addition, the consistency rates are generally lower when there exists time augmentation (Figure A.13) compared to the case without time augmentation (Figure 1a). This is consistent with Example A.6 because time augmentation tends to increase the correlation between signatures, which results in a lower consistency rate.

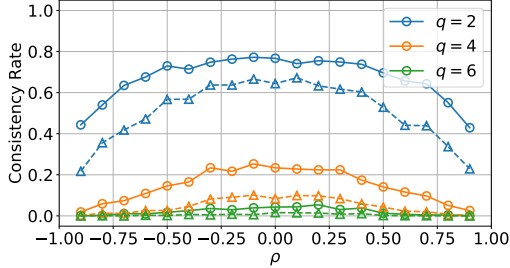

Figure A.13: Consistency rates for the time-augmented Brownian motion with different values of inter-dimensional correlation, $\rho$, and different numbers of true predictors, $q$. Solid lines correspond to Itô signatures and dashed lines correspond to Stratonovich signatures.

In summary, our simulation shows that time augmentation lowers the consistency rate of Lasso.

## I  IMPACT OF THE DIMENSION OF THE PROCESS AND THE NUMBER OF SAMPLES

All simulations in our main paper consider the case of $d = 2$ (dimension of the process) and $N = 100$ (number of samples) to confirm the theoretical results. The choice of $d = 2$ is consistent with the simulation setup commonly used in the literature on signatures; see, for example, Chevyrev & Kormilitzin (2016). In this appendix, we examine how the consistency of Lasso varies with the dimension of the process $d$ and the number of samples $N$.

Figure A.14 shows how the consistency of Lasso varies with $d$. Figure A.14a shows the results for the Brownian motion, and Figure A.14b for the OU process with $\kappa = 2$. We set the number of true predictors to be three. Other simulation setups remain the same as in the main paper.

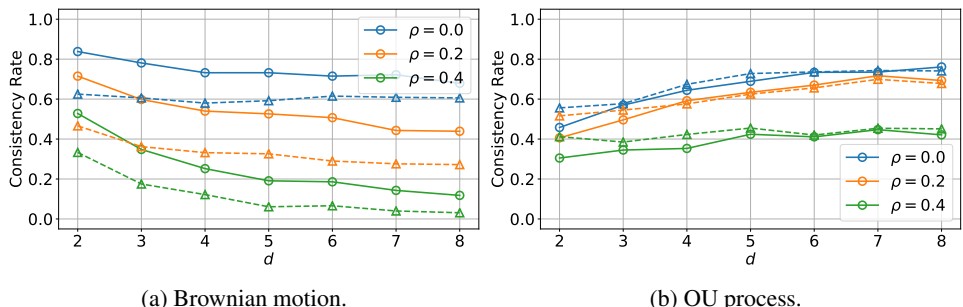

(a) Brownian motion.
(b) OU process.

Figure A.14: Consistency rates for the Brownian motion and the random walk with different numbers of dimensions, $d$, and different values of inter-dimensional correlation, $\rho$. Solid lines correspond to Itô signatures and dashed lines correspond to Stratonovich signatures.

First, for each value of $d$, the conclusions in our main paper remain valid (e.g., for the Brownian motion, the consistency rate for Itô signatures is higher than Stratonovich signatures). Therefore, it is sufficient to consider the case of $d = 2$ in our main paper. Second, for the Brownian motion, we

observe that the consistency rate decreases with $d$. This can be attributed to the fact that the inter-dimensional correlation of the process leads to stronger correlations between signatures. Third, for the OU process, the consistency rate increases with $d$ because the inter-dimensional correlation of the process is weaker than the correlation between the increments of the OU process itself.

Figure A.15 shows the relationship between the consistency rate and the number of samples. In general, we find that the consistency rate increases as the number of samples increases.

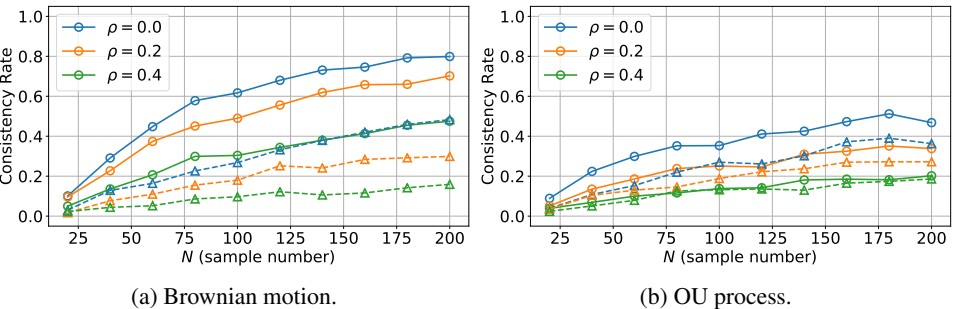

(a) Brownian motion.  (b) OU process.

Figure A.15: Consistency rates for the Brownian motion and the OU process with different numbers of samples, $N$, and different values of inter-dimensional correlation, $\rho$. Solid lines correspond to Itô signatures and dashed lines correspond to Stratonovich signatures.

## J  THE ARIMA PROCESS

This appendix examines the consistency of signatures for a more complex model—the ARIMA$(p, I, q)$ model, where $p$ is the lag of AR, $I$ is the degree of differencing, and $q$ is the lag of MA.

Figure A.16 shows how the consistency rate varies with $p$, $q$, and $I$. We find that the consistency rate does not exhibit any apparent dependence on $p$ and $q$, but does highly rely on $I$. Specifically, the consistency rate generally decreases as $I$ increases due to the stronger correlation between the increments of the ARIMA processes introduced by $I$.

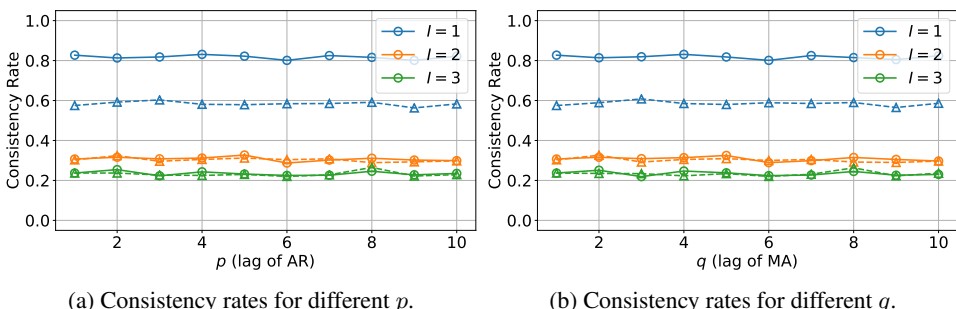

(a) Consistency rates for different $p$.  (b) Consistency rates for different $q$.

Figure A.16: Consistency rates for the ARIMA$(p, I, q)$ with different lags of AR, $p$, lags of MA, $q$, and degrees of differencing, $I$. Solid lines correspond to Itô signatures and dashed lines correspond to Stratonovich signatures.

## K  PROOFS

This appendix provides the proofs of all theoretical results in this paper.

*Proof of Proposition 1.* For the expectation, we have

$$\mathbb{E}\left[S(\mathbf{X})_t^{i_1,\ldots,i_n,I}\right] = \mathbb{E}\left[\int_0^t S(\mathbf{X})_s^{i_1,\ldots,i_{n-1}}\mathrm{d}X_s^{i_n}\right] = 0 \tag{A.11}$$

because the expectation of an Itô integral is zero.

Next we prove $\mathbb{E}\left[S(\mathbf{X})_t^{i_1,\ldots,i_n,I} S(\mathbf{X})_t^{j_1,\ldots,j_m,I}\right] = 0$ for $m \neq n$ by induction. Without loss of generality, we assume that $m > n$. When $n = 1$, for any $m > 1$, we have

$$
\begin{aligned}
\mathbb{E}\left[S(\mathbf{X})_t^{i_1,I} S(\mathbf{X})_t^{j_1,\ldots,j_m,I}\right] &= \mathbb{E}\left[\left(\int_0^t \mathrm{d}X_s^{i_1}\right)\left(\int_0^t S(\mathbf{X})_s^{j_1,\ldots,j_{m-1}}\mathrm{d}X_s^{j_m}\right)\right] \\
&= \int_0^t \mathbb{E}\left[S(\mathbf{X})_s^{j_1,\ldots,j_{m-1},I}\right]\rho_{i_1 j_m}\sigma_{i_1}\sigma_{j_m}\mathrm{d}s = 0,
\end{aligned}
$$

where the second equality uses the Itô isometry and the third equality uses Equation (A.11). Now assume that for $m > n$, we have $\mathbb{E}\left[S(\mathbf{X})_t^{i_1,\ldots,i_n,I} S(\mathbf{X})_t^{j_1,\ldots,j_m,I}\right] = 0$. Then,

$$
\begin{aligned}
&\mathbb{E}\left[S(\mathbf{X})_t^{i_1,\ldots,i_{n+1},I} S(\mathbf{X})_t^{j_1,\ldots,j_{m+1},I}\right] \\
=&\mathbb{E}\left[\left(\int_0^t S(\mathbf{X})_t^{i_1,\ldots,i_n,I}\mathrm{d}X_s^{i_{n+1}}\right)\left(\int_0^t S(\mathbf{X})_s^{j_1,\ldots,j_m,I}\mathrm{d}X_s^{j_{m+1}}\right)\right] \\
=&\int_0^t \mathbb{E}\left[S(\mathbf{X})_t^{i_1,\ldots,i_k,I} S(\mathbf{X})_s^{j_1,\ldots,j_m,I}\right]\rho_{i_{n+1}j_{m+1}}\sigma_{i_{n+1}}\sigma_{j_{m+1}}\mathrm{d}s = 0.
\end{aligned}
$$

This proves $\mathbb{E}\left[S(\mathbf{X})_t^{i_1,\ldots,i_n,I} S(\mathbf{X})_t^{j_1,\ldots,j_m,I}\right] = 0$.

We finally prove $\mathbb{E}\left[S(\mathbf{X})_t^{i_1,\ldots,i_n,I} S(\mathbf{X})_t^{j_1,\ldots,j_n,I}\right] = \frac{t^n}{n!}\prod_{k=1}^n \rho_{i_k j_k}\sigma_{i_k}\sigma_{j_k}$ by induction. When $n = 1$, we have

$$
\begin{aligned}
&\mathbb{E}\left[S(\mathbf{X})_t^{i_1,I} S(\mathbf{X})_t^{j_1,I}\right] \\
=&\mathbb{E}\left[\left(\int_0^t \mathrm{d}X_s^{i_1}\right)\left(\int_0^t \mathrm{d}X_s^{j_1}\right)\right] = \int_0^t \rho_{i_1 j_1}\sigma_{i_1}\sigma_{j_1}\mathrm{d}s = t\rho_{i_1 j_1}\sigma_{i_1}\sigma_{j_1}.
\end{aligned}
$$

Then, assume that $\mathbb{E}\left[S(\mathbf{X})_t^{i_1,\ldots,i_n,I} S(\mathbf{X})_t^{j_1,\ldots,j_n,I}\right] = \frac{t^n}{n!}\prod_{k=1}^n \rho_{i_k j_k}\sigma_{i_k}\sigma_{j_k}$. We have

$$
\begin{aligned}
&\mathbb{E}\left[S(\mathbf{X})_t^{i_1,\ldots,i_{n+1},I} S(\mathbf{X})_t^{j_1,\ldots,j_{n+1},I}\right] \\
=&\mathbb{E}\left[\left(\int_0^t S(\mathbf{X})_s^{i_1,\ldots,i_n,I}\mathrm{d}X_s^{i_{n+1}}\right)\left(\int_0^t S(\mathbf{X})_s^{j_1,\ldots,j_n,I}\mathrm{d}X_s^{j_{n+1}}\right)\right] \\
=&\int_0^t \mathbb{E}\left[S(\mathbf{X})_s^{i_1,\ldots,i_n,I} S(\mathbf{X})_s^{j_1,\ldots,j_n,I}\right]\rho_{i_{n+1}j_{n+1}}\sigma_{i_{n+1}}\sigma_{j_{n+1}}\mathrm{d}s \\
=&\int_0^t \left(\frac{s^n}{n!}\prod_{k=1}^n \rho_{i_k j_k}\sigma_{i_k}\sigma_{j_k}\right)\rho_{i_{n+1}j_{n+1}}\sigma_{i_{n+1}}\sigma_{j_{n+1}}\mathrm{d}s = \frac{t^{n+1}}{(n+1)!}\prod_{k=1}^{n+1} \rho_{i_k j_k}\sigma_{i_k}\sigma_{j_k}.
\end{aligned}
$$

Therefore, $\mathbb{E}\left[S(\mathbf{X})_t^{i_1,\ldots,i_n,I} S(\mathbf{X})_t^{j_1,\ldots,j_n,I}\right] = \frac{t^n}{n!}\prod_{k=1}^n \rho_{i_k j_k}\sigma_{i_k}\sigma_{j_k}$. This completes the proof. $\square$

*Proof of Proposition 2.* By Proposition 1, for any $n$, we have

$$
\frac{\mathbb{E}\left[S(\mathbf{X})_t^{i_1,\ldots,i_n,I} S(\mathbf{X})_t^{j_1,\ldots,j_n,I}\right]}{\sqrt{\mathbb{E}\left[S(\mathbf{X})_t^{i_1,\ldots,i_n,I}\right]^2 \mathbb{E}\left[S(\mathbf{X})_t^{j_1,\ldots,j_n,I}\right]^2}} = \frac{\frac{t^n}{n!}\prod_{k=1}^n \rho_{i_k j_k}\sigma_{i_k}\sigma_{j_k}}{\sqrt{\frac{t^n}{n!}\prod_{k=1}^n \sigma_{i_k}\sigma_{i_k} \cdot \frac{t^n}{n!}\prod_{k=1}^n \sigma_{j_k}\sigma_{j_k}}} = \prod_{k=1}^n \rho_{i_k j_k},
$$

implying that

$$
\frac{\mathbb{E}\left[S(\mathbf{X})_t^{i_1,I} S(\mathbf{X})_t^{j_1,I}\right]}{\sqrt{\mathbb{E}\left[S(\mathbf{X})_t^{i_1,I}\right]^2 \mathbb{E}\left[S(\mathbf{X})_t^{j_1,I}\right]^2}} = \rho_{i_1 j_1}
$$

and

$$\frac{\mathbb{E}\left[S(\mathbf{X})_t^{i_1,\ldots,i_n,I}S(\mathbf{X})_t^{j_1,\ldots,j_n,I}\right]}{\sqrt{\mathbb{E}\left[S(\mathbf{X})_t^{i_1,\ldots,i_n,I}\right]^2\mathbb{E}\left[S(\mathbf{X})_t^{j_1,\ldots,j_n,I}\right]^2}} = \rho_{i_nj_n}\cdot\frac{\mathbb{E}\left[S(\mathbf{X})_t^{i_1,\ldots,i_{n-1},I}S(\mathbf{X})_t^{j_1,\ldots,j_{n-1},I}\right]}{\sqrt{\mathbb{E}\left[S(\mathbf{X})_t^{i_1,\ldots,i_{n-1},I}\right]^2\mathbb{E}\left[S(\mathbf{X})_t^{j_1,\ldots,j_{n-1},I}\right]^2}}.$$

This proves the Kronecker product structure given by Equation (8).

Proposition 1 also implies that, for any $m \neq n$,

$$\frac{\mathbb{E}\left[S(\mathbf{X})_t^{i_1,\ldots,i_n,I}S(\mathbf{X})_t^{j_1,\ldots,j_m,I}\right]}{\sqrt{\mathbb{E}\left[S(\mathbf{X})_t^{i_1,\ldots,i_n,I}\right]^2\mathbb{E}\left[S(\mathbf{X})_t^{j_1,\ldots,j_m,I}\right]^2}} = 0.$$

This proves that Itô signatures with different orders are uncorrelated and, therefore, the correlation matrix is block diagonal. This completes the proof. □

*Proof of Proposition 3.* We omit the proof of Equations

$$\mathbb{E}\left[S(\mathbf{X})_t^{i_1,\ldots,i_{2n-1},S}\right] = 0$$

and

$$\mathbb{E}\left[S(\mathbf{X})_t^{i_1,\ldots,i_{2n},S}S(\mathbf{X})_t^{j_1,\ldots,j_{2m-1},S}\right] = 0$$

because they can be proven using a similar approach to the proof Proposition 5. Equation

$$\mathbb{E}\left[S(\mathbf{X})_t^{i_1,\ldots,i_{2n},S}\right] = \frac{1}{2^n}\frac{t^n}{n!}\prod_{k=1}^{n}\rho_{i_{2k-1}i_{2k}}\prod_{k=1}^{2n}\sigma_{i_k}$$

is a corollary of Proposition A.1. □

*Proof of Proposition 4.* This is a direct corollary of Proposition 3. □

*Proof of Proposition 5.* We only need to prove that, for an odd number $m$ and an even number $n$, we have

$$\mathbb{E}\left[S(\mathbf{X})_t^{i_1,\ldots,i_m}S(\mathbf{X})_t^{j_1,\ldots,j_n}\right] = 0$$

for any $i_1,\ldots,i_m$ and $j_1,\ldots,j_n$ taking values in $\{1,2,\ldots,d\}$. Here the signatures can be defined in the sense of either Itô or Stratonovich.

Consider the reflected OU process, $\tilde{\mathbf{X}}_t = -\mathbf{X}_t$. By definition, $\tilde{\mathbf{X}}_t$ is also an OU process with the same mean reversion parameter. Therefore, the signatures of $\tilde{\mathbf{X}}_t$ and $\mathbf{X}_t$ should have the same distribution. In particular, we have

$$\mathbb{E}\left[S(\tilde{\mathbf{X}})_t^{i_1,\ldots,i_m}S(\tilde{\mathbf{X}})_t^{j_1,\ldots,j_n}\right] = \mathbb{E}\left[S(\mathbf{X})_t^{i_1,\ldots,i_m}S(\mathbf{X})_t^{j_1,\ldots,j_n}\right]. \tag{A.12}$$

Now we consider the definition of the signatures:

$$S(\mathbf{X})_t^{i_1,\ldots,i_m} = \int_{0<t_1<\cdots<t_m<t}\mathrm{d}X_{t_1}^{i_1}\cdots\mathrm{d}X_{t_k}^{i_m},$$

where the integral can be defined in the sense of either Itô or Stratonovich. We therefore have

$$S(\tilde{\mathbf{X}})_t^{i_1,\ldots,i_m} = S(-\mathbf{X})_t^{i_1,\ldots,i_m} = \int_{0<t_1<\cdots<t_m<t}\mathrm{d}(-X_{t_1}^{i_1})\cdots\mathrm{d}(-X_{t_k}^{i_m})$$

$$= (-1)^m\int_{0<t_1<\cdots<t_m<t}\mathrm{d}X_{t_1}^{i_1}\cdots\mathrm{d}X_{t_k}^{i_m} = (-1)^m S(\mathbf{X})_t^{i_1,\ldots,i_m}.$$

Similarly, we have

$$S(\tilde{\mathbf{X}})_t^{j_1,\ldots,j_n} = (-1)^n S(\mathbf{X})_t^{j_1,\ldots,j_n}.$$

Therefore,

$$\mathbb{E}\left[S(\tilde{\mathbf{X}})_t^{i_1,\ldots,i_m} S(\tilde{\mathbf{X}})_t^{j_1,\ldots,j_n}\right] = (-1)^{m+n}\mathbb{E}\left[S(\mathbf{X})_t^{i_1,\ldots,i_m} S(\mathbf{X})_t^{j_1,\ldots,j_n}\right]$$

$$= -\mathbb{E}\left[S(\mathbf{X})_t^{i_1,\ldots,i_m} S(\mathbf{X})_t^{j_1,\ldots,j_n}\right],$$

and combining this with Equation (A.12) leads to the result. $\qquad\square$

*Proof of Proposition 6.* This result holds because of Proposition 2 and Zhao & Yu (2006, Corollary 5). $\qquad\square$

*Proof of Proposition 7.* This result holds because of Proposition 2, Zhao & Yu (2006, Corollary 2), and Zhao & Yu (2006, Corollary 5). $\qquad\square$

*Proof of Proposition 8.* This result holds because of Proposition 4 and Zhao & Yu (2006, Corollary 5). $\qquad\square$

*Proof of Proposition A.1.* By the relationship between the Stratonovich integral and the Itô integral, we have

$$S(\mathbf{X})_t^{j_1,\ldots,j_m,S} = \int_0^t S(\mathbf{X})_s^{j_1,\ldots,j_{m-1},S} \circ \mathrm{d}X_s^{j_m}$$

$$= \int_0^t S(\mathbf{X})_s^{j_1,\ldots,j_{m-1},S}\mathrm{d}X_s^{j_m} + \frac{1}{2}\left[S(\mathbf{X})^{j_1,\ldots,j_{m-1},S}, X^{j_m}\right]_t,$$

where $[A, B]_t$ represents the quadratic covariation between processes $A$ and $B$ from time 0 to $t$. Furthermore, by properties of the quadratic covariation,

$$\left[S(\mathbf{X})^{j_1,\ldots,j_{m-1},S}, X^{j_m}\right]_t = \int_0^t S(\mathbf{X})_s^{j_1,\ldots,j_{m-2},S}\mathrm{d}\left[X^{j_{m-1}}, X^{j_m}\right]_s$$

$$= \rho_{j_{m-1}j_m}\sigma_{j_{m-1}}\sigma_{j_m}\int_0^t S(\mathbf{X})_s^{j_1,\ldots,j_{m-2},S}\mathrm{d}s.$$

Therefore,

$$S(\mathbf{X})_t^{j_1,\ldots,j_m,S} = \int_0^t S(\mathbf{X})_s^{j_1,\ldots,j_{m-1},S}\mathrm{d}X_s^{j_m} + \frac{1}{2}\rho_{j_{m-1}j_m}\sigma_{j_{m-1}}\sigma_{j_m}\int_0^t S(\mathbf{X})_s^{j_1,\ldots,j_{m-2},S}\mathrm{d}s.$$

For any $l, t \geq 0$ and $m, n = 0, 1, \ldots$, define

$$f_{n,m}(l,t) := \mathbb{E}\left[S(\mathbf{X})_l^{i_1,\ldots,i_n,S} S(\mathbf{X})_t^{j_1,\ldots,j_m,S}\right],$$

$$g_{n,m}(l,t) := \mathbb{E}\left[S(\mathbf{X})_l^{i_1,\ldots,i_n,S} \int_0^t S(\mathbf{X})_s^{j_1,\ldots,j_{m-1},S}\mathrm{d}X_s^{j_m}\right].$$

Then, by Fubini's theorem,

$$f_{n,m}(l,t) = \mathbb{E}\left[S(\mathbf{X})_l^{i_1,\ldots,i_n,S} S(\mathbf{X})_t^{j_1,\ldots,j_m,S}\right]$$

$$=\mathbb{E}\left[S(\mathbf{X})_l^{i_1,\ldots,i_n,S}\left(\int_0^t S(\mathbf{X})_s^{j_1,\ldots,j_{m-1},S}\mathrm{d}X_s^{j_m} + \frac{1}{2}\rho_{j_{m-1}j_m}\sigma_{j_{m-1}}\sigma_{j_m}\int_0^t S(\mathbf{X})_s^{j_1,\ldots,j_{m-2},S}\mathrm{d}s\right)\right]$$

$$=g_{n,m}(l,t) + \frac{1}{2}\rho_{j_{m-1}j_m}\sigma_{j_{m-1}}\sigma_{j_m}\mathbb{E}\left[S(\mathbf{X})_l^{i_1,\ldots,i_n,S}\int_0^t S(\mathbf{X})_s^{j_1,\ldots,j_{m-2},S}\mathrm{d}s\right]$$

$$=g_{n,m}(l,t) + \frac{1}{2}\rho_{j_{m-1}j_m}\sigma_{j_{m-1}}\sigma_{j_m}\int_0^t \mathbb{E}\left[S(\mathbf{X})_l^{i_1,\ldots,i_n,S} S(\mathbf{X})_s^{j_1,\ldots,j_{m-2},S}\right]\mathrm{d}s$$

$$=g_{n,m}(l,t) + \frac{1}{2}\rho_{j_{m-1}j_m}\sigma_{j_{m-1}}\sigma_{j_m}\int_0^t f_{n,m-2}(l,s)\mathrm{d}s.$$

This proves Equations (A.1) and (A.5). In addition, by Itô isometry and Fubini's theorem,

$$g_{n,m}(l,t) = \mathbb{E}\left[ S(\mathbf{X})_l^{i_1,\dots,i_n,S} \int_0^t S(\mathbf{X})_s^{j_1,\dots,j_{m-1},S} \mathrm{d}X_s^{j_m} \right]$$

$$=\mathbb{E}\left[ \left( \int_0^l S(\mathbf{X})_s^{i_1,\dots,i_{n-1},S} \mathrm{d}X_s^{i_n} + \frac{1}{2}\rho_{i_{n-1}i_n}\sigma_{i_{n-1}}\sigma_{i_n} \int_0^l S(\mathbf{X})_s^{i_1,\dots,i_{n-2},S} \mathrm{d}s \right) \right.$$
$$\left. \cdot \int_0^t S(\mathbf{X})_s^{j_1,\dots,j_{m-1},S} \mathrm{d}X_s^{j_m} \right]$$

$$=\mathbb{E}\left[ \int_0^l S(\mathbf{X})_s^{i_1,\dots,i_{n-1},S} \mathrm{d}X_s^{i_n} \int_0^t S(\mathbf{X})_s^{j_1,\dots,j_{m-1},S} \mathrm{d}X_s^{j_m} \right]$$
$$+ \frac{1}{2}\rho_{i_{n-1}i_n}\sigma_{i_{n-1}}\sigma_{i_n} \mathbb{E}\left[ \int_0^l S(\mathbf{X})_s^{i_1,\dots,i_{n-2},S} \mathrm{d}s \int_0^t S(\mathbf{X})_s^{j_1,\dots,j_{m-1},S} \mathrm{d}X_s^{j_m} \right]$$

$$=\rho_{i_n j_m}\sigma_{i_n}\sigma_{j_m} \int_0^{l\wedge t} \mathbb{E}\left[ S(\mathbf{X})_s^{i_1,\dots,i_{n-1},S} S(\mathbf{X})_s^{j_1,\dots,j_{m-1},S} \right] \mathrm{d}s$$
$$+ \frac{1}{2}\rho_{i_{n-1}i_n}\sigma_{i_{n-1}}\sigma_{i_n} \int_0^l \mathbb{E}\left[ S(\mathbf{X})_s^{i_1,\dots,i_{n-2},S} \int_0^t S(\mathbf{X})_u^{j_1,\dots,j_{m-1},S} \mathrm{d}X_u^{j_m} \right] \mathrm{d}s$$

$$=\rho_{i_n j_m}\sigma_{i_n}\sigma_{j_m} \int_0^{l\wedge t} f_{n-1,m-1}(s,s)\mathrm{d}s + \frac{1}{2}\rho_{i_{n-1}i_n}\sigma_{i_{n-1}}\sigma_{i_n} \int_0^l g_{n-2,m}(s,t)\mathrm{d}s.$$

This proves Equations (A.2) and (A.6).

Now we prove the initial conditions. First, by the definition of 0-th order signatures, $f_{0,0}(l,t) = \mathbb{E}[S(\mathbf{X})_l^0 S(\mathbf{X})_t^0] = 1$, which proves Equation (A.3). Second,

$$g_{0,2m}(l,t) = \mathbb{E}\left[ \int_0^t S(\mathbf{X})_s^{j_1,\dots,j_{2m-1},S} \mathrm{d}X_s^{j_{2m}} \right] = 0$$

because the expectation of an Itô integral is zero, which proves Equation (A.4). Third,

$$f_{1,1}(l,t) = \mathbb{E}\left[ S(\mathbf{X})_l^{i_1,S} S(\mathbf{X})_t^{j_1,S} \right] = \mathbb{E}\left[ \int_0^l 1\circ \mathrm{d}X_s^{i_1} \int_0^t 1\circ \mathrm{d}X_s^{j_1} \right]$$
$$= \mathbb{E}\left[ X_l^{i_1} X_t^{j_1} \right] = \rho_{i_1 j_1}\sigma_{i_1}\sigma_{j_1}(l\wedge t),$$

which proves Equation (A.7). Fourth, by Itô isometry,

$$g_{1,2m-1}(l,t) = \mathbb{E}\left[ S(\mathbf{X})_l^{i_1,S} \int_0^t S(\mathbf{X})_s^{j_1,\dots,j_{2m-2},S} \mathrm{d}X_s^{j_{2m-1}} \right]$$
$$= \mathbb{E}\left[ \int_0^l 1\circ \mathrm{d}X_s^{i_1} \int_0^t S(\mathbf{X})_s^{j_1,\dots,j_{2m-2},S} \mathrm{d}X_s^{j_{2m-1}} \right]$$
$$= \mathbb{E}\left[ \int_0^l \mathrm{d}X_s^{i_1} \int_0^t S(\mathbf{X})_s^{j_1,\dots,j_{2m-2},S} \mathrm{d}X_s^{j_{2m-1}} \right]$$
$$= \int_0^{l\wedge t} \mathbb{E}\left[ S(\mathbf{X})_s^{j_1,\dots,j_{2m-2},S} \right] \rho_{i_1 j_{2m-1}}\sigma_{i_1}\sigma_{j_{2m-1}} \mathrm{d}s$$
$$= \rho_{i_1 j_{2m-1}}\sigma_{i_1}\sigma_{j_{2m-1}} \int_0^{l\wedge t} f_{0,2m-2}(s,s)\mathrm{d}s.$$

In addition, by using Equation (A.1) recursively, we can obtain that

$$f_{0,2m-2}(s,s) = \frac{1}{2^{m-1}} \frac{s^{m-1}}{(m-1)!} \prod_{k=1}^{m-1} \rho_{j_{2k-1}j_{2k}} \prod_{k=1}^{2m-2} \sigma_{j_k}.$$

Therefore,

$$g_{1,2m-1}(l,t) = \rho_{i_1 j_{2m-1}} \frac{1}{2^{m-1}} \frac{(l \wedge t)^{m-1}}{(m-1)!} \sigma_{i_1} \prod_{k=1}^{2m-1} \sigma_{j_k} \prod_{k=1}^{m-1} \rho_{j_{2k-1} j_{2k}},$$

which proves Equation (A.8). This completes the proof. $\square$

*Proof of Proposition A.3.* When $i_n = 0$ and $j_m = 0$, by Fubini's theorem,

$$\begin{aligned}
f_{n,m}(l,t) &= \mathbb{E}\left[\int_0^l S(\hat{\mathbf{X}})_s^{i_1,\ldots,i_{n-1},I} \mathrm{d}s \int_0^t S(\hat{\mathbf{X}})_\tau^{j_1,\ldots,j_{m-1},I} \mathrm{d}\tau\right] \\
&= \mathbb{E}\left[\int_0^l \int_0^t S(\hat{\mathbf{X}})_s^{i_1,\ldots,i_{n-1},I} S(\hat{\mathbf{X}})_\tau^{j_1,\ldots,j_{m-1},I} \mathrm{d}\tau \mathrm{d}s\right] = \int_0^l \int_0^t f_{n-1,m-1}(s,\tau) \mathrm{d}\tau \mathrm{d}s.
\end{aligned}$$

When $i_n = 0$ and $j_m \neq 0$, by Fubini's theorem,

$$\begin{aligned}
f_{n,m}(l,t) &= \mathbb{E}\left[\int_0^l S(\hat{\mathbf{X}})_s^{i_1,\ldots,i_{n-1},I} \mathrm{d}s \cdot S(\hat{\mathbf{X}})_t^{j_1,\ldots,j_m,I}\right] \\
&= \mathbb{E}\left[\int_0^l S(\hat{\mathbf{X}})_s^{i_1,\ldots,i_{n-1},I} S(\hat{\mathbf{X}})_t^{j_1,\ldots,j_m,I} \mathrm{d}s\right] = \int_0^l f_{n-1,m}(s,t) \mathrm{d}s.
\end{aligned}$$

When $i_n \neq 0$ and $j_m = 0$, we can similarly obtain

$$f_{n,m}(l,t) = \int_0^t f_{n,m-1}(l,s) \mathrm{d}s.$$

When $i_n \neq 0$ and $j_m \neq 0$, by Itô isometry,

$$\begin{aligned}
f_{n,m}(l,t) &= \mathbb{E}\left[\int_0^l S(\hat{\mathbf{X}})_s^{i_1,\ldots,i_{n-1},I} \mathrm{d}X_s^{i_n} \int_0^t S(\hat{\mathbf{X}})_s^{j_1,\ldots,j_{m-1},I} \mathrm{d}X_s^{j_m}\right] \\
&= \int_0^{l \wedge t} \mathbb{E}\left[S(\hat{\mathbf{X}})_s^{i_1,\ldots,i_{n-1},I} S(\hat{\mathbf{X}})_s^{j_1,\ldots,j_{m-1},I}\right] \rho_{i_n j_m} \sigma_{i_n} \sigma_{j_m} \mathrm{d}s \\
&= \rho_{i_n j_m} \sigma_{i_n} \sigma_{j_m} \int_0^{l \wedge t} \int_0^t f_{n-1,m-1}(s,s) \mathrm{d}s.
\end{aligned}$$

The initial conditions can be easily verified using Proposition 1. This completes the proof. $\square$

*Proof of Proposition A.4.* We omit the proof because one can easily use approaches similar to the proofs of Propositions 3 and A.3 to obtain the result by letting

$$\begin{aligned}
f_{n,m}(l,t) &:= \mathbb{E}\left[S(\hat{\mathbf{X}})_l^{i_1,\ldots,i_n,S} S(\hat{\mathbf{X}})_t^{j_1,\ldots,j_m,S}\right], \\
g_{n,m}(l,t) &:= \mathbb{E}\left[S(\hat{\mathbf{X}})_l^{i_1,\ldots,i_n,S} \int_0^t S(\hat{\mathbf{X}})_u^{j_1,\ldots,j_{m-1},S} \mathrm{d}X_u^{j_m}\right], \\
\tilde{g}_{m,n}(t,l) &:= \mathbb{E}\left[S(\hat{\mathbf{X}})_t^{j_1,\ldots,j_m,S} \int_0^l S(\hat{\mathbf{X}})_u^{i_1,\ldots,i_{n-1},S} \mathrm{d}X_u^{i_n}\right].
\end{aligned}$$

$\square$

*Proof of Example A.5.* The solution to stochastic differential equation (A.9) can be explicitly expressed as

$$Y_t = \int_0^t e^{-\kappa(t-s)} \mathrm{d}W_s, \quad t \geq 0,$$

where $W_t$ is a standard Brownian motion. Therefore, by Itô isometry, $Y_t$ is a Gaussian random variable with mean 0 and variance

$$\mathrm{Var}(Y_t) = \mathbb{E}\left[Y_t^2\right] = \mathbb{E}\left[\int_0^t e^{-\kappa(t-s)} \mathrm{d}W_s\right]^2 = \int_0^t \left[e^{-\kappa(t-s)}\right]^2 \mathrm{d}s = \frac{1 - e^{-2\kappa t}}{2\kappa}.$$

Now we calculate the correlation coefficient for its Itô and Stratonovich signatures, respectively.

**Itô signatures.** By the definition of signatures and Equation (A.9), we have

$$\mathbb{E}\left[S(\mathbf{X})_T^{1,1,I}\right] = \mathbb{E}\left[\int_0^T Y_t \mathrm{d}Y_t\right] = -\kappa\mathbb{E}\left[\int_0^T Y_t^2 \mathrm{d}t\right] + \mathbb{E}\left[\int_0^T Y_t \mathrm{d}W_t\right] = -\kappa\int_0^T \mathbb{E}\left[Y_t^2\right]\mathrm{d}t$$

$$= -\kappa\int_0^T \frac{1-e^{-2\kappa t}}{2\kappa}\mathrm{d}t = -\frac{T}{2} + \frac{1-e^{-2\kappa T}}{4\kappa}. \tag{A.13}$$

For the second moment, by Itô isometry, we have

$$\mathbb{E}\left[S(\mathbf{X})_T^{1,1,I}\right]^2 = \mathbb{E}\left[\int_0^T Y_t \mathrm{d}Y_t\right]^2 = \mathbb{E}\left[-\kappa\int_0^T Y_t^2 \mathrm{d}t + \int_0^T Y_t \mathrm{d}W_t\right]^2$$

$$= \kappa^2\int_0^T\int_0^T \mathbb{E}\left[Y_t^2 Y_s^2\right]\mathrm{d}t\mathrm{d}s - 2\kappa\mathbb{E}\left[\int_0^T Y_t^2\mathrm{d}t\int_0^T Y_t \mathrm{d}W_t\right] + \int_0^T \mathbb{E}\left[Y_t^2\right]\mathrm{d}t$$

$$=: (a) - (b) + (c).$$

It is easy to calculate Term (c):

$$(c) = \int_0^T \mathbb{E}\left[Y_t^2\right]\mathrm{d}t = \int_0^T \frac{1-e^{-2\kappa t}}{2\kappa}\mathrm{d}t = \frac{T}{2\kappa} + \frac{e^{-2\kappa T}-1}{4\kappa^2}. \tag{A.14}$$

To derive Term (a), we need to calculate $\mathbb{E}\left[Y_t^2 Y_s^2\right]$. Assume that $s < t$ and denote $M_t = \int_0^t e^{\kappa u}\mathrm{d}W_u$, we have $Y_t = e^{-\kappa t}M_t$, and therefore

$$\mathbb{E}\left[Y_t^2 Y_s^2\right] = e^{-2\kappa(t+s)}\mathbb{E}\left[M_t^2 M_s^2\right] = e^{-2\kappa(t+s)}\mathbb{E}\left[(M_t - M_s + M_s)^2 M_s^2\right]$$

$$= e^{-2\kappa(t+s)}\left[\mathbb{E}\left[(M_t - M_s)^2 M_s^2\right] + 2\mathbb{E}\left[(M_t - M_s)M_s^3\right] + \mathbb{E}\left[M_s^4\right]\right].$$

Because $M_t - M_s = \int_s^t e^{\kappa u}\mathrm{d}W_u$ is a Gaussian random variable with mean 0 and variance

$$\mathrm{Var}(M_t - M_s) = \mathbb{E}\left[(M_t - M_s)^2\right] = \mathbb{E}\left[\int_s^t e^{\kappa u}\mathrm{d}W_u\right]^2 = \int_s^t [e^{\kappa u}]^2\,\mathrm{d}u = \frac{e^{2\kappa t} - e^{2\kappa s}}{2\kappa},$$

and $M_t$ has independent increments, we further have

$$\mathbb{E}\left[Y_t^2 Y_s^2\right] = e^{-2\kappa(t+s)}\left[\mathbb{E}\left[(M_t - M_s)^2\right]\mathbb{E}\left[M_s^2\right] + 2\mathbb{E}\left[M_t - M_s\right]\mathbb{E}\left[M_s^3\right] + \mathbb{E}\left[M_s^4\right]\right]$$

$$= e^{-2\kappa(t+s)}\left[\frac{e^{2\kappa t} - e^{2\kappa s}}{2\kappa}\cdot\frac{e^{2\kappa s} - 1}{2\kappa} + 0 + 3\left(\frac{e^{2\kappa s} - 1}{2\kappa}\right)^2\right]$$

$$= \frac{1 + 2e^{-2\kappa t + 2\kappa s} - e^{-2\kappa s} - 5e^{-2\kappa t} + 3e^{-2\kappa t - 2\kappa s}}{4\kappa^2}$$

when $s < t$. One can similarly write the corresponding formula for the case of $s > t$ and therefore obtain that

$$(a) = \kappa^2\int_0^T\int_0^T \mathbb{E}\left[Y_t^2 Y_s^2\right]\mathrm{d}t\mathrm{d}s$$

$$= \frac{1}{4}\left(T^2 + \frac{T}{\kappa} + \frac{10Te^{-2\kappa T}}{2\kappa} + \frac{3e^{-4\kappa T}}{4\kappa^2} - \frac{9}{4\kappa^2} + \frac{3e^{-2\kappa T}}{2\kappa^2}\right).$$

For Term (b), note that

$$2\kappa\mathbb{E}\left[\int_0^T Y_t^2\mathrm{d}t\int_0^T Y_t \mathrm{d}W_t\right] = 2\kappa\int_0^T \mathbb{E}\left[Y_s^2\int_0^T Y_t \mathrm{d}W_t\right]\mathrm{d}s,$$

we therefore need to calculate $f(s) := \mathbb{E}\left[Y_s^2\int_0^T Y_t \mathrm{d}W_t\right]$ for $s < T$. To do this, by Itô's Lemma, we have

$$\mathrm{d}Y_s^2 = 2Y_s\mathrm{d}Y_s + \mathrm{d}[Y,Y]_s = -2\kappa Y_s^2\mathrm{d}s + 2Y_s\mathrm{d}W_s + \mathrm{d}s,$$

which implies that

$$Y_s^2 = -2\kappa \int_0^s Y_u^2 \mathrm{d}u + 2 \int_0^s Y_u \mathrm{d}W_u + \int_0^s \mathrm{d}u.$$

Therefore, for $s < T$, with the help of Itô isometry and Equation (A.14), we have

$$
\begin{aligned}
f(s) &= \mathbb{E}\left[ Y_s^2 \int_0^T Y_t \mathrm{d}W_t \right] \\
&= \mathbb{E}\left[ \left( -2\kappa \int_0^s Y_u^2 \mathrm{d}u + 2\int_0^s Y_u \mathrm{d}W_u + \int_0^s \mathrm{d}u \right) \int_0^T Y_t \mathrm{d}W_t \right] \\
&= -2\kappa \int_0^s \mathbb{E}\left( Y_u^2 \int_0^T Y_t \mathrm{d}W_t \right) \mathrm{d}u + 2 \int_0^s \mathbb{E}\left[ Y_t^2 \right] \mathrm{d}t + 0 \\
&= -2\kappa \int_0^s f(u)\mathrm{d}u + \frac{s}{\kappa} + \frac{e^{-2\kappa s} - 1}{2\kappa^2},
\end{aligned}
$$

and taking derivatives of both sides leads to

$$\frac{\mathrm{d}f}{\mathrm{d}s} = -2\kappa f(s) + \frac{1}{\kappa} - \frac{e^{-2\kappa s}}{\kappa}.$$

By solving this ordinary differential equation with respect to $f$ with initial condition $f(0) = 0$, we obtain that

$$f(s) = \frac{1}{2\kappa^2} - \frac{se^{-2\kappa s}}{\kappa} - \frac{e^{-2\kappa s}}{2\kappa^2}.$$

Therefore,

$$(\mathrm{b}) = 2\kappa \int_0^T f(s)\mathrm{d}s = \frac{T}{\kappa} + \frac{Te^{-2\kappa T}}{\kappa} + \frac{e^{-2\kappa T} - 1}{\kappa^2}.$$

Finally, we obtain that

$$\mathbb{E}\left[ S(\mathbf{X})_T^{1,1,I} \right]^2 = (\mathrm{a}) - (\mathrm{b}) + (\mathrm{c}) = \frac{Te^{-2\kappa T}}{4\kappa} + \frac{3e^{-4\kappa T}}{16\kappa^2} - \frac{3e^{-2\kappa T}}{8\kappa^2} - \frac{T}{4\kappa} + \frac{3}{16\kappa^2} + \frac{T^2}{4}. \quad (\mathrm{A}.15)$$

Therefore, for Itô signature, we have

$$
\begin{aligned}
\frac{\mathbb{E}\left[ S(\mathbf{X})_T^{0,I} S(\mathbf{X})_T^{1,1,I} \right]}{\sqrt{\mathbb{E}\left[ S(\mathbf{X})_T^{0,I} \right]^2 \mathbb{E}\left[ S(\mathbf{X})_T^{1,1,I} \right]^2}} &= \frac{\mathbb{E}\left[ S(\mathbf{X})_T^{1,1,I} \right]}{\sqrt{\mathbb{E}\left[ S(\mathbf{X})_T^{1,1,I} \right]^2}} \\
&= \frac{-2\kappa T - e^{-2\kappa T} + 1}{\sqrt{4\kappa Te^{-2\kappa T} + 3e^{-4\kappa T} - 6e^{-2\kappa T} - 4\kappa T + 3 + 4\kappa^2 T^2}},
\end{aligned}
$$

where we use the fact that the 0-th order signature is defined as 1.

**Stratonovich signatures.** The Stratonovich integral and the Itô integral are related by

$$\int_0^t A_s \circ \mathrm{d}B_s = \int_0^t A_s \mathrm{d}B_s + \frac{1}{2}[A, B]_t.$$

Therefore, for Stratonovich signatures, we have

$$S(\mathbf{X})_T^{1,S} = \int_0^T 1 \circ \mathrm{d}Y_t = \int_0^T 1 \mathrm{d}Y_t + \frac{1}{2}[1, Y]_T = \int_0^T 1 \mathrm{d}Y_t = S(\mathbf{X})_T^{1,I} = Y_T,$$

and

$$S(\mathbf{X})_T^{1,1,S} = \int_0^T S(\mathbf{X})_T^{1,S} \circ \mathrm{d}Y_t = \int_0^T Y_t \circ \mathrm{d}Y_t = \int_0^T Y_t \mathrm{d}Y_t + \frac{1}{2}[Y, Y]_T = S(\mathbf{X})_T^{1,1,I} + \frac{T}{2},$$

where we use the fact that $[1, Y]_T = 0$ and $[Y, Y]_T = T$. Now we can use our results for the Itô signatures, Equations (A.13) and (A.15), to obtain that

$$\mathbb{E}\left[ S(\mathbf{X})_T^{1,1,S} \right] = \mathbb{E}\left[ S(\mathbf{X})_T^{1,1,I} \right] + \frac{T}{2} = \frac{1 - e^{-2\kappa T}}{4\kappa},$$

and

$$\mathbb{E}\left[S(\mathbf{X})_T^{1,1,S}\right]^2 = \mathbb{E}\left[S(\mathbf{X})_T^{1,1,I} + \frac{T}{2}\right]^2$$

$$= \mathbb{E}\left[S(\mathbf{X})_T^{1,1,I}\right]^2 + T\mathbb{E}\left[S(\mathbf{X})_T^{1,1,I}\right] + \frac{T^2}{4} = \frac{3(1-e^{-2\kappa T})^2}{16\kappa^2}.$$

Therefore, for the Stratonovich signature, we have

$$\frac{\mathbb{E}\left[S(\mathbf{X})_T^{0,S}S(\mathbf{X})_T^{1,1,S}\right]}{\sqrt{\mathbb{E}\left[S(\mathbf{X})_T^{0,S}\right]^2 \mathbb{E}\left[S(\mathbf{X})_T^{1,1,S}\right]^2}} = \frac{\sqrt{3}}{3}.$$

This completes the proof. $\qquad\square$

*Proof of Proposition A.2.* Let $a = \#A_1^*$ and $b = \#A_1^{*c}$. Under the equal inter-dimensional correlation assumption, we have $\Sigma_{A^*,A^*} = (1-\rho)I_a + \rho\mathbf{1}_a\mathbf{1}_a^\top$, where $I_a$ is an $a \times a$ identity matrix and $\mathbf{1}_a$ is an $a$-dimensional all-one vector. In addition, $\Sigma_{A^{*c},A^*} = \rho\mathbf{1}_b\mathbf{1}_a^\top$, where $\mathbf{1}_b$ is a $b$-dimensional all-one vector. Using the Sherman–Morrison formula, we have

$$\Sigma_{A^*,A^*}^{-1} = \frac{1}{1-\rho}I_a - \frac{\rho}{(1-\rho)(1+(a-1)\rho)}\mathbf{1}_a\mathbf{1}_a^\top.$$

Therefore, because all true beta coefficients are positive, we have

$$\Sigma_{A^{*c},A^*}\Sigma_{A^*,A^*}^{-1}\text{sign}(\boldsymbol{\beta}_{A^*}) = \frac{a\rho}{1+(a-1)\rho}\mathbf{1}_a.$$

Hence, the irrepresentable condition

$$\left|\Sigma_{A^{*c},A^*}\Sigma_{A^*,A^*}^{-1}\text{sign}(\boldsymbol{\beta}_{A^*})\right| = \frac{a|\rho|}{1+(a-1)\rho}\mathbf{1}_a < \mathbf{1}_a$$

holds if and only if $\frac{a|\rho|}{1+(a-1)\rho} < 1$. One can easily verify that this holds if $\rho \in (-\frac{1}{2\#A_1^*}, 1)$, and does not hold if $\rho \in (-\frac{1}{\#A_1^*}, -\frac{1}{2\#A_1^*}]$. This completes the proof. $\qquad\square$

