# OpenReview forum: "A Note on Some Statistical Properties of Signature Transform Under Stochastic Integrals"
_ICLR.cc/2024/Conference — Submitted to ICLR 2024_

### Official Review · Reviewer_otaH · 2023-10-30

**Soundness:** 3 good
**Presentation:** 4 excellent
**Contribution:** 2 fair
**Rating:** 5
**Confidence:** 3

**Summary:**

The paper investigates signature transforms of time series data using a Lasso regression framework:

For time series resembling Brownian motion with weak correlations, Lasso regression aligns better with Ito integrals.
 For mean-reverting processes, Stratonovich integrals are more consistent.

The paper supports the theoretical findings with numerical experiments on synthetic data like Brownian motion and OU processes.

**Strengths:**

The paper is very well written, with a great review for signatures.

The propositions and the numerical results are well presented.

**Weaknesses:**

The paper is mainly concerned with the whether Ito or Stratonovic signatures to combined with LASSO will result in a higher statistical consistency. However, the investigation is limited to very specific examples (such as Brownian motions and OU processes). It would have been a stronger paper if it provided a more detailed guideline as to when each of these should be applied. Even an extensive empirical study leading to an intuitively appealing empirically supported guideline would have made the paper stronger.

**Questions:**

While it is obvious from Figure 1 that Ito signatures are more consistent, the conclusion that Stratonovic signatures are more consistent can not be drawn from Figure 2 (as far as I can see).  Is there a theoretical explanation as to why this is happening?

---

> ### Author Response · Authors · 2023-11-18
> **Response to Reviewer otaH's Comments**
>
> > ***Strengths.***
>
> We thank the reviewer for appraising the writing and presentation of our paper.
>
> > ***Weaknesses: Detailed Guideline and Empirical Study***
>
> We thank the reviewer for the comment. While our theoretical work is based on the Brownian motion and the OU process, our numerical studies also investigate the random walk (Figure 1b), AR(1) process (Figure 2b), ARIMA process (Appendix J), and time-augmented process (Appendix H). These cover many commonly used models of time series. Our main conclusions ("Our study shows that, for processes and time series that are closer to Brownian motion or random walk with weaker inter-dimensional correlations, the Lasso regression is more consistent for their signatures defined by Itô integrals; for mean reverting processes and time series, their signatures defined by Stratonovich integrals have more consistency in the Lasso regression.") in the Abstract of our paper are also based on the analysis of these results.
>
> We agree with the reviewer that conducting an empirical study would be helpful. However, we would like to point out that, with all due respect, empirical studies would not allow us to compare consistency across different definitions of the signature because it is impossible to measure consistency without knowing the true data generating process. In other words, the consistency has to be verified on a simulated dataset where the ground truth of the set of true features is known. In fact, most classical literature on Lasso consistency, including works like [1,2,3], did not incorporate empirical studies. Therefore, following this strand of literature, we focused on theoretical analysis and numerical studies.
>
> > ***Questions: Conclusion of Figure 2***
>
> We thank the reviewer's comment. We would like to clarify that in Figure 2, we did not conclude that Stratonovich signatures are consistently superior in all cases. Instead, our conclusion is that Stratonovich signatures are more consistent *when the process is sufficiently mean reverting*. In particular, in the paragraph following Figure 2, we stated the following observations:
>
> - "First, the  Itô signature reaches the highest consistency rate when $\kappa$ and $1-\phi$ approach $0$, which correspond respectively to a Brownian motion and a random walk." This means that, when $\kappa$ is small, the result is similar to a Brownian motion, and thus, Itô signatures are more consistent (the result of Figure 1). This can be observed in Figure 2 that the solid lines (Itô) are higher when $\kappa$ is small.
> - "Second, when the process is sufficiently mean reverting,  Stratonovich signatures have higher consistency rates than Itô signatures." This means that, when $\kappa$ is large ("sufficiently mean reverting"), Stratonovich signatures are more consistent. This can be observed in Figure 2 that the dashed lines (Stratonovich) are higher when $\kappa$ is large.
>
>
> Theoretically, this result can be attributed to the fact that, for the OU process, the collinearity between Itô signatures increases as $\kappa$ increases, leading to lower consistency. In contrast, the collinearity between Stratonovich signatures generally remains stable across different values of $\kappa$, which leads to their consistent performance. This theoretical explanation is included in Appendix D.
>
> **References:**
>
> [1] Zhao, P., and B. Yu, 2006, On model selection consistency of Lasso, The Journal of Machine Learning Research, 7, 2541–2563.
>
> [2] Bickel, P. J., Y. Ritov, and A. B. Tsybakov, 2009, Simultaneous analysis of Lasso and Dantzig selector, The Annals of Statistics, 1705–1732.
>
> [3] Wainwright, M. J., 2009, Sharp thresholds for high-dimensional and noisy sparsity recovery using $l_1$-constrained quadratic programming (Lasso), IEEE Transactions on Information Theory 55, 2183–2202.

---

> > ### Comment · Reviewer_otaH · 2023-11-19
> > **Thank you for your response**
> >
> > Thank you for your response. I read it carefully and I do not think that my comments are adequately addressed. I am keeping my rating.

---

> > > ### Author Response · Authors · 2023-11-19
> > > **Response to Reviewer otaH's Comment**
> > >
> > > We thank the reviewer for providing us with helpful comments and suggestions.

---

### Official Review · Reviewer_weWX · 2023-11-05

**Soundness:** 3 good
**Presentation:** 4 excellent
**Contribution:** 2 fair
**Rating:** 5
**Confidence:** 2

**Summary:**

Signature transformation transforms a time series $\mathbf{X}\_n$ into point statistics $ S(\mathbf{X}\_n)\_T^{\ldots}$. In this paper, the authors study relationship between a random process (Brownian motion, random walk, OU process, and AR(1) model) underlying $\mathbf{X}\_n$, an integral method (Itô  or Stratonovich) for defining the signature transformation, and result of signature-based Lasso: they discuss consistency properties of “estimate of sign of parameter” when performing Lasso regression for $\\{(S(\mathbf{X}\_n)\_T^{\ldots}, y\_n)\\}\_n$ with $y\_n$ generated from a linear model with noise based on $ S(\mathbf{X}\_n)\_T^{\ldots}$. Through this study, the authors state “Our study shows … the Lasso regression” and a more generic fact “Our findings highlight … and machine learning” (both of which are written in abstract).

**Strengths:**

I have understood that the authors claim “Our study shows … the Lasso regression” (claim1) and a more generic fact “Our findings highlight … and machine learning” (claim2), both of which are written in Abstract. I have focused on verifying claim 1 only since claim 2 is too generic and trivial (if this is wrong, please let me know what problems have arisen due to existing studies’ indifference to the integral method). According to this standpoint, I have wrote following comments.

1. The paper is well-written (presentation is good).
2. Propositions and Theorems are correct.
3. Feature selection techniques may be a good option to improve regression based on signature transformations. This is because users will want to use somewhat large $K$ since an appropriate value of $K$ is not known in advance, but in that case the number $\frac{d^{K+1}-1}{d-1}$ of predictors becomes large (when $d\neq 1$).
4. Discussion between Proposition 7 and Example 1 supports claim 1 to some extent.
5. Program codes are reliable.

**Weaknesses:**

6. I have thought that the sentence “Given the successful application … time series data” expresses the motivation for this study. However, this motivation is too abstract. I would like to see a concrete description as to why discussion to support claim 1 is demanded.
7. Most of the related studies seem to be published by the same research group. I am concerned about this point. For example, all the application paper cited in Section 1 (except for the arXiv paper (Arribas, 2018)) were published by the same research group. I cannot know if unrelated researchers recognize the usefulness of signature-based techniques in their applications. Please tell me about some interesting applications that unrelated researchers have done. Also, I think that the authors should cite such papers as well. I think that this is needed for an ICLR paper. This comment has nothing to do with whether the authors of this paper belong to that research group.
8. In recent years, “series-to-point non-linear (neural-network-based) regression”, which is an end-to-end regression in the setting of this paper, has been well studied. For example, studies referred in Section 3.3 of “Ahmed, S., Nielsen, I. E., Tripathi, A., Siddiqui, S., Ramachandran, R. P., & Rasool, G. (2023). Transformers in time-series analysis: A tutorial. Circuits, Systems, and Signal Processing, 42(12), 7433-7466”. Is there a practical advantage to doing “point-to-point linear regression via signature transformation”? (I may have missed a cited paper that makes such a comparison. I'm sorry if that's the case. Please tell me that paper.) Considering readers of ICLR, it is worth mentioning the comparison between such techniques and signature-based ones. Also, the input dimension $\frac{d^{K+1}-1}{d-1}$ of “point-to-point linear regression via signature transformation” can be larger than the input dimension $d T$ of “end-to-end regression”. Is this a negative factor of “point-to-point linear regression via signature transformation”?
9. I would like to ask for additional explanation of interpretations of signature $ S(\mathbf{X}\_n)\_T^{i\_1,\ldots,i\_k}$ for each combination of $i\_1,\ldots,i\_k$. There are generally two main types of evaluation strategies of Lasso: regression error or parameter estimate error. This research focuses on the latter, which means emphasizing the interpretability of the parameter estimate itself. For that position, the interpretability of signatures should be important.
10. I could not understand the meaning of the experimental comparisons for Figures 1 and 2 to support claim1. When integral methods are different, the generated data (in particular, $\\{y\_n\\}$) are different, and data analysis methods (the sets of features used for Lasso regression) are also different. For example, for data analysis in real situation, only one data is given, so analysis with different generated data is useless. In particular, since the comparison uses different data, I think that there is a gap between the experimental results and claim 1. Contrary, if the authors want to focus their discussion on the influence of the integral method on the distribution properties of the signature, it would be natural to consider with a single data analysis method. Please tell me more specifically what the authors want to claim with the experimental comparisons for Figures 1 and 2, and in what situations that claim is useful.
11. Another problem is that consideration and explanation of reasons for the experimental results are not sufficiently described. (I imagine that multicollinearity among the explanatory variables $ S(\mathbf{X})\_T^{\ldots}$ influences the experimental results; is it right?)
12. Regarding “Other feature selection techniques” in Section 5. The ridge regression is not used for feature selection typically. Rather, it will be better to cite, for example, bridge regression (Frank, I. E., and Friedman, J. H. (1993), “A Statistical View of Some Chemometrics Regression Tools,” Technometrics, 35, 109–135).
13. $\\{\epsilon\_n\\}\_n$ in (3) requires 0-mean assumption at least (otherwise $\tilde{\beta}\_0$ will be biased).

**Questions:**

I also wrote questions in the previous item. Please respond to comments 6--13. I will raise my rating if I receive satisfactory responses.

---

> ### Author Response · Authors · 2023-11-18
> **Response to Reviewer weWX's Comments**
>
> > ***Strengths.***
>
> We thank the reviewer for appraising the writing, rigor, and programming of our paper.
>
> While Claim 2 may appear to be generic or trivial, we would like to clarify that we believe it actually constitutes an important contribution from our work. To the best of our knowledge, no existing literature explicitly compares the differences in statistical performance between different signature definitions. In particular, most machine learning studies related to signatures use the default (Stratonovich) signatures directly without a comparative analysis with Itô signatures. We are the first to highlight the importance of choosing an appropriate definition of signatures both from a theoretical perspective and using numerical examples. We believe that the literature can achieve improved performance by adopting a more suitable signature definition for each of their specific application. Therefore, we consider the claim "choosing appropriate definitions of signatures and stochastic models in statistical inference and machine learning" (Claim 2) to be important and has often been overlooked in the past.
>
>
> > ***Weakness 6: Motivation***
>
> We appreciate the reviewer's feedback and we agree wholeheartedly with the reviewer that establishing the motivation for our research is important. In the revised manuscript, we have modified the paragraph to better convey the motivation behind our work.
>
> We believe that discussion to support Claim 1 is necessary for three key reasons. First, to the best of our knowledge, most empirical literature on signatures uses signatures directly without considering their underlying statistical properties. Second, the probabilistic foundation of empirical research using signatures, the universal nonlinearity, has been explored under different definitions of signatures. However, the associated statistical properties for different definitions of signatures have not been examined. Third, to our knowledge, most empirical research uses a default definition of signatures regardless of the specific scenarios under investigation. However, using an inappropriate signature definition may lead to suboptimal performance, as highlighted by our work. Together, these three reasons motivate us to study the statistical performance of signatures.
>
> > ***Weakness 7: Literature***
>
> We thank the reviewer's comment, and we fully agree with the reviewer that it is important to provide a more diverse set of references. In the revised manuscript, we have included references to research from a much wider set of research groups. For example, [1] makes a broad survey of signature applications and demonstrates the superior performance of signatures using diverse datasets including a drawing dataset (Quick, Draw!), a sound dataset (Urban Sound), and smartphone action sensory dataset (MotionSense); [2] uses the signature transform to extract speak-and-pause patterns and achieve superior results on three publicly available datasets; [3,4,5,6] explore the applications of the signature method in finance.
>
> > ***Weakness 8: Series-to-Point Regression***
>
> We appreciate the reviewer's insightful question and agree that comparing neural-network-based regression with the signature-based model can make our paper more appealing to readers of ICLR.
>
> **Practical advantages of signature transform.** We believe that the signature transform has two practical advantages. First, it enhances computational efficiency in signature-based models because it only requires training a linear model; second, the linear model allows for interpretability. Essentially, the signature transform can be understood as a powerful method of feature engineering with sound theoretical properties. We have incorporated this comparison into Section 1 of the revised manuscript and have provided an interpretation of signatures in Appendix A.
>
> **Comparison between input dimensions.** We acknowledge the reviewer's concern about potentially higher input dimensions for the signature-based model ($\frac{d^{K+1}-1}{d-1}$) compared to the neural-network-based model ($dT$). However, we do not think this is a negative factor for the signature-based model for two main reasons.
>
> First, signatures are computed using the original $dT$ input data. Therefore, $dT$ can also be regarded as the input dimension for the signature-based model. On the other hand, neural-network-based models typically involve a large number of parameters to be trained, significantly increasing their overall dimensions. In fact, as mentioned above, the signature transform itself is a feature engineering step that allows the subsequent step to use a simple linear model.
>
> Second, $\frac{d^{K+1}-1}{d-1}$ is not a concern in practice because, it has been documented that including signatures up to a small order $K$ usually suffices to achieve good performances [7,8] (see Section 2.2 of our paper).

---

> > ### Author Response · Authors · 2023-11-18
> > **Response to Reviewer weWX's Comments (Cont'd)**
> >
> > > ***Weakness 9: Interpretation***
> >
> > We thank the reviewer for the comment, and we agree wholeheartedly with the reviewer that it is important to derive intuition and interpretation from signatures. In the revised manuscript, we have added a new appendix, Appendix A, which introduces the geometric interpretation of signatures. This interpretation has been documented in previous works such as [9].
> >
> > > ***Weakness 10: Experimental Comparison***
> >
> > We thank the reviewer's question. Our objective is to compare the model consistency achieved using different *data analysis methods* (the set of features used for Lasso regression). More precisely, we want to investigate whether the consistency of Lasso is higher when using Itô signatures or Stratonovich signatures as the feature set.
> >
> > To ensure that the consistency is well defined, it is crucial that the set of features included in the Lasso ($A^* \cup A^{\*c}$) contains all true features of the true model ($A^*$). Therefore, we have to assume that $y_n$ is generated based on either Itô or Stratonovich signatures depending on which type of signatures we are using as features. Hence, when using different integral methods, different data $y_n$ are generated, and different data analysis methods are used.
> >
> > We understand that this setup might seem confusing when comparing Itô and Stratonovich signatures. However, because our goal is to compare different data analysis methods, we have to adopt this setup. For example, assume that the data $y_n$ is generated using Itô signatures. Then, given the data, it would be unfair to compare the consistency of Itô signatures with that of Stratonovich signatures, as the latter would be undefined (or, alternatively, we can say the latter one is zero, because no true signatures could be selected).
> >
> > In our numerical studies presented in Figures 1 and 2, we compute the consistency rates for Itô (Stratonovich) signatures assuming that the data $y_n$ is generated by Itô (Stratonovich) signatures, and make a comparison between them. This is useful in the following sense. For example, assume that we are given a set of real data $\{(\mathbf{X}_n, y_n)\}$, where $\mathbf{X}_n$ is a Brownian motion. We do not know whether $y_n$ is generated by Itô or Stratonovich signatures of $\mathbf{X}_n$. From the Bayesian perspective, we have a noninformative prior that $y_n$ is generated by either Itô or Stratonovich signatures each with a probability of 50\%. Our numerical results in Figure 1 imply that using Itô signatures to analyze the data would yield a higher expected model consistency, because
> > $$
> > 0.5 \times \text{Itô signatures' consistency rate} + 0.5 \times 0 > 0.5 \times 0 + 0.5 \times \text{Stratonovich signatures' consistency rate}.
> > $$
> >
> >
> > > ***Weakness 11: Result Explanation***
> >
> > We thank the reviewer for the comment. Indeed, the reviewer is correct that the collinearity between $S(\mathbf{X})_T$ influences the results. This collinearity also depends on the integral's definition. In our revised manuscript, we have added additional explanations to our results in Section 4.1.
> >
> > > ***Weakness 12: Bridge Regression***
> >
> > We appreciate the reviewer's feedback. We have revised the manuscript according to the reviewer's suggestion.
> >
> > > ***Weakness 13: Residual Term***
> >
> > We appreciate the reviewer's comment. In the revised manuscript, we have clarified that $\varepsilon_n$ should have a zero mean and finite variance.

---

> > > ### Author Response · Authors · 2023-11-18
> > > **References**
> > >
> > > [1] Fermanian, A., 2021, Embedding and learning with signatures, Computational Statistics \& Data Analysis 157, 107148.
> > >
> > > [2] Pan, Y., M. Lu, Y. Shi, and H. Zhang, 2023, A path signature approach for speech-based dementia detection, IEEE Signal Processing Letters.
> > >
> > > [3] Akyildirim, E., M. Gambara, J. Teichmann, and S. Zhou, 2022, Applications of signature methods to market anomaly detection, arXiv preprint arXiv:2201.02441.
> > >
> > > [4] Cuchiero, C., G. Gazzani, and S. Svaluto-Ferro, 2023, Signature-based models: Theory and calibration, SIAM Journal on Financial Mathematics 14, 910–957.
> > >
> > > [5] Futter, O., B. Horvath, and M. Wiese, 2023, Signature trading: A path-dependent extension of the mean-variance framework with exogenous signals, arXiv preprint arXiv:2308.15135.
> > >
> > > [6] Lemahieu, E., K. Boudt, and M. Wyns, 2023, Generating drawdown-realistic financial price paths using path signatures, arXiv preprint arXiv:2309.04507.
> > >
> > > [7] Morrill, J., A. Fermanian, P. Kidger, and T. Lyons, 2020, A generalised signature method for multivariate time series feature extraction, arXiv preprint arXiv:2006.00873.
> > >
> > > [8] Lyons, T., and A. D. McLeod, 2022, Signature methods in machine learning, arXiv preprint arXiv:2206.14674.
> > >
> > > [9] Levin, D., T. Lyons, and H. Ni, 2016, Learning from the past, predicting the statistics for the future, learning an evolving system, arXiv preprint arXiv:1309.0260.

---

> > > > ### Comment · Reviewer_weWX · 2023-11-19
> > > > **Thank you for your response: 2nd comment**
> > > >
> > > > **Weaknesses 7 and 11-13 and regarding claim 2 and "Comparison between input dimensions" in Weakness 8:**
> > > >
> > > > Thank you for your responses.
> > > > I am satisfied with these responses.
> > > >
> > > > **Weakness 6 and 10:**
> > > >
> > > > Thank you for your responses.
> > > > However, the revised description on the motivation remains abstract as that to support claim 1.
> > > > What's the point of having high sign congruency rate? I do not recognize this point.
> > > > For example, consider an extreme example where data are generated with $S(\mathbf{X})\_T^{1}= S(\mathbf{X})\_T^{2}$, $\beta\_1=1$, and $\beta\_2=0$.
> > > > Under this example, difference between an estimate with $\tilde{\beta}\_1=1, \tilde{\beta}\_2=0$ and an estimate with $\tilde{\beta}\_1=0, \tilde{\beta}\_2=1$ is meaningless.
> > > > This example is too extreme, but I think that it has small significance to compare sign consistency rates under presence of multicollinearity.
> > > >
> > > > For XdRo's Comments "Weakness 2", the authors replied "We choose sign consistency as the starting point of our research because it is a basic and well-accepted metric for studying Lasso [4,5,6]."
> > > > However, in [4,5,6], the dependence of the consistency rate of the Lasso estimator on the number of training samples was only discussed (there, the data distribution and method were the same).
> > > >
> > > > I understand your response, but I still don't see any significance in comparing sign consistency rates for different data distributions or different methods.
> > > > You may provide additional responses, but I will probably not change my opinion on this point.
> > > >
> > > > The current paper moves the discussion from the multicollinearity issue to a comparison of sign consistency rates.
> > > > We agree that the multicollinearity issue could be beneficial to claim 2.
> > > > How about moving the discussion from the multicollinearity issue to another topic?
> > > > For example, it might be useful if an argument such as "to achieve the same level of prediction performance, $K$ can be smaller (more computationally efficient) when using an integral method such that the signatures have lower collinearity" were provided along with real-world data experiments.
> > > > (I am not suggesting that this submission should be rewritten to such discussions. In the case where this submission is rejected, please refer to it.)
> > > >
> > > > **"Comparison between neural-network-based regression with the signature-based" in Weakness 8:**
> > > >
> > > > I agree with the two points you mentioned as advantages of signature transformation-based point-to-point regression.
> > > > However, I am more interested in whether that point-to-point regression can outperform neural-network-based series-to-point regression regarding the prediction performance.
> > > > This is because the two points mentioned by the authors are not often prioritized over predictive performance in practice in my sense.
> > > > I do not require the introduction of experimental studies showing that signature-based regression significantly outperforms series-to-point regression with respect to prediction performance.
> > > > I just want to be sure that the former can give a prediction performance comparable to the latter.
> > > >
> > > > Also, the paper (Levin et al., 2016) (its initial version is published in 2013) is old.
> > > > The methodology of series-to-point regression has made significant progress in the last five years.
> > > >
> > > > I will comment again with more specificity:
> > > > Please cite studies that compare a series-to-point regression methods proposed after 2018 and signature transformation-based regression methods with respect to predictive performance.
> > > >
> > > > **Weakness 9:**
> > > >
> > > > Is there a difference between Ito integral and Stratonovich integral?

---

> > > > > ### Author Response · Authors · 2023-11-19
> > > > > **Response to Reviewer weWX's 2nd Comment**
> > > > >
> > > > > > ***Weaknesses 6 and 10.***
> > > > >
> > > > > We thank the reviewer for pointing out the caveat regarding Claim 1 and acknowledging our contribution regarding Claim 2. We also greatly appreciate the reviewer for suggesting a potential direction for studying the practical influence of multicollinearity. This is helpful and we will follow the reviewer's suggestion in our future work.
> > > > >
> > > > > > ***"Comparison between neural-network-based regression with the signature-based" in Weakness 8.***
> > > > >
> > > > > We thank the reviewer for the additional comment, and we fully understand the reviewer's concern about the prediction performance of signature-based regression. In the revised manuscript, we have followed the suggestion to include more recent papers in the second paragraph of Section 1. For example, [1] use signature-based regression (referred to as "SigLasso" in the paper) to forecast the growth rate of hospitalizations in France during the COVID-19 pandemic, and the signature-based method outperforms neural-network-based methods.
> > > > >
> > > > > > ***Weakness 9.***
> > > > >
> > > > > We appreciate the reviewer's additional question. Indeed, there is a distinction between the Itô and Stratonovich integrals. In the revised manuscript, we have modified Appendix A to provide further clarification. Specifically, we explain that when the underlying process is stochastic, the difference between the Itô and Stratonovich signatures is a quadratic variation term.
> > > > >
> > > > > **Reference:**
> > > > >
> > > > > [1] Bleistein, L., A. Fermanian, A.-S. Jannot, and A. Guilloux, 2023, Learning the dynamics of sparsely observed interacting systems, arXiv preprint arXiv:2301.11647.

---

> > > > > > ### Comment · Reviewer_weWX · 2023-11-19
> > > > > > **Thank you for your response: 3rd comment**
> > > > > >
> > > > > > **"Comparison between neural-network-based regression with the signature-based" in Weakness 8**
> > > > > >
> > > > > > I am satisfied with the results of (Bleistein et al., 2023) and with citing it in your paper.
> > > > > > Sorry, I have overlooked it.
> > > > > > However, I think that it may be better to cite such paper at 'the end of the 1st paragraph in Section 1' or around 'the sentence "The nonlinearity property $\ldots$ nonlinear methods (Ahmed et al., 2023)." in the 2nd paragraph in Section 1'.
> > > > > >
> > > > > > **Weakness 9**
> > > > > >
> > > > > > Thank you for your responses. I am satisfied with the response.

---

> > > > > > > ### Author Response · Authors · 2023-11-19
> > > > > > > **Response to Reviewer weWX's 3rd Comment**
> > > > > > >
> > > > > > > > ***"Comparison between neural-network-based regression with the signature-based" in Weakness 8.***
> > > > > > >
> > > > > > > We appreciate the additional comment from the reviewer. In the latest version of our revised manuscript, we have followed the reviewer's suggestion and included a citation to the paper at the end of the first paragraph in Section 1.

---

> ### Comment · Reviewer_weWX · 2023-11-19
> **Sorry 3rd comment was sloppy: 4th comment**
>
> **"Comparison between neural-network-based regression with the signature-based" in Weakness 8.**
>
> Sorry, the previous comment was inaccurate.
> I used 'the end of the 1st paragraph in Section 1' to imply the inside of "including handwriting recognition $\ldots$; Futter et al., 2023; Lemahieu et al., 2023)".
> The authors have increased the citation in the section regarding "comprehensive reviews".
> Citing it there is inappropriate.
> My comment was sloppy. Sorry.
> I thought that it may be better to cite (Bleistein et al., 2023) in '"including $\ldots$" of the 1st paragraph in Section 1' or around 'the sentence "The nonlinearity property nonlinear methods (Ahmed et al., 2023)." in the 2nd paragraph in Section 1'.
> Moreover, I think that multiple papers showing advantages over end-to-end neural-network-based, including (Bleistein et al., 2023), (if exists) should be cited around 'the sentence "The nonlinearity property nonlinear methods (Ahmed et al., 2023)." in the 2nd paragraph in Section 1'.

---

> > ### Author Response · Authors · 2023-11-19
> > **Response to Reviewer weWX's 4th Comment**
> >
> > We appreciate the reviewer for the helpful clarification, and apologize for the misunderstanding. In the latest version of our revised manuscript, we have followed the reviewer's suggestions. In particular, we have cited Bleistein's paper in the appropriate location in the 1st paragraph of Section 1, and cited papers showing advantages over end-to-end neural-network-based models in the sentence "The nonlinearity property nonlinear methods (Ahmed et al., 2023)." in the 2nd paragraph of Section 1. We greatly appreciate the reviewer for the detailed instructions.

---

> > > ### Comment · Reviewer_weWX · 2023-11-19
> > > **Thank you for your response: 5th (final) comment**
> > >
> > > I get satisfied with the responses to all comments except "Weaknesses 6 and 10".
> > > Regarding "Weaknesses 6 and 10", I will share both the authors' attempts and my opinion with the area chair.
> > > Good luck.

---

> > > > ### Author Response · Authors · 2023-11-19
> > > > **Response to Reviewer weWX's 5th Comment**
> > > >
> > > > We thank the reviewer for providing us with helpful comments and suggestions.

---

### Official Review · Reviewer_XdRo · 2023-11-08

**Soundness:** 3 good
**Presentation:** 3 good
**Contribution:** 2 fair
**Rating:** 6
**Confidence:** 3

**Summary:**

The paper presents an analysis of signature transforms applied to Lasso regression of synthetic continuous time series data. Signature transforms are known for their ability to linearize the problem of feature selection.

The main contribution of the paper is the exploration of the correlations of these transforms when applied to Brownian motions and Ornstein-Uhlenbeck processes. This is done for signatures defined by Ito and Stratonovich integrals. They achieve this by analyzing the signatures' definitions in the context of different stochastic integrals and directly manipulating the resulting expressions. Then, using the Irrepresentable Condition (which, as proved in [Zhao & Yu '06], is almost equivalent to sign consistency of the Lasso estimator), study the consistency of the Lasso estimator when applied to the inference of labels generated from the signature transforms of these stochastic processes.

These results are then contrasted with numerical simulations that compare the performance of this signature-based regression under both Ito and Stratonovich integral definitions. Their findings suggest concrete settings under which signature transforms defined over one or the other integral definitions should be preferable.

**Strengths:**

The paper is structured in a clear way and is well written, which makes the results easy to understand. While the results are mainly theoretical, the authors contrast them with numerical simulations which allows them to strengthen their conclusions. Also, Proposition 7 an easy-to-check condition to establish the sign consistency of the Lasso estimator studied. Finally, the discussion regarding the preference for different integral definitions in signature transforms is interesting and could provide insights that could be relevant for further research in the field.

**Weaknesses:**

There are two main issues that prevent me from recommending the acceptance of the work:

- It is not entirely clear to me that Lasso regression combined with signature transforms constitutes a widespread methodology used by numerous researchers/practitioners. The references listed do not seem to be enough to support this. Furthermore, in the introduction it is claimed that, on several important Machine Learning problems, this methodology yields state-of-the-art results. But many of the references provided to support this are rather old for the standards of the field. Thus, it is not clear that this is indeed the case. All lead me to think that, although the paper has clear merits, it could maybe be a better fit for a more specialized venue than ICLR. However, if the authors could provide further references to change this opinion, I would be willing to upgrade my evaluation.

- The second one being that consistency does not appear to be the most relevant performance metric for this kind of problems. First because it only describes the large sample limit of the estimator and does not give insights about its performance for, the usually more realistic, finite sample scenarios. But also because, as discussed by the authors themselves, this measure can be too restrictive. It is true that the authors go on to explore other performance metrics in the appendices; but they do so only in a numerical way. Finally, sign consistency cannot be defined in a misspecified setting.

**Questions:**

In this section I list some minor questions and suggestions aside from the major ones expressed in the "Weaknesses" section.

- Although the results presented rely on the Irrepresentable condition, little is discussed about it. It is also stated that it "is almost a necessary and sufficient condition for Lasso estimator to be sign consistent". But the meaning of "almost" is never explained. It is my opinion that, if some further discussion were added on this respect, the results could be better appreciated.
- Is the bound in Proposition 7 expected to be tight under certain regime? When is it loose? It would be good if some details about this could be added.
- In the discussion of the main results it is said that, because experimentally good regression results are obtained for fairly small K, then this bound "can be fairly easy to verify". I think this phrase is somewhat vague.
- I think it would be good if some heuristic interpretation of why different performances for signatures defined with Ito and Stratonovich integrals are observed.

[following discussion with the authors I raise my rating]

---

> ### Author Response · Authors · 2023-11-18
> **Response to Reviewer XdRo's Comments**
>
> > ***Strengths.***
>
> We thank the reviewer for appraising the contribution and writing of our paper.
>
> > ***Weakness 1: References***
>
> We appreciate the reviewer's feedback. In the revised manuscript, we have included more related literature which provides evidence that the signature has been increasingly adopted in various tasks in machine learning. First, we have incorporated additional literature on the combination of Lasso with signature transforms in the paragraph "Consistency of Lasso" in Section 1. Second, we have included more recent papers when discussing the applications of signatures in various fields in the first paragraph of Section 1. Third, to illustrate the alignment of our paper with ICLR, we have included several signature-related references that are accepted by top computer science conferences, such as [1] at ICLR, [2] at NeurIPS, and [3] at CVPR. We hope that these additional discussions of related literature enhance the completeness of our manuscript and help address the reviewer's concerns.
>
> > ***Weakness 2: Performance Metric***
>
> We thank the reviewer for this comment. We choose sign consistency as the starting point of our research because it is a basic and well-accepted metric for studying Lasso [4,5,6]. In addition, although our theoretical results are based on large sample limits, the finite sample numerical results presented in our paper align with these theoretical results, implying that the insights gained from the large sample analysis can be informative in finite sample scenarios as well. We also believe, with all due respect, it is fairly standard in the literature to use large sample asymptotics for theoretical foundations and verify its practical effectiveness in finite sample using simulation.
>
> In the meantime, we completely agree with the reviewer that exploring theoretical results in finite sample settings and considering alternative metrics beyond consistency is valuable. Other metrics, such as the out-of-sample $R^2$, may also alleviate the reviewer's concern that sign consistency is undefined in a misspecified setting. Our paper is an initial exploration of the statistical properties related to signature selection using Lasso, which does not exist in the literature. Therefore we choose consistency as the starting point. We would like to conduct more in-depth research on the finite sample case and other metrics in our future work.
>
> > ***Question 1: Irrepresentable Condition***
>
> We thank the reviewer for the feedback. In our revised manuscript, we have added more explanations regarding the irrepresentable condition and clarified the meaning of "almost" in Appendix C. In particular, we introduce different versions of sign consistency and irrepresentable condition, and explain their relationship. In fact, we borrow the word "almost" precisely from the classical reference of [4] in this literature.
>
> > ***Question 2: Tightness of Bound***
>
> We appreciate the reviewer's insightful question. In Appendix E of our revised manuscript, we investigate a specific scenario where the inter-dimensional correlations of the Brownian motion, denoted as $\rho = \rho_{ij}$, are the same. In this setup, we prove that the bound given by Proposition 7 is tight when $\rho<0$, and is loose when $\rho>0$. This is a very helpful supplement to our manuscript, so we really appreciate the reviewer for raising this question.
>
> > ***Question 3: Vague Phrase***
>
> We thank the reviewer's comment. We have clarified this discussion in our revised manuscript. Because $K$ is usually small, $\max_{0\leq k \leq K} \{ \sharp A_k^* \}$ is also not expected to be large. Therefore, the upper bound given by Equation (12) in our main article, $\frac{1}{2\max_{0\leq k \leq K} \{\sharp A_k^* \} -1}$, should not be too small, i.e., it can be fairly easy to satisfy.
>
> > ***Question 4: Heuristic Interpretation***
>
> We thank the reviewer's feedback, and we agree with the reviewer wholeheartedly that adding heuristic interpretations can improve our manuscript. In fact, the different performances of Itô and Stratonovich signatures can be intuitively attributed to the difference in the definitions of Itô and Stratonovich integrals. In our revised manuscript, we have added these explanations at the end of Section 4.1.

---

> ### Author Response · Authors · 2023-11-18
> **References**
>
> [1] Lee, J., J. Jeon, S. yon Jhin, J. Hyeong, J. Kim, M. Jo, K. Seungji, and N. Park, 2022, LORD: Lower-dimensional embedding of log-signature in neural rough differential equations, ICLR.
>
> [2] Salvi, C., M. Lemercier, C. Liu, B. Horvath, T. Damoulas, and T. Lyons, 2021, Higher order kernel mean embeddings to capture filtrations of stochastic processes, NeurIPS.
>
> [3] Ibrahim, M. R., and T. Lyons, 2022, ImageSig: A signature transform for ultra-lightweight image recognition, CVPR.
>
> [4] Zhao, P., and B. Yu, 2006, On model selection consistency of Lasso, The Journal of Machine Learning Research, 7, 2541–2563.
>
> [5] Bickel, P. J., Y. Ritov, and A. B. Tsybakov, 2009, Simultaneous analysis of Lasso and Dantzig selector, The Annals of Statistics, 1705–1732.
>
> [6] Wainwright, M. J., 2009, Sharp thresholds for high-dimensional and noisy sparsity recovery using $l_1$-constrained quadratic programming (Lasso), IEEE Transactions on Information Theory 55, 2183–2202.

---

> > ### Comment · Reviewer_XdRo · 2023-11-20
> > **On the Authors' response**
> >
> > Thank you very much for your reply to my comments. Although you have provided some further citations to back the fact that signature transforms are currently being proposed for certain tasks, these references seem to address different models that do not include Lasso regression. Therefore, I still do not see if the theoretical study of this model would be a problem that could have a deep impact within the ICLR community. I will therefore keep the previous rating.

---

> > > ### Author Response · Authors · 2023-11-21
> > > **Response to Reviewer XdRo's Comment**
> > >
> > > We thank the reviewer's additional feedback. We would like to clarify that, in our latest version of the manuscript, we have incorporated additional literature on the combination of Lasso with signature transforms in the paragraph "Consistency of Lasso" in Section 1 (see also our previous response). For example:
> > > - [1] combines signatures with Lasso (referred to as "SigLasso" in the paper) to forecast the growth rate of hospitalizations in France during the COVID-19 pandemic;
> > > - [2] combines signatures with Lasso to deal with learning problems whose labels are only available for groups of inputs instead of individual inputs;
> > > - [3] uses Lasso to select signatures for identifying people whose diagnosis subsequently converts to Alzheimer's disease;
> > > - [4] uses Lasso to select signatures generated by temperature and salinity data;
> > > - [5] combines signatures of the time series of climate indices with Lasso to predict El Niño;
> > > - [6] combines signatures with Lasso to approximate drawdowns in financial paths.
> > > - As review articles, for example, [7] and [8] both highlighted the combination of signature transform with regularized regression such as Lasso as a useful technique in time series analysis using signatures.
> > >
> > > Therefore, with all due respect, we believe that the combination of signatures with Lasso is a widely and increasingly adopted approach in machine learning, particularly at the intersection of machine learning and time series analysis in domains such as healthcare, finance, and climate analysis. We sincerely believe that our analysis provides a useful basis for learning using the signature transform and fits the ICLR community.
> > >
> > > **References:**
> > >
> > > [1] Bleistein, L., A. Fermanian, A.-S. Jannot, and A. Guilloux, 2023, Learning the dynamics of sparsely observed interacting systems, in Proceedings of the 40th International Conference on Machine Learning, ICML’23, 2603–2640.
> > >
> > > [2] Lemercier, M., C. Salvi, T. Damoulas, E. Bonilla, and T. Lyons, 2021, Distribution regression for sequential data, in International Conference on Artificial Intelligence and Statistics, 3754–3762, PMLR.
> > >
> > > [3] Moore, P., T. Lyons, J. Gallacher, and A. D. N. Initiative, 2019, Using path signatures to predict a diagnosis of Alzheimer’s disease, PloS one 14, e0222212.
> > >
> > > [4] Sugiura, N., and S. Hosoda, 2020, Machine learning technique using the signature method for automated quality control of Argo profiles, Earth and Space Science 7, e2019EA001019.
> > >
> > > [5] Sugiura, N., and S. Kouketsu, 2021, Simple El Niño prediction scheme using the signature of climate time series, arXiv preprint arXiv:2109.02013.
> > >
> > > [6] Lemahieu, E., K. Boudt, and M. Wyns, 2023, Generating drawdown-realistic financial price paths using path signatures, arXiv preprint arXiv:2309.04507.
> > >
> > > [7] Lyons, T., and A. D. McLeod, 2022, Signature methods in machine learning, arXiv preprint arXiv:2206.14674.
> > >
> > > [8] Cuchiero, C., G. Gazzani, and S. Svaluto-Ferro, 2023, Signature-based models: Theory and calibration, SIAM Journal on Financial Mathematics 14, 910–957.

---

> > > > ### Comment · Reviewer_XdRo · 2023-11-22
> > > > **On the Authors' follow up**
> > > >
> > > > Thank you for your response. My comment regarded the references [1-3] provided in your first response. But I see that you have a revised version (not uploaded yet) where there is a stronger motivation for the study of this model. I am now more convinced that this is indeed a methodology that is capturing more attention in the community. I will therefore raise my rating.

---

> > > > > ### Author Response · Authors · 2023-11-22
> > > > > **Response to Reviewer XdRo's Comment**
> > > > >
> > > > > We are grateful for the reviewer's comments and suggestions, and for acknowledging our revision and the importance of our work to the community. We really appreciate that.

---

### Official Review · Reviewer_PPAS · 2023-11-09

**Soundness:** 4 excellent
**Presentation:** 3 good
**Contribution:** 4 excellent
**Rating:** 8
**Confidence:** 4

**Summary:**

This paper studies the Lasso consistency of signature transformation of time series. A signature of order k for a d dimensional time series is defined as a path integral of the time series over a sequence of indices $i_1, i_2, \cdots, i_k$. Signatures are powerful tools in machine learning.  The universal non-linearity property states that any continuous function of the time series may be approximated arbitrarily well by a linear function of its signature.
This paper studies the consistency issue of Lasso for signature transforms.  Feature selection with lasso regression has been studied extensively in the literature. Consistency is an important metric for out-of-sample model performance. This paper determines which signature gives a more lasso consistency of a given time series. Particularly, Ito integrals are more suitable for time series  close to
Brownian motion or random walk; whereas Stratonovich integrals have more consistency for mean reverting processes.

**Strengths:**

This is a solid paper that provides both a theoretical and numerical study of the lasso consistency of different signatures. The paper rigorously defines and analyzes the consistency of Lasso regression in signature transformations. Given the useful properties of signature transformations in feature selection, this is a nice result determining which signature transform provides better consistency for Lasso regression.

The paper reads well and seems correct, but I did not check all the proofs.

**Weaknesses:**

I do not see any major weakness in the paper. I think the paper is a bit compressed, which probably makes it harder to read for a broader set of readers. It would be nice if the authors gave more explanations of their results in the main body of the paper.

**Questions:**

What is the domain of $\beta$'s values in (3)? How does this equation relate to the feature selection problem where one selects a subset of features from a the larger original collection of the features?

---

> ### Author Response · Authors · 2023-11-18
> **Response to Reviewer PPAS's Comments**
>
> > ***Strengths.***
>
> We thank the reviewer for appraising the soundness, rigor, and contribution of our paper.
>
> > ***Weaknesses: More Explanations***
>
> We thank the reviewer for the suggestion to help us increase the readability of our paper. We have provided additional explanations for our results within the main body of our paper. For example, we have discussed the relationship with feature selection in Section 2.2 and offered heuristic interpretations for different performances of Itô and Stratonovich signatures in Section 4.1. We have added these explanations while adhering to the constraint of maintaining the paper within its nine-page limit.
>
> > ***Questions: Domain of $\beta$ and Relationship with Feature Selection***
>
> We thank the reviewer for the question.
>
> **Domain of $\beta$'s values:** The parameter $\beta$ may take any real number. The linear expression (3) is based on universal nonlinearity of signatures, which is proven in previous work such as [1]. These proofs do not impose any restrictions on the domain of $\beta$ values.  In particular, if the signature does not contribute to explaining the target variable $y_n$, the corresponding $\beta$ is zero.
>
> **Relationship with feature selection.** Equation (3) is a linear model with signatures as features, and only signatures with nonzero beta coefficients can explain the target variable, $y_n$. Therefore, we expect to select signatures with nonzero beta coefficients from all signatures included in the linear model (3). This is essentially a feature selection problem, where signatures serve as features, and our goal is to select the subset of signatures with nonzero beta coefficients from all features. This is also the reason why the literature uses feature selection methods, such as the Lasso, to select signatures. We have added more explanation to clarify this relationship in Section 2.2 of the revised manuscript.
>
> **Reference:**
>
> [1] Levin, D., T. Lyons, and H. Ni, 2016, Learning from the past, predicting the statistics for the future, learning an evolving system, arXiv preprint arXiv:1309.0260.

---

### Author Response · Authors · 2023-11-18
**Global Response**

We thank the reviewers for their time and comments on our work. We are pleased that the reviewers found that the paper is clearly structured, well-written, and easy to read (Reviewers XdRo, weWX, and otaH), and that the theoretical and numerical results are solid, rigorous, and reliable (Reviewers PPAS and weWX). Their feedback is instructive for us to improve the manuscript.

We provide detailed responses to each reviewer's comments in separate rebuttals. The revised manuscript after incorporating the questions and suggestions raised by the reviewers is also attached.

---

### Meta-Review · Area_Chair_yPTC · 2023-12-11

**Metareview:**

**Summary**

This paper is concerned with nonlinear regression from continuous time series via the signature transform. Specifically, it studies the sign consistency of the Lasso estimator when either the Itô signatures or the Stratonovich signatures are used.

**Strengths**

This paper provides concrete analysis on the correlation structures of the signature transform of multidimensional Brownian motions and multidimensional OU processes using the Itô signatures and the Stratonovich signatures, as summarized in Propositions 2, 4, 5, which allows the author to apply the sign consistency result by Zhao & Yu (2006), yielding the main results of this paper, Propositions 6-8. These arguments would stimulate further theoretical studies on statistical properties of the signature-transform-based data analysis.

**Weaknesses**

The rating/confidence of the four reviewers were 8/4, 6/3, 5/2, 5/3, with two of them having rated below the acceptance threshold. Although some of the concerns raised by the reviewers have been addressed properly in the revision, the author response did not seem to have fully addressed all the concerns raised. In particular, as reviewers weWX and otaH mentioned, this paper only considers very specific examples (multidimensional Brownian motions and multidimensional OU processes), for which the correlation structures of the signatures are fully specified in Propositions 2, 4, and 5. These observations would make the problem discussed in this paper to the sign consistency of the Lasso estimator where the covariates have the prescribed correlation structures, to which the existing result by Zhao & Yu (2006) is readily applicable, as can be seen in the proofs of the relevant propositions in Appendix H. Furthermore, when one deals with real-world data one typically may not know what the data-generation process underlying the data, which would make the practical significance of the contributions of this paper quite obscure.

Upon my own reading of this paper, I noticed the following minor points which would need appropriate revision.
- Page 3, lines 14-15: "expected signatures" do they mean expectations of the signatures?
- Page 4, line 13: On the right-hand side of the equation, the denominator is the square root of the **sum** of the squares of the signature samples, not the square root of the **sample average** of the squares. I did not understand why the sum was used.

**Justification For Why Not Higher Score:**

Two reviewers rated this paper below the acceptance threshold.  Among several concerns raised by the reviewers, the following point would prevent me from positively evaluating this paper the most: Although the main contributions of this paper were obtained by assuming the data-generation process as either the Itô signatures or the Stratonovich signatures, when one deals with real-world data one typically may not know what the data-generation process underlying the data, which would make the practical significance of the contributions of this paper quite obscure.

**Justification For Why Not Lower Score:**

N/A

---

### Decision · Program_Chairs · 2024-01-16

Reject